

# Evaluation and Error Apportionment of an Ensemble of Atmospheric Chemistry Transport Modelling Systems: Multi-variable Temporal and Spatial Breakdown

Efisio Solazzo[1*], Roberto Bianconi[2], Christian Hogrefe[3], Gabriele Curci[4,5], Ummugulsum Alyuz[6], Alessandra Balzarini[7], Rocío Baró[8], Roberto Bellasio[2], Johannes Bieser[9], Jørgen Brandt[10], Jesper H. Christensen[10], Augustin Colette[11], Xavier Francis[12], Andrea Fraser[13], Marta Garcia Vivanco[11,14], Pedro Jiménez-Guerrero[8], Ulas Im[10], Astrid Manders[15], Uarporn Nopmongcol[16], Nutthida Kitwiroon [17], Guido Pirovano[7], Luca Pozzoli[6,1], Marje Prank[18], Ranjeet S. Sokhi[12], Paolo Tuccella[5], Alper Unal[6], Greg Yarwood[16], Stefano Galmarini[1]

European Commission, Joint Research Centre (JRC), Directorate for Energy, Transport and Climate, Air and Climate Unit, Ispra (VA), Italy
Enviroware srl, Concorezzo, MB, Italy
Atmospheric Model Application and Analysis Branch - Computational Exposure Division - NERL, ORD, U.S. EPA
CETEMPS, University of L'Aquila, Italy
Dept. Physical and Chemical Sciences, University of L'Aquila, Italy
Eurasia Institute of Earth Sciences, Istanbul Technical University, Turkey
Ricerca sul Sistema Energetico (RSE SpA), Milano, Italy
University of Murcia, Department of Physics, Physics of the Earth, Campus de Espinardo, Ed. CIOyN, 30100 Murcia, Spain
Institute of Coastal Research, Chemistry Transport Modelling Group, Helmholtz-Zentrum Geesthacht, Germany
Aarhus University, Department of Environmental Science, Frederiksborgvej 399, 4000 Roskilde, Denmark
INERIS, Institut National de l'Environnement Industriel et des Risques, Parc Alata, 60550 Verneuil-en-Halatte, France
Centre for Atmospheric and Instrumentation Research (CAIR), University of Hertfordshire, Hatfiled, UK
Ricardo Energy & Environment, Gemini Building, Fermi Avenue, Harwell, Oxon, OX11 0QR, UK
CIEMAT. Avda. Complutense, 40. 28040. Madrid, Spain
Netherlands Organization for Applied Scientific Research (TNO), Utrecht, The Netherlands
Ramboll Environ, 773 San Marin Drive, Suite 2115, Novato, CA 94998, USA
Environmental Research Group, Kings' College London, London, United Kingdom
Finnish Meteorological Institute, Atmospheric Composition Research Unit, Helsinki, Finland

*Author for correspondence: E.Solazzo, efisio.solazzo@jrc.ec.europa.eu, Phone: +390332789944

**Abstract.** Through the comparison of several regional-scale chemistry transport modelling systems that simulate meteorology and air quality over the European and American continents, this study aims at *i)* apportioning the error to the responsible processes using time-scale analysis, *ii)* helping to detect causes of models error, and *iii)* identifying the processes and scales most urgently requiring dedicated investigations.

The analysis is conducted within the framework of the third phase of the Air Quality Model Evaluation International Initiative (AQMEII) and tackles model performance gauging through measurement-to-model comparison, error decomposition and time series analysis of the models biases for several fields (ozone, CO, $SO_2$, NO, $NO_2$, $PM_{10}$, $PM_{2.5}$, wind speed, and temperature). The operational metrics (magnitude of the error, sign of the bias, associativity) provide an overall sense of model strengths and deficiencies, while apportioning the error to its constituent parts (bias, variance and covariance) can help to assess the nature and quality of the error. Each of the error components is analysed independently and apportioned to specific processes based on the corresponding timescale (long scale, synoptic, diurnal, and intra-day) using the *error apportionment* technique devised in the former phases of AQMEII.

The application of the error apportionment method to the AQMEII Phase 3 simulations provides several key insights. In addition to reaffirming the strong impact of model inputs (emissions and boundary conditions) and poor representation of the stable boundary layer on model bias, results also highlighted the high inter-dependencies among meteorological and chemical variables, as well as among their errors. This indicates that





the evaluation of air quality model performance for individual pollutants needs to be supported by
complementary analysis of meteorological fields and chemical precursors to provide results that are more
insightful from a model development perspective. The error embedded in the emissions is dominant for
primary species (CO, PM, NO) and largely outweighs the error from any other source. The uncertainty in
meteorological fields is most relevant to ozone. Some further aspects emerged whose interpretation requires
additional consideration, such as, among others, the uniformity of the synoptic error being region and model-
independent, observed for several pollutants; the source of unexplained variance for the diurnal component;
and the type of error caused by deposition and at which scale.

## 1. INTRODUCTION

The Air Quality Model Evaluation International Initiative (AQMEII, Rao et al., 2010) has been active since 2008
with the aim of promoting the research on regional air quality model evaluation across the modelling
communities of Europe and North America. It is coordinated by the European Joint Research Centre (JRC) and
the U.S. Environmental Protection Agency (EPA) and it has now reached its third phase, referred to as AQMEII3
hereafter. The experience gathered in the first two phases consisted of important advancement in the model
evaluation research as well as establishing a large community of participating regional modeling groups, and
have made AQMEII a natural candidate to collaborate with the Hemispheric Transport of Air Pollution (HTAP)
initiative. HTAP, a taskforce of the Long Range Transport of Air Pollution program (LTRAP) acting within the
UNECE program, relies on a community of global scale chemical transport models to investigate the fate of air
pollutants emitted in the Northern hemisphere and determine the contribution of remote sources as well as
their impacts to the background concentration in different parts of the globe. HTAP is in its second phase and
the activities undertaken during this second phase include coordinating simulations by both global and
regional scale models. The regions of interest in the Northern hemisphere are North America, Europe and
South East Asia. The regional-scale modelling component of this activity for Europe and North America is being
coordinated by AQMEII while the Asian component is being coordinated by MICs-ASIA (Model Intercomparison
Study-Asia). Global models participating in HTAP are used by the AQMEII regional models as boundary
conditions and special attention has been given to the emission inventory to ensure that it is consistent
between the global and regional-scale simulations as described in Janssens-Maenhout et al. (2015). The
activity described here relates to the evaluation of the base case scenario set up within the context of HTAP
and AQMEII (a description of the HTAP program can be found at www.htap.org).
Following the simulation strategy developed over the first two phases of the AQMEII activity, two continental-
scale domains have been used in the exercise - one over Europe (EU) and one over North America (NA) (Figure
1). The modelling groups participating in AQMEII3 performed air quality (AQ) simulations over one or both of
these domains. Each group has been provided the same inputs for anthropogenic emissions and boundary
conditions and has been left the choice of the optimal configuration of the modelling systems, including
meteorology, grid spacing, and natural emissions. To facilitate the cross-comparison among models, the
modelled outputs have been successively interpolated to a common regular grid of 0.25° spacing over both
continents. The comparison with observational data is performed by interpolating (or by simply taking the
value from the grid cell where the monitoring sites are situated) the model values to prescribed observation
stations (receptors) for surface measurements and at specified vertical heights for comparisons against
measured profiles. As in the previous two phases of AQMEII, the ENSEMBLE system (Galmarini et al., 2012)
hosted by the JRC has been used to accommodate all of the data and to pair modelled to observational values
in time and space to provide direct comparison and statistical analysis.
The model evaluation approach proposed and applied in this study combines aspects of operational and
diagnostic evaluation as defined by Dennis et al. (2010). It makes use of the classical statistical indicators
typically employed for operational evaluation based on the direct comparison with observations, but also
provides more indications on the processes contributing to model errors, which is the focus of diagnostic



model evaluation (Solazzo and Galmarini, 2016). The data used in the analysis are not process specific but are
ordinary time series of modelled and monitoring data which are decomposed into four spectral components:
ID (intra-day), DU (diurnal), SY (synoptic), and LT (long-term), each determined by different physical and
chemical processes (Rao et al., 1997). The error apportionment applied to each spectral component can
provide indications on the possible sources of error. The scope, as also highlighted by Gupta et al. (2009), is to
move beyond the usual aggregate metrics that only offer a statistical interpretation, towards the use of
measures selected for the quality of the information they can provide to model developers and users.
The evaluation of the AQMEII3 suite of model runs is carried out for surface temperature (Temp) and wind
speed (WS), and for the species CO, NO, $NO_2$, ozone, $SO_2$, $PM_{10}$ (EU) and $PM_{2.5}$ (NA). Additional analyses
making use of emission reduction scenarios (CO and NO) and vertical profiles (Temp, WS, ozone) are also
presented.
The main scope of the analysis is to present a detailed overview of the skill of AQ models when compared
against measurements, for several regulatory pollutants and their precursors. For each species, the error is
1.  quantified seasonally for three sub-regions of each continent;
2.  qualified in terms of bias, variance, or covariance type of error, and
3.  apportioned to the atmospheric time-scale, i.e. ID, DU, SY, or LT.
Given the large amount of models and species for two continents and the screening scopes of this work, maps
of model metrics at individual receptors are omitted. Instead, spatial averaging over pre-selected homogenous
sets of measurement points is presented. Investigation of signal associativity through clustering analysis has
been performed for ozone and PM ($PM_{10}$ for EU and $PM_{2.5}$ for NA) over both continents following the
procedure outlined by Solazzo and Galmarini (2015), allowing the detection of three sub-regions (hereafter
referred to as EU1, EU2, EU3 and NA1, NA2 NA3) (Figure 1) where the LT and SY components have shown
robust clustering features. For consistency and to facilitate the interpretation of the results, the same sub-
regions have been adopted for all species.
The error break-down, the time series decomposition, and the models and observational data used are
presented in Section 2. In Section 3, the results of the error apportionment analysis are presented and
discussed. A novel analysis based on the autocorrelation function (acf) of the LT component is presented in
Section 4 for ozone. Conclusions are drawn in Section 5.

## 2. METHODOLOGY

The first step of the analysis is the spectral decomposition of the time series of modelled and observed
species, as outlined in the methodology proposed in Solazzo and Galmarini (2016). Because each spectral
component represents a range of processes in a specific spectral range, the deviation of the modelled from the
observed spectral component is informative about the process(es) causing the error. The second step is to
separate the mean square error (MSE) of each spectral component into its constituent parts: the bias, variance
and covariance. These time-scale specific errors, expressed in terms of bias, variance, and covariance then
allow a more precise diagnosis of their cause.

### 2.1 ERROR BREAK DOWN

The MSE is the squared difference of the modelled and observed values:

$$MSE = E(mod - obs)^2 = \frac{\sum_{i=1}^{nt}(mod_i - obs_i)^2}{n_t}$$     **EQ 1**


where E(·) denotes expectation and $n_t$ is the length of the time series. The bias is:





$$bias = E(mod - obs)$$ **EQ 2**

i.e., $bias = \overline{mod} - \overline{obs}$ (the overbar indicates temporal averaging). The following relationship holds:

$$MSE = var(mod - obs) + bias^2$$ **EQ 3**


(var(·) is the variance operator). By applying known the known property of the variance for correlated fields:

$$var(mod - obs) = var(mod) + var(obs) - 2cov(mod, obs)$$ **EQ 4**


the MSE can be expressed as:

$$MSE = bias^2 + var(mod) + var(obs) - 2cov(mod, obs),$$ **EQ 5**


where the covariance term (last term on the right hand side of Eq 5) accounts for the degree of correlation
between the modelled and observed time series. Following Solazzo and Galmarini (2016), the MSE Eq 5 is
rewritten as:

$$MSE = \left(\overline{mod} - \overline{obs}\right)^2 + (\sigma_{mod} - r\sigma_{obs})^2 + mMSE$$ **EQ 6**

where

$$mMSE = \sigma_{obs}^2(1 - r^2)$$ **EQ 7**

is the minimum error achievable by an accurate (unbiased, $\overline{mod} = \overline{obs}$) and precise ($\sigma_{mod} = \sigma_{obs}$) modelling
system ($r$ is the linear correlation coefficient). *mMSE* is the unexplained portion of the error and reflects the
amount of observed variance not explained by the models (Solazzo and Galmarini, 2016). The *mMSE* type of
error is caused by the variability of the observation not reproduced by the models, which includes
incommensurability, noise, and timing of the signal summarised by the coefficient of determination (Solazzo
and Galmarini, 2016), as well as by the error induced by the meteorological drivers (for primary and secondary
species) and by the short and long range transport of precursors (for secondary species such as ozone)).
The decomposition in Eq 6 includes all the operational metrics commonly adopted to evaluate the AQ models
(bias, variance, correlation coefficient, and their sum, the MSE), and is thus suitable to be used as compact
estimator of model performance.
2.2. SPECTRAL DECOMPOSITION AND ERROR ATTRIBUTION
Spectral filtering has been applied to the measured and modelled hourly-averaged time series at the
monitoring sites using the Kolmogorov-Zurbenko (kz) low-pass filter (Zurbenko, 1986). This allows to separate
different phenomena having distinct signals, such as long-term and short-term fluctuations in the observed
and modelled time series (Rao et al., 1997). Applications of the *kz* filter to ozone have been described in a
number of previous studies (Rao et al., 1997; Wise and Comrie, 2005; Hogrefe et al., 2000; 2014; Galmarini et
al., 2013; Kang et al., 2013; Solazzo and Galmarini, 2015 and 2016).
The kz filter depends on the length of the moving average window $m$ and the number of iterations $k$ ($kz_{m,k}$) ($k$
also indicates the level of noise suppression). Since the kz is a low-pass filter, the filtered time series consists of
the low-frequency component, while the difference between two filtered time series (with different $k$ and $m$)
provides a band-pass filter. This latter property has been used in this study, as well as in a number of previous
studies, to decompose the modelled and observed time series as:

$$FT(S) = LT(S) + SY(S) + DU(S) + ID(S)$$ **EQ 8**






where *S* is the time series of the species being analysed and FT is the full (un-decomposed) time series.
The base line component LT is the long term component (periods longer than 21 days) and accounts for the
temporal fluctuations determined by low frequencies, such as boundary conditions and seasonal variation in
emissions and photo-chemistry. SY is the synoptic component containing fluctuations related to weather-
processes and precursor emissions occurring on scales between 2.5 and 21 days. The DU (diurnal) component
accounts for fluctuations due to diurnal periodicity occurring on temporal scales between 0.5 and 2.5 days,
and ID is the intra-day component, accounting for fast-acting, local-level processes (time scale less than 12
hours) (the spectral components have the same units as the un-decomposed time series).
The decomposition Eq 8 is such that the un-decomposed time series is perfectly returned by the summation
(or by the exponential product, see Appendix 1 for details) of the components. The band-pass nature of the SY,
DU, and ID components is such that they only describe the processes in the time window the filter allows the
signal to 'pass'. For instance, the DU component is insensitive to processes outside the range between 0.5 and
2.5 days.
Because the kz filter was originally developed to deal with ozone, the parameters *k* and *m* (Appendix 1) are
specifically tailored for ozone, taking into consideration its chemistry and life-time. In this study we have
applied the kz filter to other species and kept the same values for *k* and *m* for consistency and to facilitate the
comparison of the results. Although some species (e.g. PM, CO, $SO_2$) may be less sensitive to day/night cycles
than ozone, the distinction between DU and ID are still revealing of emission patterns like vehicular traffic and
industrial activities as well as diurnal variations in vertical mixing. Moreover, the SY and LT are associated with
transport and other weather processes common to all species.
Two aspects of the signal filtering having a profound impact on model evaluation are:
1. The non-orthogonality of the spectral components is one of the major drawbacks of the signal
decomposition. A clear-cut separation of the components of Eq 8 is not achievable, since the separation is a
non-linear function of the parameters *m* and *k* (Rao et al., 1997; Kang et al., 2013) and the leakage among
components mixes together in each component different physical processes. Galmarini et al. (2013) found that
the explained variance by the spectral components accounts for 75 to 80% of the total variance while the
remaining portion of the variance is due to the interactions between the estimated components. The effect of
these interactions on the error apportionment pursued in this study is outlined and quantified in section 3.
Other spectral techniques could be used but either they not guarantee the absence of signal leakage (e.g.
anomaly perturbation method) or require special treatment of missing data (e.g. wavelet transform method)
(Rao et al., 1997; Eskridge et al., 1997).
2. The bias is calculated as the distance between the time average modelled and observed time series. In such
a 'time average' sense, the base line LT is the only biased component, containing the entire bias of the original
time series. The other components are zero-mean fluctuations about LT and are unbiased. Although inaccuracy
at each time step can also derive from the SY, DU and ID components (Johnson, 2008), in this study the signal
is taken as time-averaged over a finite period, and therefore the entire bias is apportioned to the base-line (LT)
component.
**2.3 MODELS AND OBSERVATIONAL DATA**
Table 1 summarises the modelling systems participating in AQMEII3. Twelve modelling groups produced
outputs over EU and four over NA (although not all fields were made available by all groups). Sensitivity
simulations performed by two groups, in which alternate emission inventories were used, raises the number of
EU contributions to fourteen.



The 'standard' emission inventories are those developed for the second phase of AQMEII for EU and NA and extensively described in Pouliot et al. (2015). For EU, the 2009 inventory of anthropogenic emissions was used, although biogenic emissions (meteorology-dependent) were specifically calculated for the year of 2010 by several groups. In regions not covered by the standard inventory, such as North Africa, five modelling systems (Table 1) have complemented the standard inventory with the HTAPv2.2 (Janssens-Maenhout et al., 2015) datasets. The two inventories are the same over EU and in the MACC inventory the non-European emissions are not included. Other small differences might exist among the two inventories (like in the shipping emissions), but we consider them to be of small impact for the spatial averaged analysis carried out in this study. Emissions from lightning and volcanic sources are not contained in the EU and NA emissions inventories, since not all participating models include robust methods for estimating these emissions.

Two EU modelling systems (CHIMERE, SILAM) made results available with both inventories. For both continents the regional scale emission inventories where embedded in the global scale inventory (Janssens-Maenhout et al., 2015) to guarantee coherence and harmonization of the information used by the two communities. The ability of some modelling groups to perform sensitivity simulations with both the TNO MAC and the HTAP v2.2 information allowed also to determine the impact of North African emissions on the European domain. For Chimere, the MACC inventory over France and the UK was spatially redistributed considering national inventories (having higher spatial resolution), while for the other countries it was redistributed by considering point source locations, land-use and population. For processing the HTAP inventory, population was not used as a parameter for spatially distributing the emissions.

For the NA domain, the 2008 National Emission Inventory was used as the basis for the 2010 emissions, providing the inputs and datasets for processing with the SMOKE emissions processing system (Mason et al., 2012). Year specific updates for the year of 2010 were made for several sectors, including mobile sources, power plants, wildfires, and biogenic emissions. Additional details can be found in Im at al. (2015a,b) and Pouliot et al. (2015).

Chemical boundary conditions were provided by the Composition – Integrated Forecast System (C-IFS) model (Flemming et al., 2015), including ozone, $NO_x$, CO, $CH_4$, $SO_2$, NMVOCs, dust, organic matter, black carbon and sulphate. Sea salt at the boundaries, although provided, was not used due to unrealistically high values.

*[Table 1 here]*

### 2.3.1 MODEL FEATURES

This section presents the main features of the modelling systems participating to AQMEII3. Complementary information is provided in Table 1.

The FMI (Finnish Meteorological Institute) has taken part with the ECMWF-SILAM system (ECMWF-SILAM_M and ECMWF-SILAM_H of Table 1, indicating the instances of the SILAM model using the MACC and the HTAP emission inventory, respectively) (ECMWF: European Centre for Medium-Range Weather Forecasts). SILAM v5.4 (Sofiev et al., 2015) has been used, with meteorological input extracted from the ECMWF operational archives. The thickness of the first layer is 30m. The simulation included sea-salt emissions as in Sofiev et al. (2011) (but not from the boundaries), biogenic VOC (volatile organic compounds) emissions as in Poupkou et al. (2010) and wild-land fire emissions as in Soares et al. (2015). The wind-blown dust is only included from the lateral boundary conditions. Anthropogenic $NO_x$ emissions have been treated as 10% $NO_2$ and 90% NO. The volatility distribution of anthropogenic OC was taken from Shrivastava et al. (2011). The gas phase chemistry was simulated with CBM-IV, with reaction rates updated according to the recommendations of IUPAC (http://iupac.pole-ether.fr) and JPL (http://jpldataeval.jpl.nasa.gov). The secondary inorganic aerosol formation was computed with updated DMAT scheme (Sofiev, 2000) and secondary organic aerosol formation



with the Volatility Basis Set (VBS, Ahmadov et al., 2012). A known deficiency of the SILAM version used in this
study is the overestimation of ozone dry deposition.
The LOTOS-EUROS modelling system (Schaap et al. 2008, Sauter et al. 2012) has been applied by TNO (the
Netherlands Organization for Applied Scientific Research), using version v1.10.1. The meteorological inputs
have been extracted from the ECMWF operational archives. For biogenic emissions the approach as described
in Beltman et al. (2013) has been used. Gas-phase chemistry is based on CBM-IV (modified reaction rates, see
Sauter et al., 2012), secondary inorganic aerosol (SIA) formation on Isorropia II (Fountoukis and Nenes, 2009)
and for semivolatile species the VBS approach was used (Donahue et al. 2006, Bergström et al. 2012), with
100% of the emitted OC mass in the 4 lowest volatility classes that are predominantly solid and an additional
150% in the five higher volatility bins. Modelled terpene emissions were reduced by 50% to limit their
contribution to SOA (secondary organic aerosol) formation which was found to be too high otherwise. This is
justified since contributions of terpene to SOA formation is known to be very uncertain and at the same time
the model is very sensitive to terpene emissions (Bergström et al., 2012). 3% of the total anthropogenic $NO_x$
emissions were attributed to $NO_2$ while 97% were attributed to NO. No $NO_x$ emissions from soil were taken
into account. The model includes pH dependent conversion rates for $SO_2$ (Banzhaf et al., 2012), while only
below-cloud scavenging is used for wet deposition. Mineral dust emissions were calculated on-line, including
emissions from road resuspension and agricultural activities, according to Schaap et al. (2009). For sea spray
the parameterizations by Monahan et al. (1986) and Martensson et al. (2003) were used. A specific feature of
LOTOS-EUROS is that it only covers the lower 3.5 km of the atmosphere, with a static 25 m surface layer, a
dynamic mixing layer and two dynamic reservoir layers. This makes the model relatively fast in terms of
computation time but has implications for the vertical mixing of species for instances where the mixing layer
rapidly changes in height.
WRF-WRF/Chem1 is applied by the University of L'Aquila (Italy). The version 3.6 of the Weather Research and
Forecasting model with Chemistry model (WRF/Chem) (Grell et al., 2005) has been used for AQMEII3. This
version of the model has been modified to include the new chemistry option implemented by Tuccella et al.
(2015) that includes in the simulation of direct and indirect aerosol effects a better representation of the
secondary organic aerosol mass, calculated as in Ahmadov et al. (2012). Here only direct effects have been
included in the simulation, for computational expediency. The model uses RACM-ESRL gas phase chemical
mechanism (Kim et al., 2009), an updated version of the Regional Atmospheric Chemistry Mechanism (RACM)
(Stockwell et al., 1997). The inorganic aerosols are treated with the Modal Aerosol Dynamics Model for Europe
(MADE) (Ackermann et al., 1998). The parameterization for SOA production is based on the VBS approach. The
aerosol direct and semi direct effects are taken in account following Fast et al. (2006). Cloud chemistry in the
convective updraft is modelled using the scheme of Walcek and Taylor (1986), while the aqueous phase
oxidation of $SO_2$ by $H_2O_2$ in the grid-resolved clouds is parameterized with the scheme used in GOCART
(Goddard Chemistry Aerosol Radiation and Transport). Wet deposition from convective and resolved
precipitation is included following Grell and Freitas (2014). The photolysis frequencies are calculated with the
Fast-J scheme (Fast et al., 2006), the dry deposition velocities are simulated with the parameterization
developed by Wesely (1989). Dry deposition and photolysis schemes were modified to take in account the
effects of the soil snow coverage following Ahmadov et al. (2015). The anthropogenic emissions are taken
from TNO-MACC inventory for 2009 (Kuenen et al., 2014) and have been adapted to the chemical mechanism
used following the method of Tuccella et al. (2012). The biogenic emissions have been calculated online by
using the Model of Emissions of Gases and Aerosols from Nature (MEGAN) (Guenther et al., 2006).
Anthropogenic $NO_x$ sources were assumed 95% of NO and 5% of $NO_2$. The main physical parameterization used
include the Rapid Radiative Transfer Method for Global (RRTMG) for solar and infrared radiation (Iacono et
al. 2008), Morrison microphysics (Morrison et al., 2010), the Mellor-Yamada Nakanishi-Niino (MYNN) planetary
boundary layer (PBL) scheme (Nakanishi-Niino, 2006), the NOAH land-surface model (Chen and Dudhia, 2001)
and the Grell-Freitas scheme for cumulus clouds (Grell and Freitas, 2014). The meteorological analysis used to
initialize WRF are provided by the ECMWF with a horizontal resolution of 0.5° every 6 hours. Chemical



boundary conditions are taken from C-IFS. A series of 72-hour simulations has been performed on each day
starting at 00 UTC. Each run is preceded by a pre-forecast of 12 hours (from 12 to 00 UTC) only with
meteorology, in which the model is nudged toward analysis above the PBL in order to prevent a drift from
synoptic circulation patterns. The last hour of this spin-up is then used as meteorological initial condition for
WRF/Chem. The chemical state is restarted from the previous 72-hours run.
WRF-WRF/Chem2 applied by the University of Murcia (Spain) relies on the WRF-Chem model (Grell et al.,
2005). The following physics options have been applied for the simulations: RRTMG long-wave and short-wave
radiation scheme; Lin microphysics (Lin et al., 1993), the Yonsei University (YSU) PBL scheme (Hong et al.,
2006), the NOAH land-surface model and the updated version of the Grell-Devenyi scheme (Grell and
Devenyi, 2002) with radiative feedback. Chemical options include: RADM2 chemical mechanism (Stockwell et
al., 1990); MADE/SORGAM aerosol module (Schell et al., 2001) including some aqueous reactions; Fast-J
photolysis scheme. The modelling domain covers Europe and a portion of Northern Africa.
Simulations of WRF-CAMx over EU have been performed by RSE (Italy) using CAMx version 6.10 (Environ,
2014) with Carbon Bond 2005 (CB05) gas phase chemistry (Yarwood et al., 2005) and the Coarse-Fine (CF)
aerosol module. Input meteorological data were generated by WRF-Chem model version 3.4.1 (Skamarock et
al., 2008a,b), driven by ECMWF analysis fields. Grid nudging of wind speed, temperature and water vapour
mixing ratio has been employed within the PBL, with a nudging coefficient of 0.0003 sec-1. WRF-Chem has
been adopted to predict GOCART dust emissions (Ginoux et al., 2001) along with the meteorology. The
WRFCAMx pre-processor (version 4.2; ENVIRON, 2014) was used to create CAMx ready input files collapsing
the 33 vertical layers used by WRF to 14 layers in CAMx but keeping identical the layers up to 230 m above
ground level. Anthropogenic emissions were derived by the TNO-MACC data applying a $NO_2/NO_x$ ratio of 5%
for each emission category. Biogenic VOC emissions were computed by applying the MEGAN emission model
v2.04. Sea salt emissions were computed using published algorithms (de Leeuw et al., 2000; Gong, 2003).
Aarhus University (Denmark) applied the WRF-DEHM modelling system over EU and NA. The DEHM model
used anthropogenic emissions from the EDGAR-HTAP database and biogenic emissions are calculated using the
MEGAN model. The gas-phase chemistry module includes 58 chemical species, 9 primary particles and 122
chemical reactions (Brandt et al., 2012). Secondary organic aerosols (SOA) are calculated following the two-
product approach assuming that hydrocarbons undergo oxidation through $O_3$, OH and $NO_3$ and for only  two
semi-volatile gas products (Zare et al., 2014). However, the module is simple as it does not include aging
processes and further reactions in the gas and particulate phase (Zare et al., 2014). Other modelling options
include the Noah Land Surface Model (Chen and Dudhia, 2001), Eta similarity surface layer (Janjic, 2002), the
Mellor-Yamada-Janjic (Eta operational) boundary layer scheme (Mellor and Yamada, 1982), the Kain-Fritsch
(Kain, 2004) scheme for cumulus parameterisation, the WRF Single-Moment 5-class Microphysics scheme
(Hong et al., 2004), and the CAM scheme for both long and short radiation (Collins et al., 2004).
WRF-CMAQ1 has been applied by the ITU (Istanbul Technical University) over EU. The WRFv3.5 model has
been used with the following physical options: WSM3 microphysics scheme (Hong et al., 2004), RRTM (long-
wave radiation scheme, Dudhia shortwave radiation scheme (Dudhia, 1989), NOAH land surface model, Yonsei
University PBL scheme and Kain–Fritsch cumulus parameterization scheme (KF2, Kain, 2004). The NCEP
(National Centers for Environmental Prediction) FNL Operational Model Global Tropospheric Analyses has
been used for boundary conditions and nudging the meteorological simulation. The MCIP version 3.6 (Otte and
Pleim, 2010) has been used to process WRF output for CMAQ. The MEGANv2.1 (Guenther et al., 2012) model
has been used to calculate the biogenic VOC emissions from vegetation, using surface temperature and
radiation from MCIP output. CMAQv4.7.1 (Foley et al., 2010) was configured with the CB05 chemical
mechanism and the AERO5 module (Foley et al., 2010) for the simulation of gas-phase chemistry and aerosol
and aqueous chemistry, respectively. 95% of $NO_x$ anthropogenic emissions were considered as NO.





The WRF-CMAQ2 system has been applied by Ricardo Energy & Environment (Ricardo-E&E) over EU. It has
been configured using WRFv3.5.1 and CMAQ v5.0.2. The WRF model adopted the KF2 cumulus cloud
parameterization and Morrison microphysics scheme (Morrison et al., 2009), the ACM2 (Asymmetric
Convective Model version 2, Pleim, 2007) for the PBL, the Pleim-Xiu land-surface model (Xiu and Pleim, 2001),
and the RRTMG radiative module. The NCEP FNL Operational Model Global Tropospheric Analyses has been
used to generate boundary conditions for the European meteorological simulation. Nudging of temperature,
wind speed, and water vapour mixing ratio has been applied above the PBL (Gilliam et al., 2012). The CMAQ
model adopted the CB05-TUCL chemical mechanism (Whitten et al., 2010; Sarwar et al., 2011a), the AERO6
three mode aerosol module (Appel et al., 2013). The MCIP version 4.2 has been used to process WRF output
for CMAQ. The MEGANv2.0.4 model has been used to calculate the biogenic VOC emissions from vegetation,
using surface temperature and radiation from MCIP output. For road transport, 86% of NOx anthropogenic
emissions were considered as NO and 95% of NOx anthropogenic emissions were considered as NO for all
other emissions.
The WRF-CMAQ3 modelling system has been applied by the University of Hertfordshire and utilized the
uncoupled version of the WRF-v3.4.1 model and CMAQ v5.0.2. The WRF simulations were performed using
18km x 18km horizontal grid resolution with 36 vertical sigma layers. The simulations used Unified Noah Land
Surface Model as the land surface scheme, Pleim-Xiu Scheme for the surface layer, RRTMG as the long-wave
and shortwave radiation scheme, Morrison 2-moment scheme for microphysics parameterization, KF2 scheme
for cumulus parameterization, and ACM2 scheme for PBL parameterization. Meteorological initial and lateral
boundary conditions were derived from the ECMWF analysis. In order to constrain the meteorological model
towards the analyses a grid nudging technique was employed every 6 hours of WRF simulation. The results
from WRF simulations were pre-processed for CMAQ using Meteorology-Chemistry Interface Process (MCIP)
version 3.6 (Otte et al., 2005). In CMAQ model, the gas phase chemical mechanism was based on carbon bond
chemical mechanism version 5 (Foley et al., 2010) with updated toluene and chlorine chemistry (CB05-TUCL)
and the aerosol chemical reaction were treated with AERO6 module. The CMAQ model consisted of 35 vertical
layers and extending up to ~16 km height with the thickness of lowest layer is approximately 20 m. The EDGAR
HTAP V2 emissions ($0.1^{o}$ x $0.1^{o}$) as well as TNO emissions data (~7 km x 7 km) were used as anthropogenic area
and point sources emission data respectively in CMAQ. The biogenic emissions were derived from MEGAN.
The WRF-CMAQ4 simulation has been performed by the Kings College (UK) using CMAQ v5.0.2 (Byun and
Schere, 2006) with CB05 chemical mechanism that includes aqueous and aerosol chemistry. The CMAQ model
is driven by meteorological fields from the WRF v3.4.1. The lateral boundary conditions for WRF are taken
from the Global Forecast System (GFS) model with 6-hr interval and 1° grid resolution. The WRF physic
schemes include RRTM radiation module KF2 cumulus parameterization, WSM6 microphysics (Hong and Lim,
2006), Pleim-Xiu surface layer scheme (Pleim and Xin, 2003), RUC land surface model (Benjamin et al., 2004),
and ACM2 PBL scheme. The anthropogenic emissions for most part of the model domain are from MACC and
the missing information have been filled with the emissions provided by EDGAR/HTAP. The biogenic emissions
were estimated using the Biogenic Emission Inventory System version 3 (BEIS3) model in SMOKE v2.6
(https://www.cmascenter.org/smoke). The dust (Tong, et al 2011) and sea-salt (Gantt et al., 2015) emissions
are generated using CMAQ inline modules. The ratio for $NO_2/NO_x$ emissions is ~10% (Bieser et al., 2011a).
The INERIS and CIEMAT institutes jointly applied the ECMWF-Chimere system. CHIMERE (version CHIMERE
2013) has been run for a 0.25x0.25 horizontal resolution and 9 vertical levels, extending up to 500 hPa with a
first (lower)-layer depth of 20 m, using the meteorology provided by ECMWF IFS (Integrated Forecast System).
Biogenic VOC emissions from vegetation and soil NO emissions have been calculated with the MEGAN model
(version 2.04; Guenther et al., 2006, 2012). Sea salt emissions inside the domain have been calculated
according to Monahan (1986). No sea salt condition was considered at the boundaries. The wind-blown dust is
only included from the lateral boundary conditions. CHIMERE uses the MELCHIOR2 chemical mechanism
(Lattuati, 1997) and ammonium nitrate equilibrium was calculated with ISORROPIA (Nenes et al., 1999). Dry




deposition is based on the resistance approach (Emberson 2000a,b) and both in-cloud and sub-cloud
scavenging have been considered for wet deposition.
HZG has used the COSMO-CLM meteorological model to drive the CMAQ model. For AQMEII3 the CMAQ
version 5.0.1 was used, with the CB05-TUCL scheme and the multi-pollutant aerosol module AERO6. CMAQ is
run on a 24x24km² horizontal grid, using 30 vertical layers up to 50hPa (lowest layer of approximately 40m).
CMAQ was run using the optional in-line calculation of dry deposition velocities. Wet deposition processes
include in-cloud and sub-cloud scavenging processes. All atmospheric parameters were taken from regional
atmospheric simulations with the COSMO-CLM (CCLM) mesoscale meteorological model (version 4.8) for the
year 2010 (Geyer, 2014) using NCEP forcing data employing a spectral nudging method for large-scale effects
(Kalnay et al., 1996). CCLM is the climate version of the regional scale meteorological community model
COSMO (Rockel et al., 2008; Steppeler et al., 2003; Schaettler et al. 2008). CCLM uses the TERRA-ML land
surface model (Schrodin and Heise, 2001), a TKE closure scheme for the PBL (Doms et al., 2011), cloud
microphysics after Seifert and Beheng (2001), the Tiedtke scheme (Tiedtke, 1989) for cumulus clouds and a
long wave radiation scheme following Ritter and Geleyn (1992). The meteorological fields were afterwards
processed to match the 24x24km² CMAQ grid using the LM-MCIP pre-processor. The emission input for CCLM-
CMAQ is based on the EDGAR HTAPv2 database, interpolated to the CMAQ model grid and aggregated
following the SNAP emission sector nomenclature. Sector specific hourly temporal profiles and speciation
factors of PM and VOC species were applied by the SMOKE for Europe emissions model (Bieser et al., 2011a).
The temporal profiles used were fixed monthly, weekly, and diurnal profiles. NOx emissions were split using a
NO/$NO_2$ ratio of 0.9/0.1 for mobile sources and a fixed ratio of 0.9/0.1 for all other source sectors. Biogenic
emissions and NO emissions from soil were calculated using BEISv3.14. Sea-salt emissions are calculated in-line
by CMAQ including sulphate emissions based on an average sulphate content of 7.7%. Finally, fixed vertical
profiles were applied for each source sector (Bieser et al., 2011b).
The WRF-CMAQ system applied over NA by the US EPA (Environmental Protection Agency) has been
configured using WRFv3.4 and CMAQv5.0.2 (Appel et al., 2013; see also Foley et al., 2010 and Byun and
Schere, 2006). The options used in these WRF and CMAQ simulations are identical to those described in
Hogrefe et al. (2015) except that the current simulations were performed in offline rather than two-way
coupled mode. Temperature, wind speed, and water vapor mixing ratio were nudged above the PBL following
the approach described in Gilliam et al. (2012). Soil temperature and moisture were nudged following Pleim
and Xiu (2003) and Pleim and Gilliam (2009). The $NO_2$/$NO_x$ split applied during SMOKE emissions processing
varies for different categories. For many categories is the assumed split 90% NO / 10% $NO_2$, but for mobile
sources the split varies for different types of vehicles and different emission processes.
Ramboll Environ used CAMx (version 6.2, Ramboll Environ, 2015) for simulations over NA, with CB05 chemical
mechanism for gas-phase. The modeling domain covers the CONUS US with 459 by 299 grid cells of 12 by 12
km size and 26 vertical layers. Height of first layer is 20 m. Biogenic emissions were obtained from the MEGAN
model version 2.1 (Guenther et al., 2006). Meteorological fields were produced by the US EPA (Environmental
Protection Agency) using WRF model and reformatted using the WRFCAMx pre-processor to be readily used by
the CAMx model.
2.3.2 OBSERVATIONAL DATA USED
The observational data used in this study is the same as the dataset used in second phase of AQMEII (Im et al.,
2015a,b) and was derived from the surface air quality monitoring networks operating in EU and NA. In EU,
surface data were provided by the European Monitoring and Evaluation Programme (EMEP;
http://www.emep.int/) and the European Air Quality Database (AirBase; http://acm.eionet.europa.eu/
databases/airbase/). In NA observational data were obtained from the NAtChem (Canadian National
Atmospheric Chemistry) Database and from the Analysis Facility operated by Environment Canada
(http://www.ec.gc.ca/natchem/). For the purposes of comparing the models against observations, only



stations with data completeness greater than 75% for the whole year and elevation above ground below 1000
m have been included in the analysis. Stations with continuous missing records for periods longer than 15 days
have been removed from the dataset.
In addition, we also make use of vertical profiles of ozone, temperature and wind speed measured by
ozonesondes. Ozonesonde data have been extracted from the World Meteorological Organization (WMO)
World Ozone, and Ultraviolet Radiation Data Centre (Toronto, Canada) and made available to the AQMEII
community. These measurements report vertical profiles of ozone at several vertical levels. Further details on
these data are given in Solazzo et al. (2013).
Time-averaged statistics have been calculated after the spatial aggregation of the modelled and observed time
series and prior to the spectral decomposition (the original time series have been spatially averaged first and
then this spatial average time series has been spectrally decomposed). As a consequence of the spatial
averaging, the relative importance of the ID component is likely reduced, since the ID fluctuations are highly
variable in space (Hogrefe et al., 2014). Further, no land-use type filtering has been applied to the stations
used for evaluation. While this choice has limited impact on the SY and LT components (Solazzo and Galmarini,
2015; Galmarini et al., 2013), the DU components of some species (such as PM, $NO_x$) might be strongly
influenced by the vicinity of urban stations to emissions sources.
Details of the modelled regions and number of receptors are reported in Table 2.
*[Table 2 here]*

## 3. RESULTS

The analyses presented in this section focus on evaluating the performance of the models. The accuracy of the
spectral components is first analysed in terms of the root MSE and quantified on a seasonal basis. The season
most affected by error is then further investigated by applying the error apportionment (Eq 6) to the spectral
components. Results are presented for one sub-region only (EU2 and NA1 or NA2) in the main portion of the
manuscript while results for the other sub-regions are included in the supplementary material.
The combination of the spectral decomposition and error apportionment approaches has the effect of
neglecting the error associated with the cross components (twelve spectral interaction terms, see Solazzo and
Galmarini (2016) for details) since the apportionment only deals with the error of the 'diagonal' components
LT, DU, SY, ID. The reason is that while the contribution of the cross components to the overall error can be
quantified, the associated time series needed to carry out the apportionment analysis cannot. The neglected
part of the error is quantified in Table S1. In some instances, such portion can be as high as 20% of the total
error for ozone.
Tables summarising the operational statistics (MB: Mean Bias; *r*: Pearson Correlation coefficient; RMSE: Root
Mean Square Error) are reported in the Supplementary material and have been calculated using the 'openair'
package (Carslaw and Ropkins, 2012).

### 3.1 METEOROLOGICAL DRIVERS: TEMPERATURE AND WIND SPEED

#### 3.1.1 NEAR-SURFACE MODEL EVALUATION

The RMSE for surface temperature and wind speed is reported in Figure 2 (EU) and Figure 3 (NA). For EU
(Figure 2a), the RMSE of the full (i.e. not spectrally decomposed and denoted as "FT" in the plots) time series
of temperature for the entire year is, on a seasonal average, on the order of ~0.5-2K (but often exceeding 3K
in EU3), with higher values typically occurring in spring and winter. The CHIMERE and SILAM models (both
directly driven by the global meteorological fields provided by ECMWF) report the smallest error in EU1 and
EU2, while the WRF/Chem2 model has the largest error in all sub-regions (up to ~5K for EU3 in summer) which





is largely caused by the unusually large error in the SY component when compared to other models. The RMSE
of the LT component resembles the behaviour of the full time series, with the highest error in spring and
winter (on average). The RMSE of the SY component is below ~2K (slightly higher in EU3) except for
WRF/Chem2, whereas the DU component shows a more marked regional dependence, with the EU3 sub-
region reporting, on average, approximately 50% higher seasonal error than the other two sub-regions, more
pronounced in summer.
The bias is predominantly negative (model underestimation) for all EU models and sub-regions, except for
WRF-CMAQ4 in EU3, where the model overestimates the measured temperature in summer and winter.
Finally, the correlation coefficient is higher than 0.90 for the majority of models and spectral components
(Table S2).
For NA (Figure 3a) the temperature RMSE of the WRF-DEHM and CCLM-CMAQ models (peaking in winter and
autumn) is ~ 1-1.5K larger than the WRF-CMAQ model. The error of the SY component is of ~0.5K, while that
of the DU component is significantly higher (between 0. 5K and 2K). The WRF-CMAQ model has a small bias (LT
error small) so that the overall error is dominated by the error in the DU component. The bias is negative for
the WRF-DEHM model in all sub-regions and has the same sign for CCLM-CMAQ and WRF-CMAQ, i.e. negative
in spring and positive in the other seasons (although for NA2 and NA3 WRF-CMAQ reports a slightly negative
bias also in winter) (Table S2).
The RMSE of the surface WS for EU shows large model-to-model variability, more markedly for the LT and SY
components (all sub-regions, Figure 2b), whereas the error of the DU component is more evenly distributed
across models (and significantly higher in EU3, where low-wind speed conditions are predominant). Although
the meteorological fields are assimilated within the models (either from NCEP or from ECMWF, see Table 2),
there are profound differences in the way these fields are ingested and interpolated to the model grid, as well
as differences in the parameterisation of the boundary and surface layer which impact the modelled wind
speed and temperature. For example, the two instances of WRF/Chem applied the assimilation of the
meteorological fields (wind speed, temperature, and relative humidity) of global meteorological fields only
above the PBL, whereas other models (e.g. WRF-CAMx) assimilated the global data also within the PBL. For the
models directly driven by the global fields, (e.g SILAM, Chimere) the seasonal error for WS (~0.5-1 ms$^{-1}$) and
temperature (0.4-1.2K) (Figure 2a,b) can be considered as the uppermost limit the accuracy of the models can
achieve. Thus, the assimilation and interpolation methods errors (which are specific to the configuration of the
meteorological model) can add up more than 1.5K and 2ms$^{-1}$ to the total error.
The full WS time series of the WRF-DEHM, WRF/Chem1 and WRF/Chem2 models report the largest error (in
excess of 1.5m/s), and the WRF-CAMx model even up to 2.4 m/s in winter (all sub-regions, Figure 2b). On
average, the remaining models have an error of 0.5-0.7m/s. Most of the error is apportioned to the LT
component, with the SY and DU below 0.3 m/s (except for WRF-CAMx and the other models mentioned
above).
The WS bias is positive for all models (model over-prediction), for all seasons and sub-regions (only exception
is the CCLM-CMAQ model, biased low during spring and summer in EU3 and WRF-CMAQ2 during summer in
EU1). The correlation coefficient is above 0.9 for the majority of models and components (except for the
models affected by large errors such as the WRF-CAMx model). In general, $r$ is slightly lower in EU3, and is at
maximum for the SY component (Table S3).
For NA (Figure 3b), the WRF-DEHM model reports an error of ~1-1.2 m/s during all seasons and sub-regions,
while the error of the WRF-CMAQ model ranges between 0.45 and 0.75 m/s for all seasons and sub-regions.
The error of the SY and DU components is small (below 0.3m/s for each season) for both models. Both models
are biased high (all instances) and the correlation coefficient is in the order of ~0.9 or above (Table S3).





### 3.1.2 VERTICAL PROFILES

Vertical profiles of mean bias for Temp and WS are reported in Figure 4 to Figure 7. The modelled profiles have been evaluated using ozonesondes measurements. The frequency and local time of the launches are summarised in Table 3. The launches in EU predominately occurred during daylight hours, whereas for NA measurements are available also for night-time and late afternoon. The sign and magnitude of the bias are informative about error in the PBL processes, which will help the discussion on the error of the modelled pollutants (section 3.3).

The bias for temperature in EU ranges between -3K (CCLM-CMAQ at station 308, Figure 5) and +2K (WRF-CMAQ4 at station 308 and SILAM at station 156) at the surface. In most cases the temperature bias profiles fluctuate around zero (station 053, located between EU1 and EU2; station 043; station 242 in EU2, and partially station 316 in EU2), whereas for some stations the bias keeps the same sign throughout the troposphere, negative for station 156 (launches at 10-12 LT) and positive for station 099 (early morning launches). The difference in altitudes (491 m asl the former and 1000 m asl the latter) and the complex terrain of the alpine region might also be factors for the large model differences at these two (relatively close) stations.

Vertical profiles of Temp in NA (Figure 6) shows strong surface bias (negative) at station 021 and 457 (both close to the western border of the domain), for both models. At station 021 (data collected under daylight conditions) the bias becomes positive and small in magnitude above the PBL, whereas at station 457 (data collected under night-time conditions) the bias keeps the same sign throughout the troposphere. At the other stations, the bias within the PBL is overall small and either positive (107, 456) or slightly negative (stations 458, 338).

Bias profiles for WS at eight ozonesondes stations in EU (Figure 4) show a tendency of overestimation in the PBL and of underestimation above ~1000m, although there are some exceptions for different models and/or launching stations. The WRF/Chem1 has the largest positive bias at all sites, with the bias staying positive well above the PBL at all stations in contrast with all other models (WRF/Chem1 model adopted the assimilation of meteorological fields only above the PBL, and only during the first 12 hours of meteorological spin-up). WS overestimation by WRF/Chem is a known concern (e.g. Tuccella et al., 2012b; Jimenez and Dudhia, 2012; Mass and Ovens, 2011) and it is likely to have a major impact on the dispersion of pollutants. As for EU, the WS bias profiles in NA are biased high near the surface (except for the station 338 and, partially, station 021) (Figure 6). Above the PBL the tendency is to underestimate the WS (up to ~1.5m/s), although less dramatically than in EU. As both NA models are driven by WRF for meteorology, the WS profiles are alike and the magnitude of the bias very similar.

### 3.2 DRY DEPOSITION

The simulated annual accumulated dry deposition per unit area over the continental areas for $NO_2$, ozone, and $PM_{2.5}$ is reported in Figure 8 for EU and NA. The graphs report the modelled values only (no observations are readily available). The model-to-model variability in dry deposition is mainly attributable to land cover and model grid size, as the majority of the models employ variations of the resistance scheme (Table 1). As recently noted by Valmartin et al. (2014), developments of the dry deposition schemes can have a profound impact on the overall model bias and on the accuracy of the modelled cycle of the pollutants.

The deposition of $NO_2$ is very similar among all the models for both continents, with the only exception of the WRF-DEHM model in EU and NA, whose median and 75[th] percentile values are below 0.5 and 1.5 kg/km2, respectively. For ozone, the medians of the distribution are in the range ~80-200 kg/km$^2$ for EU (nine models), whereas the 75[th] percentile shows larger variability, ranging between ~150kg/km$^2$ for WRF-CMAQ1 and ~500kg/km$^2$ for the WRF-DEHM and WRF/Chem1 models. The median difference for ozone is more marked in



NA (Figure 8b), between ~200 and ~300 kg/km$^2$ (two models), with a notably relative impact of 50% and 33%
of the median values for WRF-CMAQ and WRF-DEHM respectively.
Finally, deposited PM$_{2.5}$ in EU is modelled with varying magnitudes, from below 5 kg/km$^2$ (SILAM, WRF-
CMAQ1, WRF-CMAQ2) up to 35 kg/km$^2$ (WRF/Chem1) (median values). The median values for the two NA
models are very similar ~25kg/km$^2$, but there is a large discrepancy between the 75$^{th}$ percentile values, with
that of WRF-CMAQ (~170kg/km$^2$) more than four times higher than values predicted by WRF-DEHM.

### 3.3 CHEMICAL SPECIES: MEAN SQUARE ERROR AND ERROR APPORTIONMENT

#### 3.3.1 CO
CO is a moderately long-lived primary pollutant principally produced by incomplete combustion of fossil fuels,
wildfires and, on the global scale, by the oxidation of methane. CO also acts as precursor to ozone. Results of
the AQMEII3 models for CO are reported in Figure 9 and Figure 10, and in Table S5.
In general, there are profound differences between the CO statistics for EU and NA, with the latter showing a
more marked temporal and spatial dependency as well as model-to-model variability (the yearly mean
observed values of CO in EU and NA are of 336 ppb and of 248 ppb, respectively). The EU error (Figure 9a) is,
generally, uniform across models and sub-regions, approximately three times higher in winter than in summer.
The magnitude of the SY and DU errors is comparable (~15-25 ppb on average in EU1 and EU2, sensibly higher
in EU3). Also for NA (Figure 9b) the DU and SY errors are similar, but varying by model, sub-region, and season.
The homogeneity of error in EU suggests that it is originated by a common source. Previous investigations
(Innes et al., 2013; Giordano et al., 2015) indicate that the boundary conditions have a limited contribution to
the bias of CO within the interior of the domain, where the emissions are far more important. In particular, the
MACC inventory used by the EU regional models likely underestimates the CO emissions (especially in winter)
(Giordano et al., 2015). We conclude that most probably the cause of model bias for CO is attributable to the
emissions and, to a lesser extent, the generally overestimated surface wind speed (section 3.1.1). Sensitivity of
the model error to emission changes for CO is discussed in the next section.
The correlation coefficient for EU generally peaks in spring (LT component) while it is at a minimum for the LT
component in winter and overall poor for the DU and SY components. In contrast, for NA the minimum
correlation coefficient is observed in spring/summer (LT component), with the correlation for DU component
having a mixed behaviour depending on the sub-region, but it is typically low in summer (Table S5 of the
supplementary material).
The winter LT error for EU is of ~140-220ppb in EU1 and EU2, and up to 600ppb in EU3, typically higher  than
in NA (~100 ppb, peaking in autumn and mostly due to model underestimation), while the opposite holds for
the DU and ID error which are significantly lower in EU (Figure 10) than in NA (except for EU3). Since CO is a
primary pollutant, its error is affected by the diurnal dynamics of the PBL height, which is most problematic in
winter, when modelled PBL has the tendency to become too stable too early, anticipating the evening
transition (Pleim et al., 2016). In fact the biases of CO and that of PM$_{10}$ (another primary pollutant) in winter
are highly correlated for almost all models (not shown), indicating a common causes of the error.
The error due to variance in EU (under-estimated by the models) and *mMSE* are significant in the DU and SY
components in winter (Figure 10a). In particular, the variance error of winter DU is small compared to the
*mMSE*, which accounts for almost the entire DU error, up to over 30 ppb. For SY, the model SILAM_H shows an
*mMSE* error of over 75 ppb, the variance part being approximately null. On average, the DU and SY errors are
approximately similar for all EU models (~45ppb for DU and ~65ppb for SY), indicating some common error
source such as missing sources and process and strong emission underestimation at these time-scales. A
further reason could stem from the lack of temperature dependent emissions (the current emission inventory



processing approach employs constant temporal emission profiles, and therefore cold/warm episodes are not
incorporated in the modelled emissions while these episodes do affect real-world emissions). The lack of
temperature-dependant emission is likely to have a strong effect for CO, as about 50% CO emissions comes
from
residential heating (at least in mid/north European countries). A test to this hypothesis is currently under
investigation by running the CCLM-CMAQ model with a set of emissions using temperature data for the
temporal disaggregation for residential heating emissions.
While the SY error is comparable for the two continents, the DU and ID errors are remarkably higher in NA (all
sub-regions, also due to an excess of variance) and for several instances comparable or even higher than the LT
error. With the exception of the WRF-DEHM model (variance error negligible), the DU and ID error for the NA
models are due to both *mMSE* and variance.
3.3.1.A SENSITIVITY SIMULATIONS WITH REDUCED EMISSIONS AND BOUNDARY CONDITIONS
Additional sensitivity runs have been carried out by the majority of modelling groups, in which the amount of
anthropogenic emissions are reduced by 20% in both the boundary conditions  and the modelling domain. It is
instructive to assess the error variation between the sensitivity runs (denoted as 's20%') and the base case for
primary species such as CO:
$$\%RMSE = 100 * \frac{RMSE_{CO}^{s20\%} - RMSE_{CO}^{base}}{RMSE_{CO}^{base}}$$
Figure 11 reports the error variation for central Europe (sub-region EU2), where the effect of local CO
outweighs the influence of the CO entering from the boundaries (similar plots for the other two EU sub-
regions are reported in the Supplement). A decrease of 20% CO produces a RMSE variation of ~10% (averaged
over models and components). A naïve projection indicates that a reduction of 100% (thus removing CO from
emissions and boundary conditions altogether) would produce a variation of the error of ~50%. The sign of the
error variation indicates that there are circumstances where a reduction of the base case emissions is actually
beneficial as the error is reduced (even substantially in the instances where the emissions were overestimated
in the base case).
The DU component for CO is the most sensitive to emissions changes with an average of ~24% error variation
in summer. The SILAM model is the most sensitive to changes in the amount of pollutants entering the
domain. Striking error differences with respect to the base case are detected for summer CO (DU error
improved by 50%), possibly pointing to false peaks in the base case that contribute heavily to the RMSE (as
suggested by the low correlation coefficient, Table S5). The reduction of the emission by 20% lowers the peaks
and could be the explanation for the improvement observed for the 's20%' scenario for SILAM.
3.3.2 NO
NO is emitted by both natural and anthropogenic sources and its chemistry patterns are closely connected to
those of $NO_2$ and ozone. Due to the ozone-NO titration reaction (timescale < 1 hour at all temperatures), the
uncertainty in emissions, transport, and vertical mixing dominates the uncertainty in chemistry. As no
observational data was available for NA, the discussion is limited to EU. The European Environment Agency
(EEA) reports an estimated uncertainty for $NO_x$ emission of ~20% (EEA, 2011); Vestreng et al. (2009) found ±8-
25% uncertainties in EU $NO_x$ emissions, in line with other similar bottom-up uncertainty studies (see Pouliot et
al., 2015). A further source of uncertainty and model to model difference is the vertical emission profiles
adopted and how this is interpolated to the vertical grids used by the models. Within the SILAM model, for
example, the vehicular traffic emissions are released largely at the bottom of the first layer and this sub-grid
information about the vertical location of the plume used in the vertical transport scheme further supresses
the mixing to the upper layers, thus keeping the surface concentrations higher.



The analysis of the RMSE for NO in Figure 12a shows how the largest modelling error for NO occurs in winter
and autumn, similar in magnitude for EU1 and EU2 (~7ppb), while is more than double in EU3 (up to 30 ppb).
The DU and SY errors are comparable in magnitude (although the DU error is slightly higher), and are
approximately evenly distributed among the models. Also for NO the error of the SY component is model-
independent, as noted for CO and as will be discussed for ozone and $PM_{10}$. Because it is mainly composed by
*mMSE* error (Figure 12b) it can be hypothesized that the unexplained meteorological variance is responsible
for the majority of the SY error.
The winter bias and variance errors are predominantly negative, indicating model underestimation and
reduced variability. The opposite holds for the two instances of SILAM, for which the bias and variance are
positive (all sub-regions). This can be associated with the underestimated ozone concentrations in this model
also the applied vertical emission profiles mentioned earlier for this model could have an influence. The
correlation coefficient varies greatly by model, by components and by season and typically degrades for the
summer seasons (LT component, most models). The SY component also exhibits low values of *r*, especially in
summer for EU1 and autumn (Table S6). The large variability of the correlation coefficient indicates that the
models are not able to capture the fluctuations of this important precursor at all scales.
From the error decomposition plots (Figure 12b) it emerges that
-    the LT components shows a *mMSE* error approximately uniform for all modelling systems (between
~3 and 4 ppb);
-    in the majority of the cases the *mMSE* error dominates the ID, DU and SY components;
-    the SY component has an error comparable to that of DU for the *mMSE* part, but overall higher due to
a predominant lack of variance (as high as 50% of the total SY error for some models).

Due to its fast chemistry and short travelling distance,  the error of representativity for NO (mismatch of the
area of representativeness between models with grid spacing of ~15 km up to 50 km and point measurements)
is likely more significant than for other pollutants with longer life-time. NO is almost a primary pollutant with
negligible deposition (Wesely and Hicks, 2000) and small influence of the boundary conditions (Giordano et al.,
2015), therefore observational sites are affected by local scale effects in the range of a few kilometres, below
the grid spacing of the majority of the models. This has the effect of higher observed mean values compared to
the models (enhancing the bias error) and stronger variability in the observations than the models (variance
error).
The correlation between the bias of NO with the bias of the other species reveals strong links at several
temporal scales (less for the DU time scale though) and also in terms of processes, although it varies greatly by
model. For instance, *corr*(bias$_{NO}$, bias$_{O3}$) is overall strong (and negative) for the majority of the models, but for
different time scales, i.e. stronger for the SY components for some models (e.g. LOTOS-EUROS), or for the LT
(SILAM), or for the DU (CHIMERE). Additional analysis are envisioned to determine the causes of such a
behaviour.
3.3.2.A SENSITIVITY SIMULATIONS WITH REDUCED EMISSIONS AND BOUNDARY CONDITIONS
The analysis discussed in Section 3.3.1.A is repeated here for NO and results are presented in Figure 13. A
decrease by 20% of the amount of NO in the domain produces a variation of RMSE of ~8% (averaged over
models and spectral components). A naïve projection indicates that a reduction of 100% (thus removing the
production of NO from emissions and boundary conditions) would produce a variation of the error of ~35%.
Such an amount is less than that found for CO (~50%, section 3.3.1.A), which is consistent with the
photochemical processes involving NO but not CO.
The LT component is the most sensitive to changes for NO, with an average of ~17% error variation ((and up to
20% in autumn, both positive and negative). Again, the SILAM model is the most sensitive to changes in the





amount of pollutants entering the domain. Remarkable differences between the 's20%' scenario and the base
case are detected for summer and autumn (LT error variation of 100%) (Figure 13). The improvement of the
error of SILAM (and of the other models) for the 's20%' scenario is due to the overestimation of NO mean
concentration in the base case (positive bias, Table S6).
### 3.3.3 $NO_2$
Primary $NO_2$ is emitted by a variety of combustion sources and plays a major role in atmospheric reactions that
produce ground-level ozone. $NO_2$ is also a precursor to nitrates, which contribute to PM formation. As for NO,
only a small portion of the total error is expected to stem from the boundary conditions. The AQMEII3
modelling systems attribute a fraction of $NO_2$ emission ranging between 3% and 10% of the total $NO_x$
emissions (some models treat the $NO_2$ emission from the transport sector differently, see Table 1). The results
of the error analysis discussed hereafter do not reveal, though, grouping of model behaviour consistent with
the choice of the $NO_2$ to $NO_x$ emissions ratio.
The RMSE distribution (Figure 14a,b) shows a marked model-to-model variability in the LT and DU
components, while it is more uniform for the SY component, also in the seasonal stratification. Moreover, the
error distribution shows to be weakly dependant on the specific sub-region (for both continents, especially for
the DU component), suggesting that regional features (e.g. differences in climate between the regions) have
little impact on $NO_2$ performance, which is most affected by chemistry and error in the meteorology. Local-
scale features (e.g. representation of urban / rural emission differences) may still be important, but they may
have similar errors in all regions.
The largest error occurs in winter (both continents), and is shared approximately equally between the SY and
DU components (for some models the SY and LT errors are comparable due to the little bias).
The bias is the main contributor to the $NO_2$ error and stems from a model under-prediction of the mean
observed concentration (but, with the exception of the winter season, is positive for WRF-CMAQ in NA) (Table
S7). However, the tendency of $NO_2$ measurements to be likely overestimated by the majority of commercially
available instruments for detecting $NO_x$ (Steinbacher et al., 2007) needs to be taken in to account. The
magnitude of the bias higher in EU (from ~1.3pbb of WRF-CMAQ1 in EU1 to ~-12.5ppb of CCLM-CMAQ in EU3)
than in NA (the maximum being ~5.5ppb in NA3 by the WRF-DEHM model), with the mean observed values
being of 11.5ppb and 10.5ppb for EU and NA, respectively.
The correlation coefficient is typically higher in spring/autumn and poorer in summer/winter (in summer there
are several instances of negative correlation) (Table S7). The LT component for EU, and the LT and SY
components for NA, are those with higher correlation coefficients, while SY and DU are the poorest in EU and
DU the poorest in NA (but still higher than ~0.4).
The median value of the modelled accumulated deposition per unit area (Figure 8) for $NO_2$ ranges from ~0.4 to
~1.9 $kg/km^2$ for EU (nine models) and from ~0.3 to 2.3 $kg/km^2$ for NA (two models). With the exception of the
WRF-DEHM model (similar values for EU and NA of 0.3-0.4 $kg/km^2$), the modelled values for $NO_2$ deposition
are uniform across the EU models, while the deviation between the two NA models for deposition is not
negligible, also in light of the different native grid sizes of 50km and 12km (WRF-DEHM and WRF-CMAQ,
respectively). Therefore, for the majority of the EU models model-to-model differences in the error are
unlikely due to significant difference in the deposition, while it remains a possibility for NA.
The magnitude of the error for $NO_2$ resembles that of NO and ozone, although the apportionment reveals
significant differences. In fact, while NO includes variance error and a uniform share of *mMSE*, the LT error of
$NO_2$ for winter is almost completely determined by the bias, for both continents (Figure 15 and Figure 16). The
other $NO_2$ spectral components (ID, DU, SY) reveal more profound difference with respect to NO, both in terms
of bias and of error apportionment. The ID error for $NO_2$ is even smaller than that of NO (less than 1 ppb) and



can be regarded as noise. Also the DU (~1.5 ppb) and SY (~1 ppb) errors are considerably smaller than for NO
(both continents), although the DU error presents some excess of variance for WRF-CMAQ3 and the two
instances of the CHIMERE model (Figure 15).
The model-to-model variability of RMSE for the LT component Figure 15) is very similar to that of NO (Figure
12), while the DU variability resembles that of ozone (Figure 18), although for $NO_2$ the DU error is lower in
magnitude and more uniform across seasons.
Moreover, $NO_x$ observations are strongly affected by local emissions and thus the error may stem from the
incommensurability of comparing grid-averaged values against point measurements highly affected by local-
scale emissions. However, the error apportionment analysis carried out separately for 'rural' and 'urban'
background stations (the area type classification is taken for the stations metadata) does not reveal any
relevant differences (Figure 15 for EU2 and Figure 16 for NA1), if not a slight increase of the variance error
over both continents.
### 3.3.4 OZONE
Due to the adverse effects on human health and to the impact on climate, tropospheric ozone is regulated in
EU and NA and substantial efforts are made to improve the models' predictive skill for this pollutant.
Tropospheric ozone can be either transported from regions outside the modelled domain, be the result of
stratosphere/troposphere exchange, or be produced locally by photochemistry through oxidation of VOCs
(volatile organic compounds) and CO in the presence of $NO_x$ and sunlight. Due to its photochemical nature,
ozone production is directly influenced by temperature through speeding up the rates of the chemical
reactions and increasing the emissions of VOCs (e.g. isoprene) from vegetation (Jacob and Winner, 2009).
Along with dry deposition, chemistry can act as local sink to ozone depending on the photochemical regime.
Results of the AQMEII3 modes for ozone are reported in Figure 17and Figure 18, and in Table S4. Overall, the
correlation between modelled and observed ozone time series is higher for the winter and fall seasons than
the spring and summer seasons in EU, while the opposite holds true in NA where the maximum correlation is
observed in summer (all sub-regions) (Table S4). In EU, the RMSE is generally lower in winter than in the warm
seasons (summer and spring) (RMSE in summer ranges between 4.3 ppb of WRF/Chem1 in EU1 and 21 ppb of
WRF-CMAQ1 in EU3), with the exception of the CCLM-CMAQ model for which the RMSE peaks in autumn (all
sub-regions).
Due to the strong and well defined diurnal cycle characterized by ozone formation and loss, the correlation
coefficient is generally higher for the DU component, while it tends to be lowest for the SY component (Table
S4 and Figure 18). The SY component often exhibits the lowest correlation among all components, especially in
summer (EU) and spring (NA), possibly due to the combined effect of transport of precursors, deposition and
chemistry (formation/depletion of ozone from precursor emission in the regions where the ozone is
transported) (Bowdalo et al., 2016). However, the SY error is generally small (~2-3 ppb, although higher for
EU3, where the SY error is double that of the other sub-regions) and is mostly due to *mMSE*, it is thus
characterised by poor coefficients of determination and underestimated variability (Eq 7). Therefore, the SY
component suffers from low precision (for some models $r < 0.3$) meaning that the variability of the synoptic
mechanisms needs further attention, especially in the meteorological conditions leading to high ozone level
episodes, especially in relation to temperature, cloudiness, and radiation. The WRF/Chem2 model (having the
highest error for temperature, Figure 2b) reports the largest SY error for ozone (especially the variance part).
For this model, the correlation between the ozone and the Temp error for the SY component $corr(bias_{O3},
bias_{Temp})_{SY}$ is 0.44 for the summer months in EU2 (not shown), among the highest, which helps explain part of
the SY error for ozone. In order to characterise better the *mMSE* part of the error for the periodic components,
such as DU and SY, analysis of the phase and amplitude are foreseen.




The error of the DU component is largely due to the *mMSE* term (Figure 18a) which is comparable for all
models in the range of 2-5 ppb, with some significant excess of variance for WRF-CMAQ2 and WRF-CMAQ3 in
EU2 (~5 ppb). One possible reason is the dynamics of the nocturnal PBL as well as the timing of the ozone
cycle, with an either too fast or too slow modelled ozone peak (e.g. Pirovano et al., 2012). Limitations of the
models to reproduce the amplitude and phase of the daily ozone cycle were already highlighted in the first and
second phase of AQMEII, mostly related to the representation of night-time and stable conditions. Further, the
variance error for WRF-CMAQ2 and WRF-CMAQ3 can be induced by the bias of temperature and/or
concentration of ozone precursors. For WRF-CMAQ2 (WRF-CMAQ3), $corr(bias_{O3}, bias_{Temp})_{DU}$ is 0.88 (0.94) and
$corr(bias_{O3}, bias_{NO2})_{DU}$ is 0.86 (0.83) (summer months, EU2) (not shown), which indicates that the bias of the
Temp and $NO_2$ fields are strongly associated with the error of ozone at the DU scale. According to Bowdalo et
al. (2016) the bias of the ozone amplitude cycle linearly evolves with $NO_x$ emissions, suggesting that correction
of the error for ozone needs to start from $NO_x$ emissions. Otero et al. (2106) have shown that meteorological
drivers account for most of the explained variance of ozone, especially over central and northwest Europe.
One of the main drivers of ozone is the daily maximum temperature, in relation to the effect of temperature
on emissions of VOCs. Therefore, while part of the bias error is likely due to $NO_x$ emissions, the *mMSE* and
variance error are also induced by error in meteorology. Other sources of biases are transcontinental transport
in winter (Hogrefe et al., 2011) and missing processes during spring and summer, such as the large scale effect
of forested areas.
The error in the LT component is dominated by the bias error (Figure 18) (almost completely for NA) although
with significant exceptions in EU (for CCLM-CMAQ the *mMSE* error of the LT component is larger than the bias
portion). The May-September ozone LT bias for EU2 peaks at 12-13 ppb (WRF-CMAQ1), while it is ~6 ppb in
NA3 (but in excess of 20ppb in NA2 by the WRF-DEHM model) (the yearly average measured ozone mixing
ratio is  26.5 and 29ppb for EU and NA, respectively). The bias of the precursors and of the meteorological
fields is typically highly correlated with the bias of ozone. For instance, in EU2 for the WRF-CMAQ1 model
$corr(bias_{O3}, bias_{Temp})_{LT}$ is 0.65 and $corr(bias_{O3}, bias_{WS})_{LT}$ is 0.81 (summer months).
Although the concentration peaks are associated with the ID and DU components, the contribution to the total
error of the ID component is small (< 2ppb) due to the flattening of the spikes operated by the spatial
averaging carried out prior of the spectral decomposition. The noise of the ID component is reflected by the
correlation coefficient being lower than the correlation of the DU component. However, the acf (auto-
correlation function) associated with the signal of the ID component is well structured and periodic (not
shown), indicating that the ID component for ozone is not entirely a white noise-type of signal, but
incorporates useful information, although there is the possibility that the ID periodicity is due to a periodic
leakage from the DU component, due to the imperfect separation of the ID and DU components. This latter
aspect will require additional investigation.
3.3.4.A OZONE VERTICAL PROFILES
A further analysis aimed at detecting errors introduced by the vertical mixing is carried out by using modelled
and observed ozone profiles from ozonesondes. A summary of the records provided by the ozonesondes for
ozone are reported in Table 3. Plots of the simulated and observed ozone levels at fixed heights (through the
ENSEMBLE system models and measurements are paired at the heights of 0, 100, 250, 500, 750, 1000, 2000,
3000, 4000, 5000, 6000 m) are reported in Figure 19 and Figure 20. The ozonesonde data are mainly available
during daylight, although two stations with night-time data are available for NA (Table 3).
Overall, the general tendency of the models in both continents is to underestimate the ozone levels above the
PBL, suggesting that not enough ozone enters the continental domains through the inflow boundaries. The
most significant underestimation (~10 ppb) is observed at the two stations closer to the west boundary for EU
(stations 318 and 043). The boundary layer deficit of ozone is a long standing issue, as similar conclusions were
derived for the first (Solazzo et al., 2013) and second (Im et al., 2015; Giordano et al., 2015) phase of AQMEII,




as well as in other studies (Katragkou et al., 2015), emphasizing the strong dependence of regional models on
the lateral boundary, whose effects propagate far into the interior of the domain.
Towards the interior of the EU domain (stations 134, 157, 242) the profiles are in closer agreement with the
observations, with the WRF-CMAQ1 model performing the best throughout the troposphere, possibly due to
the overestimation of the entrainment of upper tropospheric ozone, as revealed by the strong gradient of
WRF-CMAQ1 at 6000m (Figure 19).
For NA (Figure 20), the general tendency is of good agreement within the PBL and underestimation aloft for
the WRF-CMAQ model and of overestimation (stations 107, 456, and 458 – afternoon/night launches) at the
surface and mild underestimation above the PBL for the WRF-DEHM model.
3.3.4.B RELATIONSHIP BETWEEN THE BIAS OF OZONE, $NO_x$ AND TEMPERATURE
The relationship between the bias of NO and the bias of ozone is reported in Figure 21 for the EU2 region
(similar plots including the bias of $NO_2$ for EU and NA are provided in the supplementary material). A linear
relationship between the biases of the two species is detectable, more evident in winter. Large, positive ozone
bias is driven by underestimation of NO (a primary species) whereas the largest negative ozone bias
correspond to the largest overestimation of NO. The role of the temperature bias is less clear, but the $NO_2$ and
ozone relationship (Figure S7) suggests that large $NO_2$ bias is associated with temperature under-prediction.
The partition of $NO_x$ emission into primary NO and $NO_2$ seems to suggest that the models adopting a 95%-5%
ratio suffer lower ozone bias (at least in winter), although in general the clustering of models based on the
$NO/NO_2$ share of total $NO_x$ emission is far from robust. A simple linear regression between NO bias and ozone
bias (based on the yearly time series) among the EU models suggests that the $NO_x$ and temperature biases can
explain, on average, ~35% and ~16% of the variability of the ozone bias, respectively.
3.3.5 $SO_2$
$SO_2$ is another primary regulated pollutant which, in EU and NA, is mainly emitted from coal power plants and
also from the residential heating and waste disposal sector. $SO_2$ acts as a precursor to sulphates, which are one
of the main components of PM in the atmosphere. Any error in $SO_2$ is likely inherited by these secondary
species. The EEA reports an estimated uncertainty for $SO_2$ emission of ~10% (EEA, 2011), therefore $SO_2$
emissions are expected to be more accurate than $NO_x$ emissions. This is reflected in the low bias in both
continents (~1-2ppb in winter, mostly due to model underestimation) (Figure 22 and Figure 23). The averaged
observed concentration of $SO_2$ is of 1.92ppb and 2.7ppb in EU and NA, respectively.
The seasonal modelled error for $SO_2$ ranges, on average, between 0.65 and 1.3ppb in EU and between in
excess of ~1 and 5ppb in NA (the maximum error in NA2), peaking in autumn.
In EU and NA1, the error of ID, DU and SY components is comparable for all seasons and, on average, below
0.6ppb. There are some exceptions, most notably the WRF-CMAQ3 model, which is the only one significantly
biased high (Figure 23a) and shows an excess of variance significantly larger than the other models.
The large variability of the model-to-model error (especially in EU) and correlation coefficient in both
continents is an indication that the mechanisms governing the initial mixing and subsequent transport and
chemical transformation suffer from different sources of error, mostly covariance, at all scales. In no instance
is the correlation coefficient consistently above 0.5 for all seasons and spectral components while there are
several instances of negative correlation between the spectral components of observed and modelled $SO_2$ (e.g.
CCLM-CMAQ model in EU and several others). The poor correlation coefficient of, especially, the ID and DU
components for both continents, indicates that the peaks of the $SO_2$ concentration are not caught by the
models, leading to low precision. Although the mean fluctuations are, generally, well reproduced (low variance
error in both continents), it remains a significant portion of unexplained variance (*mMSE*) error, which can
derive from meteorology and chemistry. Bieser et al. (2011b) showed that the height of the release and



vertical distribution of the $SO_2$ emission influence the $SO_2/SO_4$ ratio as the oxidation (aging) of $SO_2$ is more
effective if the emissions are higher up. As power plants are the major source of $SO_2$ further analysis should
investigate the impact of differences in the vertical emission distribution between models.
3.3.6 PARTICULATE MATTER
Particulate matter (PM), both in the fine and coarse fraction, is directly emitted by biomass and fossil fuel
combustion in domestic and industrial activities, and also formed from precursors in the atmosphere.
From the AQMEII3 suite of model runs, the error for PM is evaluated for $PM_{10}$ in EU and $PM_{2.5}$ in NA. The
choice is dictated by the availability of hourly measurements in the two continents. The RMSE distribution is
reported in Figure 24 ($PM_{10}$ for EU) and Figure 25 ($PM_{2.5}$ for NA). The error distribution for EU reveals that,
despite the large numbers of modelling options and parameters characterising the chemistry and physics of
particles, the error distribution for DU and SY is homogeneous among the EU models. For these components
the error is approximately uniform over seasons, although with some exceptions (significantly higher in EU3,
although based on two receptors only). EU3 is a small area compared to EU1 and EU2, but is densely
populated, intensively farmed, with a large amount of wood burning in winter, and  agricultural area in
summer. It is surrounded by mountains and stagnant flow conditions are predominant. It is, thus, a challenging
area for current modelling systems, especially for primary species such as PM.
The LT component shows some significant model-to-model variations due to the WRF-CAMx and WRF-CMAQ1
models which have lower error in spring and summer compared to the other models, while the CCLM-CMAQ
model has higher LT error in EU1.
The magnitude of the SY error in EU is, on average, of ~6 µg m$^{-3}$ during winter, with a peak of 10.5 µg m$^{-3}$ in
EU2 (WRF-CAMx model). The magnitude of the DU error is lower (~2-2.5 µg m$^{-3}$ in EU1 and EU2, and ~5-6 µg
m$^{-3}$ in EU3) with the largest share in autumn, spring, and winter and slightly lower in summer. The error of the
LT component ranges between ~11-15 µg m$^{-3}$ in EU1 and EU2 and up to 25 µg m$^{-3}$ during winter in EU3.
The analysis of the correlation coefficient reveals that the model to model differences in the correlation
coefficient with the observed component time series tend to be most pronounced for the DU and ID
components, indicating that these two components are pivotal in determining the overall model skill in terms
of capturing observed fluctuations in $PM_{10}$ concentration. In more detail, the correlation is poor for the DU
component (especially in EU2 and EU3, Table S9), possibly due to PBL dynamics and emission profiles (as
discussed above for the RMSE at the DU scale). The LT component has correlation values highly varying among
models and, for the same model, among seasons (e.g. the LT correlation of the WRF-CMAQ4 model in EU3 is
~0.9 during spring but only of 0.35 in summer).
In winter the LT and SY error is more severe likely due to the larger uncertainties in $PM_{10}$ emissions of
combustion processes (wood burning, residential heating) (Van der Gon et al., 2014), as well as due to the
current limitations in modelling the vertical mixing during stable conditions, as mentioned for the gaseous
species (especially CO, being another primary species). The majority of the EU models show an LT error in
winter between 12 and 16 µg m$^{-3}$, eight models above 16 µg m$^{-3}$ and only one (WRF-CAMx) below 10 µg m$^{-3}$.
The SY winter error exceeds 5 µg m$^{-3}$ for all models (all sub-regions) and three instances (WRF-CAMx,
WRF/Chem1 and WRF/Chem2, this latter showing the highest accumulated deposition for $PM_{2.5}$, Figure 8b)
report an error above 7.5 µg m$^{-3}$. All the remaining models have comparable *mMSE* and variance errors (Figure
26), and are biased low (model under-prediction), possibly due to missing PM source and overestimated
surface wind speed.  As for the WRF-CAMx model, the low bias on LT component and the relatively high error
on covariance in SY fraction suggest that the model was able to capture the mean magnitude of PM
concentration over the entire year, but failed in reconstructing the correct variability of the different episodes,
whose timing is generally driven by the synoptic time scale.



The PM$_{2.5}$ evaluation in NA is restricted to two models, WRF-DEHM and WRF-CMAQ, which show comparable
error (Figure 25). The WRF-CMAQ (WRF-DEHM) model has an error ranging between ~3.5 (~2) and ~6 (~8.5)
µg m$^{-3}$. The main contribution to the total error stems from the LT component (predominantly negative bias)
and from the SY component (2-3 µg m$^{-3}$). The DU component contributes to about 1.5 µg m$^{-3}$ (comparable
*mMSE* and variance error).
Both NA models are biased low in summer (all sub-regions), which can be attributed to limitations in the SOA
mechanism (Zare et al., 2014). Because of the higher contribution of primary PM$_{2.5}$ to total PM$_{2.5}$ during
wintertime, differences in horizontal and vertical resolution (Table 1) likely contribute to the difference in
wintertime LT bias. The correlation coefficient for the two models is in general higher in winter (full time
series) and deteriorated for the DU component (all seasons and sub-regions).
As inferred for the species discussed above, the uniformity of model behaviour is indicative of errors stemming
from external fields, likely emissions, where missing sources of PM can affect the error within certain time
scales for all models. Further common causes of error are intrinsic to the model-observation comparison as
modelled PMs is commonly dry while this is not always the conditions for the measurements. For instance, the
filter-based gravimetric measurements as recommended by the European Committee for Standardization
(CEN) are likely to retain part of the particle-bound water after the filter conditioning at a constant
temperature of 20° C and relative humidity of 50%. Recent findings by Prank et al. (2016) report the aerosol
water content from the gravimetric measurements to range between 5 and 20% for PM$_{2.5}$ and between 10 and
25% for PM$_{10}$. The particle-bound water was found to be associated with hygroscopic particles such as
sulphate, nitrate, and organic compounds. This remaining water content can be up to approximately 10-35%
depending on the chemical composition of aerosols being measured (Tsyro, 2005, Kajino, et al., 2006, Jones
and Harrison, 2006). The water aerosols should therefore be accounted when compared with these
measurements. Part of the problem lies in secondary organic aerosol. In winter, in particular for wood burning
part of the emissions are condensable gases that rapidly change to the aerosol phase (Van der Gon et al 2014),
but are missed since they are not part of the presently used PM emission inventory. In summer, biogenic
emissions that contribute to SOA formation and their yields are quite uncertain. A good representation of SOA
is still a problem for all models. In spring, the application of manure and fertilizer leads to peaks of NH$_3$
emissions and subsequent NH$_4$ aerosol formation, contributing to PM$_{10}$ and PM$_{2.5}$. The timing of these
emissions is parameterized based on long-time averages, whereas in practice they are strongly related to
meteorology. This can explain part of the discrepancy on the diurnal to synoptic time scale (Hendriks et al
2015).

# 4. MEMORY OF THE SIGNAL AND REMOVAL PROCESSES: THE CASE OF OZONE
The evaluation of the removal processes (chemical transformation, transport, and deposition) is difficult to
assess in isolation with respect to other sources of error because of the bias of the signal. In this section we
propose a bias-independent spatial analysis aimed at the quantification of the 'memory' of the signal. The
analysis seeks the time interval (or memory) after which the signal loses any memory of its past. The memory
of the modelled and observed signals is then compared. The methodology consists of:
1. calculating the autocorrelation function (*acf*) of the modelled and observed LT component;
2. then, calculating the quantity $acf_{mod=0}$ and $acf_{obs=0}$, i.e. the lag (time interval) where the *acf* of the modelled
and observed LT component falls to zero, and finally
3. determining the difference between the two, yielding the difference between the modelled and the
observed memory of the signal:

$$\Delta_{memory} = acf_{mod=0} - acf_{obs=0} \qquad \text{Eq 9}$$






The *acf* is simply a measure of the degree of associativity of a time series with its lagged version. The associativity is typically measured through the correlation coefficient, and the lag extends from one time step (one hour in the case of hourly time series) to, generally, a third of the length of the time series. Because the correlation is bias-independent, we conclude that the *acf* is also bias-independent therefore information from $\Delta_{memory}$ is useful for the interpretation of the variance and covariance errors discussed in section 3.1. The memory of the signal is different from the persistence indicator (previous day concentration) as used e.g. by Otero et al. (2016) for accounting for pollutant episodes. As we deal with the LT component of the signal, short term and synoptic episodes are in fact filtered out in this analysis.

In the supplementary material Figure S9 and Figure S10, the *acf* for the network-wide spatial average and for the full year is reported. The *acf* is calculated for the LT component of the observed (first panel) and modelled ozone time series. The zero of the *acf* and the slope of the decay of *acf* of the observations (approximately a straight line from 1 to 0 in 2000 hours) are replicated by the models with various degree of success (Figure S10). Our intent is to apply this analysis to the seasonal ozone time series at each receptor, and derive useful information about the modelled removal/production processes. The spatial analysis is proposed for ozone, for the months of May to September (Figure 28 and Figure 29) and for the full year (supplementary material Figure S9 and Figure S10).

The average life time of ozone in the troposphere is of approximately 20-30 days (Solomon et al., 2007). By analysing the LT component (processes > ~21 days) we therefore screen out the daily removal/transformation due to chemistry and can focus on seasonal transport, deposition of the free tropospheric ozone, long term chemistry (seasonal changes in vegetation that affect biogenic VOCs emissions and ozone deposition, and also the monthly variations applied to the anthropogenic emission) and influence of boundary conditions. The structure of the *acf* also benefits from the removal of short time scale processes as it is less affected by noise and the results are easier to interpret.

The spatially distributed $\Delta_{memory}$ shows some clear regional effects for the majority of the models. The $\Delta_{memory} > 0$ along the Mediterranean coast of Spain and France, with some severe excess of ozone production (or underestimation of sinks) in southern/central France for some models (SILAM, WRF-CAMx, WRF-CMAQ1, WRF-CMAQ2 and especially the L.-Euros model, for which the *acf* at the French receptors did not reach zero).

The region covering the Po valley, Austria and extending into the continental eastern EU is affected by negative $\Delta_{memory}$ (sometimes a deficit of one month for some models). The negative memory indicates that the observed signal is more persistent than the modelled one, and that long term weather transitions are smoother in gradient and longer in duration, and thus that the seasonal modulation of the signal is overestimated by the models, thus producing variance error. Coupling the two behaviours (excess of ozone in south France and south Spain with the short memory from the interior of east EU extending to the Po valley), might indicate an easterly synoptic transport of ozone (or of LT ozone precursor, such as the impact of $CH_4$ and CO on OH and photochemistry) masses whose duration is underestimated by the models. The relationship between the sign of $\Delta_{memory}$ and the land use type (vegetation vs urban) is subject of on-going investigations in the attempt to determine the role of VOCs emissions and deposition over different land types.

The central part of Germany is affected by positive (on average in the range of 7 to 10 days) $\Delta_{memory}$, mostly visible for the HTAP-emission based SILAM and Chimere results in contrast with the MACC-emission based ones of the same models. When the HTAP inventory is used the largest differences are observed in the central EU regions, indicating that also the LT chemistry plays a role.

The deposition aspect of removal can be equally important as transport and chemistry. The memory of the signal directly depends on the amount of ozone available and a large, negative $\Delta_{memory}$ might indicate that the deposition is too high.



For NA (Figure 29), the feature common to all models is the excess of removal in the Southern Atlantic coast
and across the Eastern Canadian border. In contrast, the central-east part of the US shows large positive
$\Delta_{memory}$ values (up to ~1.3 month for the WRF-DEHM model), with the exception of the WRF-CMAQ model,
which is overall in line with the observed memory of the signal in this part of the domain. This result agrees
with the seasonal phase analysis for ozone in global models by Bowdalo et al. (2016), where a delay of up to 4
months was detected for east USA.
The west coast has a mixed behaviour, but prevalently $\Delta_{memory}$ is negative. The hypothesis that too little
ozone enters the domain trough the boundary conditions is contradicted by the $\Delta_{memory}$ ~0 for the full year in
the west coast (see Figure S10). A potential excess of transport in this region also seems to be contradicted by
the large number of stations for which $\Delta_{memory}$ is positive. A possible conclusion is that localised biogenic
emission sources, radiation budget, and deposition are the main factors responsible for the negative sign of
$\Delta_{memory}$ in this region.

## 5. CONCLUSIONS

The work presented in this paper summarises the results of the ongoing third phase of the AQMEII activity
focusing on AQ model evaluation, applied to the continental scale domains of Europe and North America. The
evaluation of the AQMEII3 suite of model runs is carried out for surface temperature and wind speed, and for
the species CO, NO, $NO_2$, ozone, $SO_2$, $PM_{10}$ (EU) and $PM_{2.5}$ (NA). Additional analyses making use of emission
reduction scenarios (CO and NO) and vertical profiles have also been performed.
This work is primarily meant to provide a wide overview of the performance of current regional AQ modelling
systems and to set the basis for additional diagnostic analysis that is currently in progress.
The model evaluation is carried out by quantifying the components of the error (bias, variance, $mMSE$) at four
time-scales (ID, DU, SY, LT) each describing physical processes in a specific time range. The bias and variance
measure the departure from the first and second moment of the observed distribution (mean and standard
deviation), while the $mMSE$ accounts for the unexplained observed variability and relates to the ability of the
models to reproduce timing and shape as measured by the correlation coefficient. The apportionment of the
error to the relevant time-scales and the analysis of the quality of the error have revealed that the LT bias is,
by far, the first cause of error, followed by the variance error (fluctuations about the mean value) of the DU
component and the unexplained variance of the DU and SY components, depending on the species and
season. In more detail:
- The mean concentration of the primary species (NO, CO, $PM_{10}$, $SO_2$) is underestimated by the vast majority
of the models in both continents, more markedly during the winter and autumn seasons. The largest share
of error for these species is the bias of the LT components, most probably due to emissions and the effects
of comparing point measurements to volume averaged concentrations.
- The meteorological fields of temperature and wind speed are consistently biased low and high,
respectively. Based on the results of the European models directly driven by the global fields for
meteorology (e.g SILAM, Chimere) the error for wind speed is of ~0.5-1 ms$^{-1}$ and of ~0.4-1.2K for
temperature. These errors can be considered as the uppermost limit the accuracy of the models can
currently achieve. The use of nudging and interpolation methods (specific to the configuration of the
meteorological model) can add more than 1.5K and 2ms$^{-1}$ to the total error. The analysis of the available
vertical profiles suggests that the models overestimate the wind speed within the PBL and vice versa above
the PBL, possibly inducing a net outward flux of pollutants at the PBL interface.
- Modelled CO is affected by high errors, uniformly across models and components, more pronounced in
winter and predominantly driven by the negative bias of the LT component, followed by variance error of
the SY component. Modelled NO and $NO_2$ also report negative bias but, in contrast to CO, there is



significant model-to-model difference in error variability, possibly due to the chemistry of $NO_x$. The SY and
DU errors of NO are comparable in magnitude (3-5 ppb) and mostly due to *mMSE* error. Preliminary
sensitivity investigations for CO and NO seem to suggest that at most ~50% and ~35% of the total error,
respectively, could be due to emissions. Finally, based on spatially averaged analysis, the error for $NO/NO_2$
is the same for urban and rural stations (i.e. the error is insensitive to the area-type of the stations).
• The error analysis for ozone shows large model-to-model variability for all errors and spectral components,
with the exception of the SY component for which the error is similar among models and possibly driven by
the error in temperature and in the boundary conditions, as modelled vertical ozone profiles near the
domain's boundaries are typically underestimated in both continents by all models. The bias is prevalently
positive, while the variance error is generally small. While the bias error for ozone is likely driven by error in
$NO_x$ emissions, the error in meteorology may factor in determining the *mMSE* and variance error. In fact,
there are several models for which the bias of temperature and the bias of $NO_2$ are strongly associated
with the DU error of ozone. A simple linear regression between $NO_x$ bias and ozone bias (based on the
yearly time series) among the EU models suggests that the $NO_x$ and temperature biases can explain, on
average, ~35% and ~16% of the variability of the ozone bias, respectively. Ongoing analyses are focusing on
explaining the origin of the *mMSE* error by investigating the phase shift between the modelled and
observed DU and SY components as well as on looking at maximum daily values rather than to the full time
series.
• PM analysis ($PM_{10}$ for Europe and $PM_{2.5}$ for North America) reveals that, for Europe, the error distribution
for DU and SY is homogeneous and season independent among the models, despite the large numbers of
modelling options and parameters characterising the chemistry and physics of particles. A common source
of model bias (model underestimation, especially in winter) for $PM_{10}$ likely lies in the emissions (missing
sources) and in the overestimation of surface wind speed, whereas variance error may stem from PBL
dynamics under stable conditions and missing processes in the model (SOA formation is a known issue for
all models). The analysis of $PM_{2.5}$ (based on two models only) shows an excess of variance and low
correlation coefficient in the DU component, possibly due to the timing of the PM cycle. Further analyses
dealing with the PM components are needed to draw further considerations.
• The analysis of the memory of the ozone signal has revealed a strong model deficit in continental Europe,
where the seasonal modulation of ozone is overestimated by the majority of the models. The opposite
holds true in the continental US.
Although remarkable progress has been made since the first phase of AQMEII, both in terms of model
performance and also in terms of developing a more versatile and robust evaluation procedure, results of AQ
model evaluation and inter-comparison remain generic as they fail to associate errors with processes, or at
least to narrow down the list of processes responsible for model error. AQ models are meant to be applicable
to a variety of geographic (and topographic) scenarios, under almost any type of weather, season, and
emission conditions. For such a wide range of conditions the inherent non-linearity among processes are
difficult to disentangle and specifically designed sensitivity runs seems the only viable alternative. A model
evaluation strategy relying solely on the comparison of modelled vs. observed time series would never be able
to quantify exactly the error induced e.g. by biogenic emissions, vertical emission profiles and their
dependence on temperature, deposition, vertical mixing, chemistry, and the analysis approach presented in
this work is no exception. In fact, the methodology devised to carry out the evaluation activity in this study has
not succeeded in determining the 'actual' causes of model error, although providing much clearer indications
of the processes responsible for the error with respect to conventional operational model evaluation.
The highly non-linear nature of current AQ models requires the study of the relationships among error fields,
those of the meteorological drivers and those of the precursors. When the seasonal and spectral structures of
these relationships is analysed together with the error of the input fields (emissions and boundary conditions),
then it would be possible to diagnose and explain accurately the processes responsible for the error. Future
evaluation activities should envision sensitivity simulations and process specific analyses.





APPENDIX 1.
Following Hogrefe et al. (2000) and Galmarini et al. (2013) the time windows ($m$) and the smoothing
parameter ($k$) have been selected as follow:

$$ID(t) = \mathbf{x}(t) - kz_{3,3}(\mathbf{x}(t))$$
$$DU(t) = kz_{3,3}(\mathbf{x}(t)) - kz_{13,5}(\mathbf{x}(t))$$
$$SY(t) = kz_{13,5}(\mathbf{x}(t)) - kz_{103,5}(\mathbf{x}(t))$$
$$LT(t) = kz_{103,5}(\mathbf{x}(t))$$
$$\mathbf{x}(t) = ID(t) + DU(t) + SY(t) + LT(t)$$

Eq. S1

where $\mathbf{x}(t)$ is the time series vector. The additive property of the components whose summation returns the
original time series might be questioned. In the original work by Rao et al. (1997) it is highlighted the
importance of log-transform the components to stabilize the variance. In the case of log-transformation the
original time series is obtained by the product of exponential functions whose exponents are the spectral
components. For the purposes of the error apportionment analysis presented here, the results of using
additive time series component of log-transformed did not produce substantial differences.
A clear-cut separation of the components of Eq. S1 is not achievable, since the separation is a non-linear
function of the parameters m and k (Rao et al., 1997). It follows that the components of Eq. S1 are not
completely orthogonal and that there is some level of overlapping energy (Kang et al., 2013). Galmarini et al.
(2013) found that the explained variance by the spectral components account for 75 to 80% of the total, the
remaining portion being on account of the interactions between the components.
APPENDIX 2.
Statistical indicators:
Root Mean Square Error

$$RMSE = \left(\frac{\sum_{i=1}^{n}(M_i - O_i)^2}{n}\right)^{0.5}$$

Mean Bias (MB)

$$MB = \frac{1}{n}\sum_{i=1}^{n} M_i - O_i$$

Pearson correlation coefficient (r)

$$r = \frac{1}{n-1}\sum_{i=1}^{n}\left(\frac{M_i - \overline{M}}{\sigma_M}\right)\left(\frac{O_i - \overline{O}}{\sigma_O}\right)$$

Where $M$ and $O$ are the $n$-element modelled and observed time series, respectively, $\sigma$ is the standard
deviation and the overbar indicates temporal averaging.
ACKNOWLEDGEMENTS
We gratefully acknowledge the contribution of various groups to the third air Quality Model Evaluation
international Initiative (AQMEII) activity. The following agencies have prepared the data sets used in this
study: U.S. EPA (North American emissions processing and gridded meteorology); U.S. EPA, Environment
Canada, Mexican Secretariat of the Environment and Natural Resources (Secretaría de Medio Ambiente y
Recursos Naturales-SEMARNAT) and National Institute of Ecology (Instituto Nacional de Ecología-INE) (North
American national emissions inventories); TNO (European emissions processing); Laboratoire des Sciences du
Climat et de l'Environnement, IPSL, CEA/CNRS/UVSQ (gridded meteorology for Europe); ECMWF/MACC



(Chemical boundary conditions). Ambient North American concentration measurements were extracted from Environment Canada's National Atmospheric Chemistry Database (NAtChem) PM database and provided by several U.S. and Canadian agencies (AQS, CAPMoN, CASTNet, IMPROVE, NAPS, SEARCH and STN networks); North American precipitation-chemistry measurements were extracted from NAtChem's precipitation-chemistry data base and were provided by several U.S. and Canadian agencies (CAPMoN, NADP, NBPMN, NSPSN, and REPQ networks); the WMO World Ozone and Ultraviolet Data Centre (WOUDC) and its data-contributing agencies provided North American and European ozonesonde profiles; NASA's AErosol RObotic NETwork (AeroNet) and its data-contributing agencies provided North American and European AOD measurements; the MOZAIC Data Centre and its contributing airlines provided North American and European aircraft takeoff and landing vertical profiles; for European air quality data the following data centers were used: EMEP European Environment Agency/European Topic Center on Air and Climate Change/AirBase provided European air- and precipitation-chemistry data. The Finnish Meteorological Institute for providing biomass burning emission data for Europe. Data from meteorological station monitoring networks were provided by NOAA and Environment Canada (for the US and Canadian meteorological network data) and the National Center for Atmospheric Research (NCAR) data support section. Joint Research Center Ispra/Institute for Environment and Sustainability provided its ENSEMBLE system for model output harmonization and analyses and evaluation. Although this work has been reviewed and approved for publication by the U.S. Environmental Protection Agency, it does not necessarily reflect the views and policies of the agency.

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

TABLES















TABLE 1. PARTICIPATING MODELLING SYSTEMS AND KEY FEATURES

| Operated by | Modelling system | Horizontal grid | Vertical grid | Deposition scheme | Global meteo data provider | NOx emission share of NO and NO2 | Gaseous chemistry module |
|---|---|---|---|---|---|---|---|
| **EUROPEAN DOMAIN** | | | | | | | |
| Finnish Meteorological Institute | ECMWF-SILAM_H, SILAM_M | 0.25 x 0.25 deg Lat x Lon | 12 uneven layers up to 13km. First layer ~30m | Dry: Kouznetsov and Sofiev (2012) Wet: Kouznetsov and Sofiev (2014) | ECMWF (nudging within the PBL) | 90/10 | CBM-IV |
| Netherlands Organization for Applied Scientific Research | ECMWF-L.-EUROS | 0.5 x 0.25 deg Lat x Lon | Surface layer (~25m depth), mixing layer, 2 reservoir layers up to 3.5km. | Wet: below-cloud scavening Dry: Zhang et al. (2001) for particles, Depac (Zanten et al., 2012) for gases | Direct interpolation from ECMWF | 97/3 | CBM-IV |
| University of L'Aquila | WRF-WRF/Chem1 | 270x225 cells, 23 km | 33 levels up to 50hPa. 12 layers below 1km. First layer ~12m | Dry: Wesely (1989) Wet: Grell and Freitas (2014) | ECMWF (nudging above the PBL) | 95/5 | RACM-ESRL |
| University of Murcia | WRF-WRF/Chem2 | 270 x 225 cells,t 23 km x 23 km | 33 levels, from ~24m to 50hPa | Dry: Wesley resistance approach, (Wesley, 1989) Wet: Grid scale wet deposition (Easter et al, 2004) and convective wet deposition | ECMWF (nudging above the PBL) | 90/10 | RADM2 |
| Ricerca Sistema Energetico | WRF-CAMx | 265x220 cells, 23 km x 23 km | 14 layers up to 8km. First layer ~25m. | Dry: Resistance model for gases (Zhang et al., 2003) and aerosols (Zhang et al., 2001) Wet: Scavenging model for gases and aerosols (Seinfeld and Pandis, 1998) | ECMWF (nudging within the PBL) | 95/5 | CB05 |
| University of Aarhus | WRF-DEHM | 50 km x 50 km | 29 layers up to 100hPa | Wet and dry as in Simpson et al. (2003) | ECMWF (no nudging within the PBL) | 90/10 | Brandt et al. (2012) |
| Istanbul Technical University | WRF-CMAQ1 | 184 x 156 cells, 30 km x 30 km | 24 layers up to 10hPa | Wet and Dry as in Foley et al. (2010) | NCEP (nudging within PBL) | 95/5 | CB05 |
| Kings College | WRF-CMAQ4 | 15 km x 15 km | 23 layers up to 100hPa, 7 layer below 1km. First layer ~14m | Wet: Taken from the RADM (Chang et al., 1987) Dry: Electrical resistance analog model | NCEP (Nudging within the PBL) | 90/10 | CB05 |
| Ricardo E&E | WRF-CMAQ2 | 30 km x 30 km | 23 layers up to 100hPa, 7 layers below 1km. First layer ~15m | Wet: Byun and Schere (2006) Dry: Pleim and Ran (2011) | NCEP (nudging above the PBL) | Road transport: 86/14; non-road: 95/5 | CB05-TUCL |



| Helmholtz-Zentrum Geesthacht | CCLM-CMAQ | 24 km x 24 km | 30 vertical layers from ~40m to 50hPa | Wet: Byun and Schere (2006) Dry: Pleim and Ran (2011) | NCEP (spectral nudging above free troposhere) | 90/10 | CB05-TUCL |
|---|---|---|---|---|---|---|---|
| University of Hertfordshire | WRF-CMAQ3 | 18 km x 18 km | 35 vertical layers from ~20m to ~16km | Dry: resistance analogy model (Wesley, 1989). Wet: Asymmetric Convective model algorithm in CMAQ cloud module | ECMWF (nudging above PBL) | 90/10 | CB05-TUCL |
| INERIS/CIEMAT | ECMWF-Chimere_H Chimere_M | 0.25 x 0.25 deg Lat x Lon | 9 layers up to 500hPa. First layer ~20m | Wet: in-cloud and sub-cloud scavenging for gases and aerosols (Menut et al. 2013) Dry: resistance approach as Emberson (2000a,b) | Direct interpolation from ECMWF | 95% NO 4.5% NO$_2$ 0.5% HONO | MELCHIOR2 |
| **NORTH AMERICAN DOMAIN** | | | | | | | |
| Helmholtz-Zentrum Geesthacht | CCLM-CMAQ | 24 km x 24 km | 30 vertical layers from ~40m to 50hPa. | Wet: Byun and Schere (2006) Dry: Pleim and Ran (2011) | NCEP (spectral nudging above free troposhere) | 90/10 | CB05-TUCL |
| Environmental Protection Agency of the USA | WRF-CMAQ | 459x299 cells 12 km x 12 km | 35 layers, up to 50hPa. First layer ~19m | Wet: Byun and Schere (2006) Dry: Pleim and Ran (2011) | NCEP (nudging above the PBL) | 90/10 Calculated by MOVES for transport | CB05-TUCL |
| RAMBOLL Environ | WRF-CAMx | 459x299 cells, 12 Km x 12 km | 26 layers up to 97.5hPa | Dry: Resistance model for gases (Zhang et al., 2003) Wet: Scavenging model for gases and aerosols (Seinfeld and Pandis, 1998) | NCEP (nudging above the PBL) | 90/10 | CB05 |
| University of Aarhus | WRF-DEHM | 50 km x 50 km | 29 layers up to 100hPa | Wet and dry as in Simpson et al. (2003) | Direct interpolation from ECMWF | 90/10 | Brandt et al. (2012) |









**TABLE 2. EXTENSION OF THE SUB-REGIONS AND NUMBER OF RECEPTORS USED IN THE ANALYSIS**

|  | EU1/NA1 42–57.2N; -9–1.3W / 40–49.5; -83– -66W | EU2/NA2 47.5–56N; 1.3–18W / 30–38N; -91–-75W | EU3/NA3 43.5–46N; 7–14W / 33.5–43; -124–-118.5W | EU/NA 30–65N; -10–33W / 26–51N; -125–-55W |
|---|---|---|---|---|
| **Ozone** | 134/165 | 352/63 | 120/93 | 972/667 |
| **CO** | 32/29 | 91/8 | 70/12 | 418/103 |
| **NO (EU)** | 27 | 367 | 161 | 836 |
| **NO₂** | 149/97 | 529/21 | 176/54 | 1390/340 |
| **SO₂** | 96/69 | 296/3 | 55/3 | 865/141 |
| **PM₁₀ (EU)** | 47 | 347 | 2 | 619 |
| **PM₂.₅ (NA)** | 89 | 9 | 22 | 226 |
| **WS** | 168/229 | 305/245 | 5/59 | 827/1721 |
| **Temp** | 168/232 | 305/243 | 5/46 | 830/1546 |


**TABLE 3. SUMMARY OF OZONDESONDES DATA FOR OZONE**

| EU | | | |
|---|---|---|---|
| **Station** | **O₃ Records** | **Period** | **Local time** |
| 316 | 52 | Year(4-5 launches per month) | 11-12 |
| 308 | 52 | Year(4-5 launches per month) | 10-11 |
| 318 | 37 | Year(3-4 launches per month, mostly winter and autumn) | 11-12 |
| 242 | 46 | January-April(10-12 launches per month) | 11-12 |
| 156 | 144 | Year(12 launches per month) | 10-12 |
| 099 | 66 | Year(5-6 launches per month) | Mostly early morning 4-6 |
| 053 | 149 | Year(11-13 launches per month) | 11-12 |
| 043 | 51 | Year(4-5 launches per month) | 11-12 |
| **NA** | | | |
| 021 | 44 | Year(3-4 launches per month) | 11-12 |
| 107 | 54 | Year(4-5 launches per month) | 16-20 |
| 338 | 50 | Year(2-4 per month; 17 in July; none in September) | 14-15 July-August 17-18 other months |
| 456 | 57 | 2-5 per month; 25 in July | 17-18 |
| 457 | 75 | Year(2-5 per month; 18-20 in May-June) | 23-00 |
| 458 | 71 | Year(3-8 per month; 20 in July) | 23-00 |





## FIGURES

Figure 1. Sub-regions of the two continental domains ( a) EU; b) NA ). Overlaid are the ozone monitoring stations classified based on the network

Figure 2. RMSE for a) Temp and b) WS in Europe

**FIGURE 3** RMSE for a) Temp and b) WS in North America

Figure 4. Mean Bias (mod − obs) for the vertical profiles of Wind Speed measured by ozonesondes launched from the European locations indicated on the inset map of each panel. The number of hourly profiles available for each site is reported in the parenthesis at the top of each panel

Figure 5. Mean Bias (mod − obs) for the vertical profiles of Temperature measured by ozonesondes launched from the European locations indicated on the inset map of each panel. The number of hourly profiles available for each site is reported in the parenthesis at the top of each panel

Figure 6. Mean Bias (mod − obs) for the vertical profiles of Wind Speed measured by ozonesondes launched from the North American locations indicated on the inset map of each panel. The number of hourly profiles available for each site is reported in the parenthesis at the top of each panel

Figure 7. Mean Bias (mod − obs) for the vertical profiles of Temperature measured by ozonesondes launched from the North American locations indicated on the inset map of each panel. The number of hourly profiles available for each site is reported in the parenthesis at the top of each panel

Figure 8. Cumulated modelled deposition per unit area over the continental regions of a) EU and b) NA for the full year of 2010. The boxes extend between the minimum and the $5^{th}$ percentile, while the maximum is reported by the number at the top of each box. results are displayed for the models and species for which data have been made available

Figure 9. RMSE (ppb) for CO by spectral component and season (panel *a* for Europe and *b* for North America). FT is the full (un-filtered) time series, LT, SY, DU, are the Long Term, Synoptic and diurnal components, respectively.

Figure 10. MSE ($ppb^2$) breakdown into bias squared, variance and *mMSE* for the spectral components of the spatial average time series of CO during the months of December, January, and February (DJF), based on EQ.6. The bias is entirely accounted for by the LT component. The signs within the bias and variance portion of the bars indicate model overestimation (+) or underestimation (-) of the bias and variance. The colour of the *mMSE* share of the error is coded based on the values of *r*, the correlation coefficient, according to the colour scale at the bottom of each plot. Top panel: EU; lower panel: NA. Similar plots for the other two sub-regions are reported in the supplementary material.

Figure 11. RMSE variation between the 's20%' scenario (anthropogenic emission and boundary condition reduced by 20%) and the base case for CO in EU2

Figure 12. Top panel: as in Figure 9 for NO (EU only). Lower panel: as in Figure 10 for NO (EU only)

Figure 13. RMSE variation between the 's20%' scenario (anthropogenic emission and boundary condition reduced by 20%) and the base case for anthropogenic NO (aNO) in eu2

Figure 14. As in Figure 9 for $NO_2$.

Figure 15. As in Figure 10 for $NO_2$ in EU2. Upper panel: Urban sites only (223 stations); lower panel: Rural sites only (159 stations)

Figure 16. As in Figure 10 for $NO_2$ in NA1. Upper panel: Urban sites only (39 stations); Lower panel: Rural sites only (10 stations).

Figure 17. As in Figure 9 for ozone

Figure 18. As in Figure 10 for ozone during the months from May to September



Figure 19. Ozone mixing-ratio profiles measured by ozonesondes launched from the European location indicated on the
inset map (lower-right corner) of each panel. The profiles are time-averaged over the number of hourly records reported in
the parenthesis at the top of each panel. Legend as in the first panel.
Figure 20. As in Figure 19 for North America
Figure 21. Ozone vs NO modelled mean bias for the EU2 sub-region, color-coded by temperature bias and symbols
according to the $NO_x$ emission fraction of NO and $NO_2$. Each point represents a model. *a)* winter months and *b)* summer
months.
Figure 22. As in Figure 9 for $SO_2$
Figure 23. As in Figure 10 for $SO_2$
Figure 24. As in Figure 9 for $PM_{10}$ in Europe (error units in $\mu g/m^3$)
Figure 25. As in Figure 9 for $PM_{2.5}$ in North America (error units in $\mu g/m^3$)
Figure 26. As in Figure 10 for $PM_{10}$ in Europe (error units in $\mu g/m^3$)
Figure 27. As in Figure 10 for $PM_{2.5}$ in North America (error units in $\mu g/m^3$)
Figure 28. Spatial map of the ozone monitoring stations coloured based on the 'delta hour' values, i.e. the difference in
hours between the zero of the autocorrelation function (acf) for the modelled ozone minus the zero of the acf of the
observed one. The acf is calculated on the long term component for the months of May to September. Negative values
indicate too short memory and excess of removal (vice-versa for positive values). The box on the right summarises the
delta hour percentile distribution.
Figure 29. As in Figure 28 for North America.






FIGURES

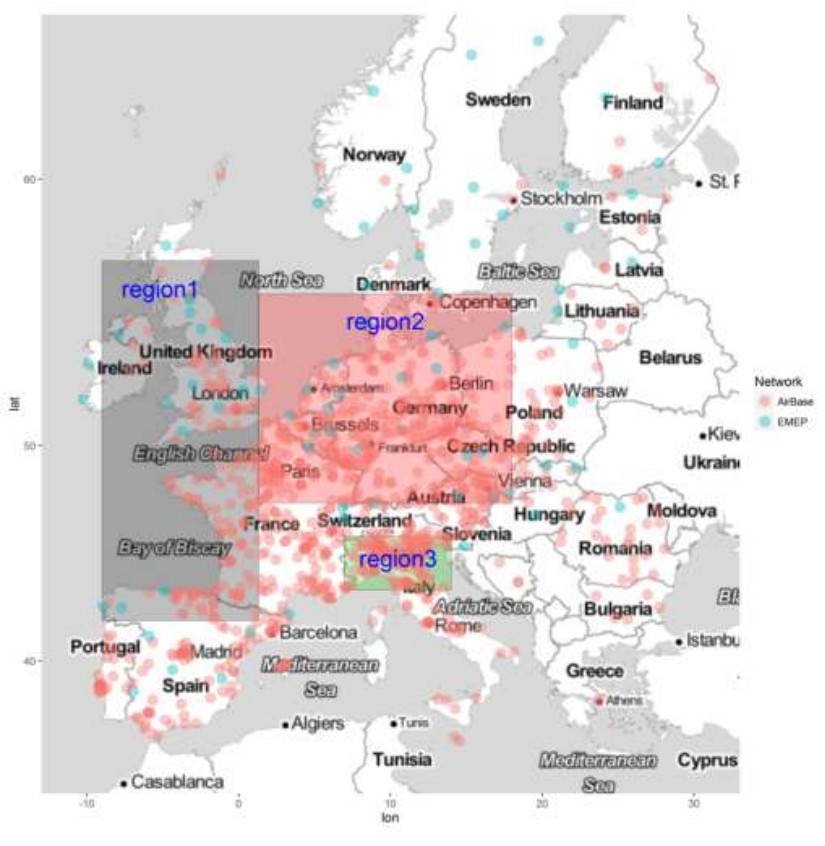

a)

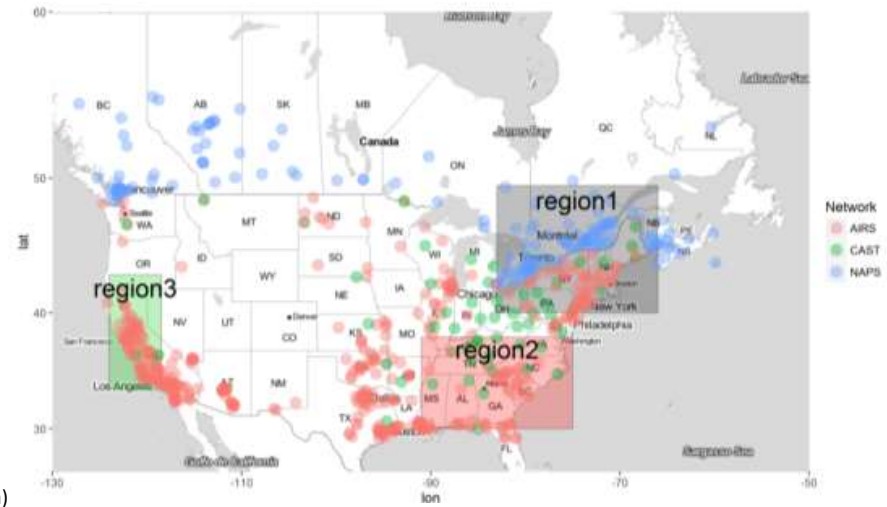

a)                                                                              b)

**FIGURE 1.** Sub-regions of the two continental domains ( a) EU; b) NA ). Overlaid are the ozone monitoring stations classified based on the network



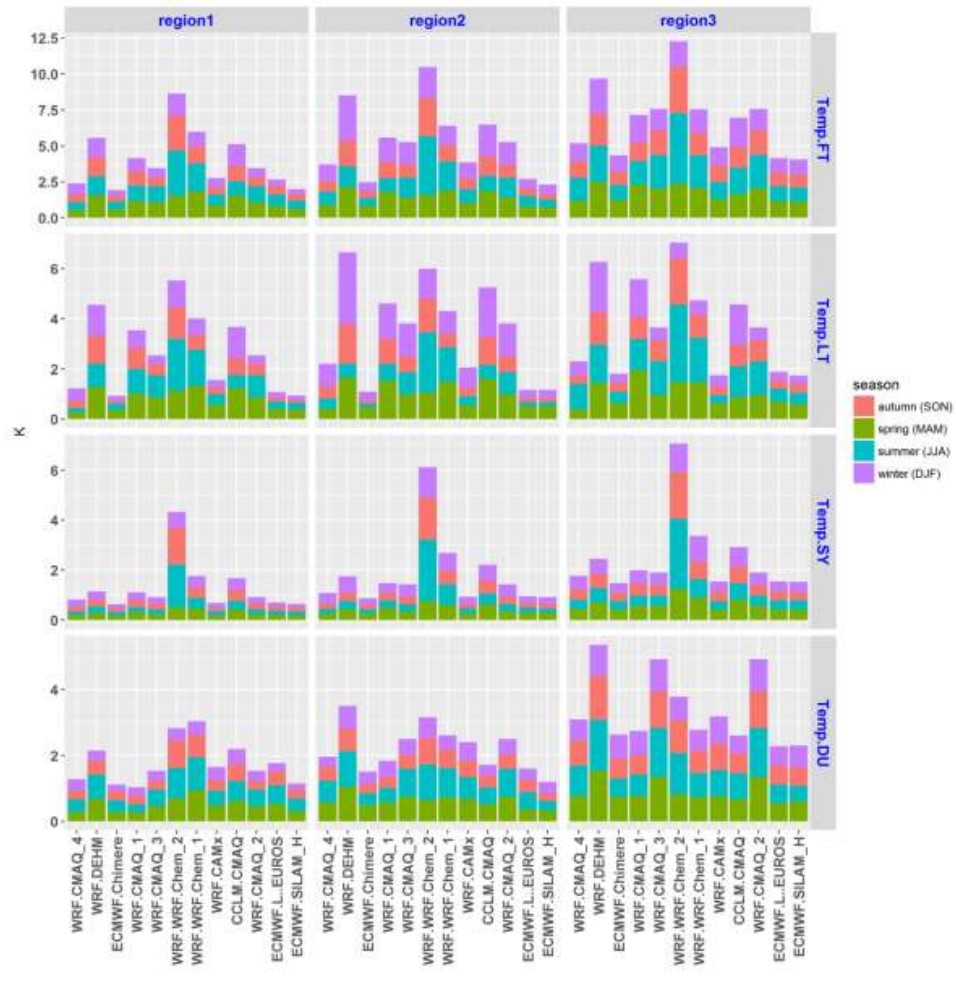

a)



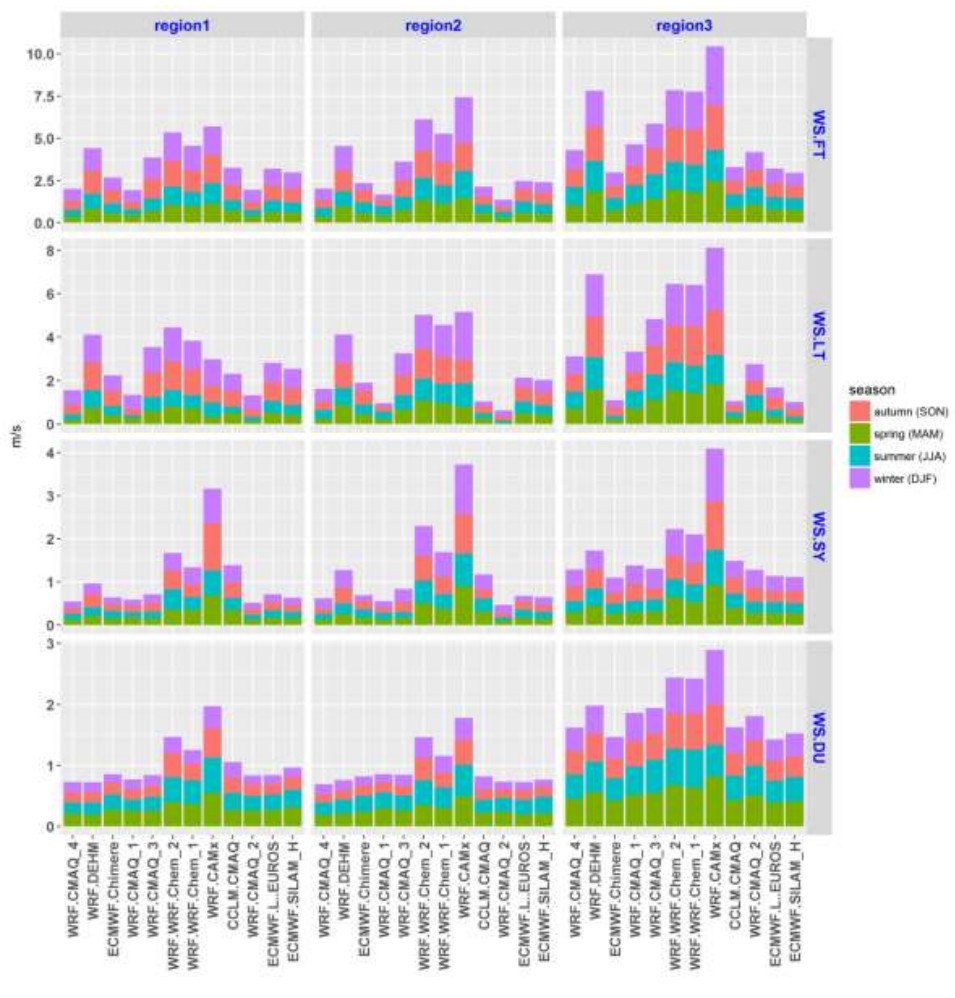

b)

**FIGURE 2. RMSE FOR A) TEMP AND B) WS IN EUROPE**



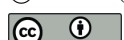

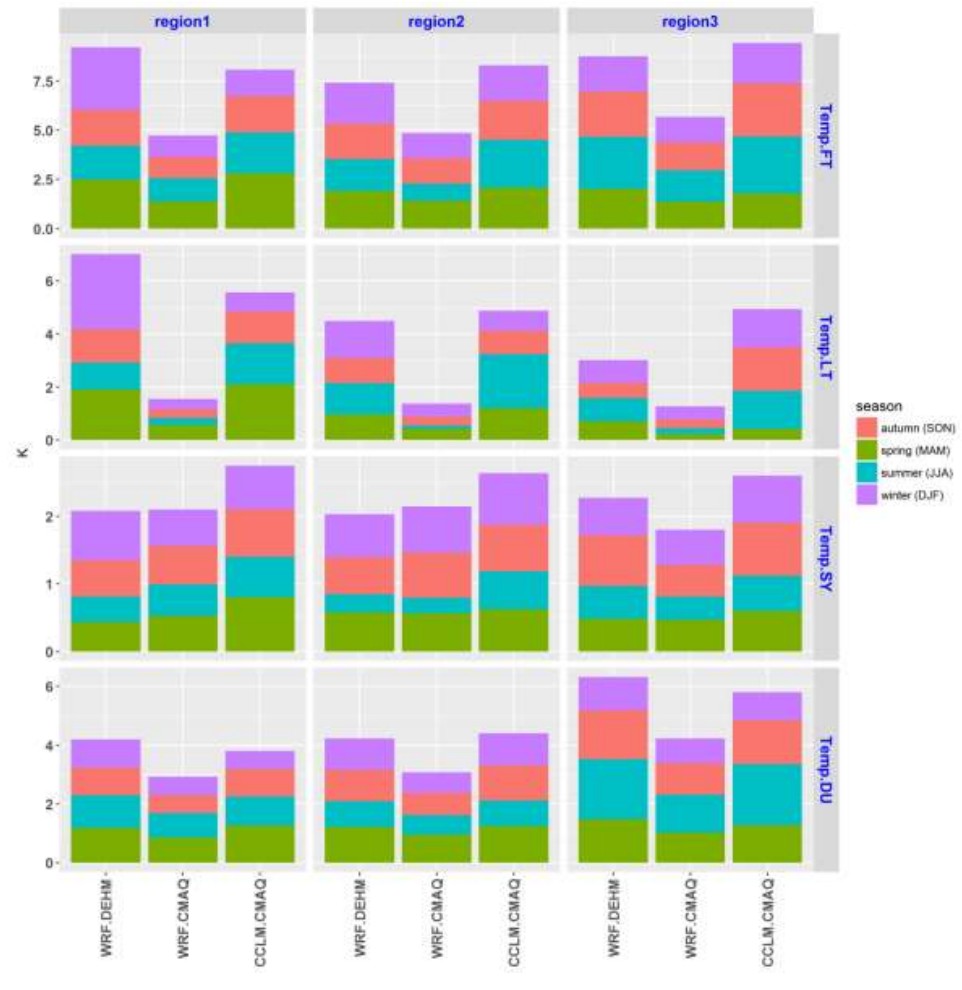

a)



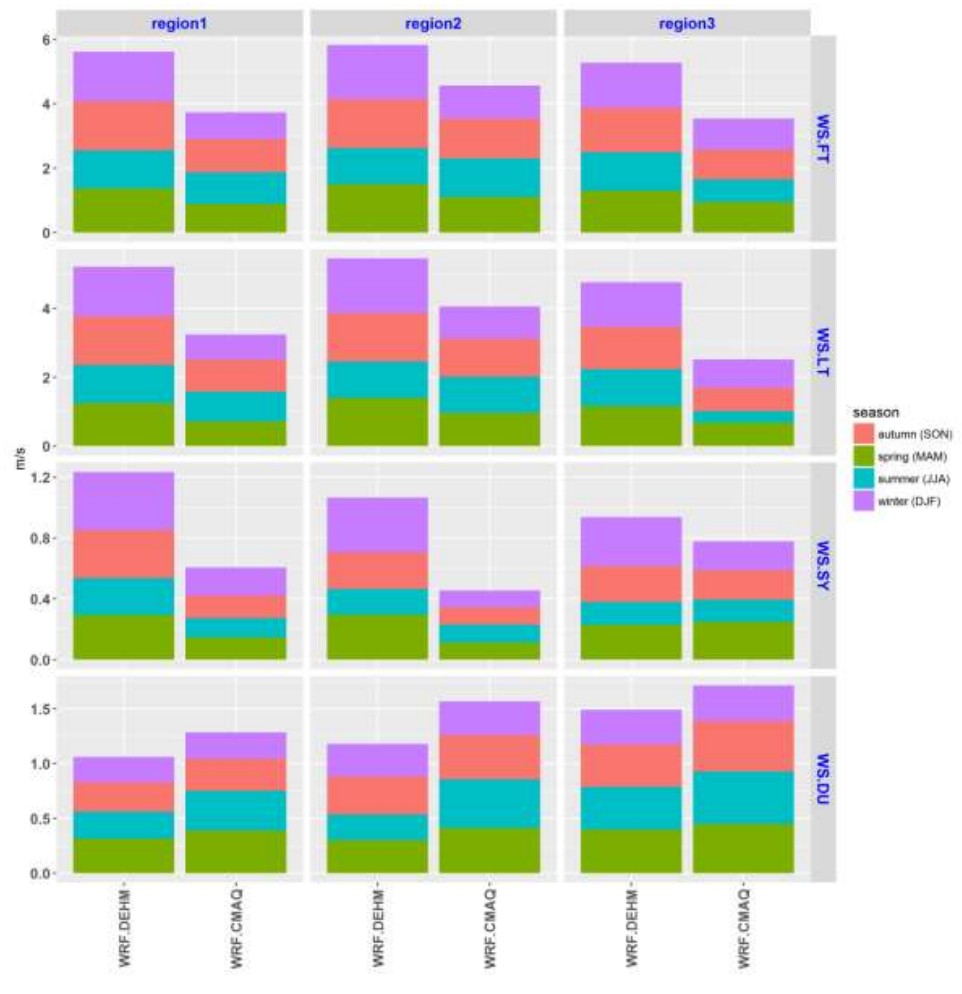

b)

**FIGURE 3. RMSE FOR A) TEMP AND B) WS IN NORTH AMERICA**



**Figure 4.** Mean Bias (mod – obs) for the vertical profiles of wind speed measured by ozonesondes launched from the European location indicated on the inset map of each panel. The number of hourly profiles available for each site is reported in the parenthesis at the top of each panel









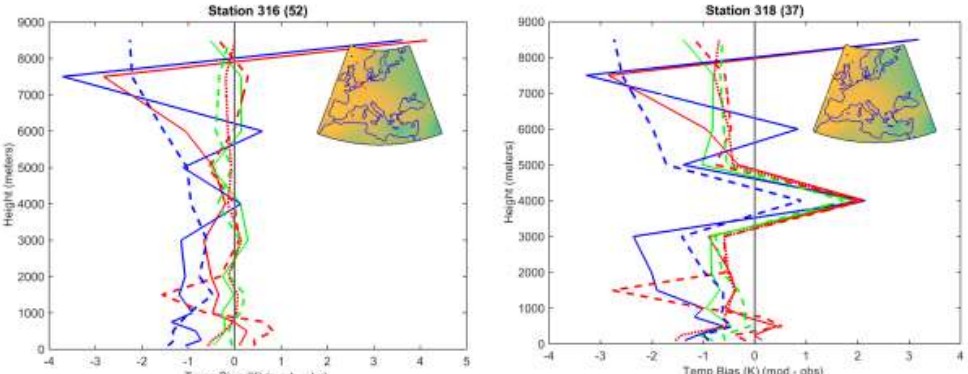

**Figure 5.** Mean Bias (mod – obs) for the vertical profiles of temperature measured by ozonesondes launched from the European location indicated on the inset map of each panel. The number of hourly profiles available for each site is reported in the parenthesis at the top of each panel

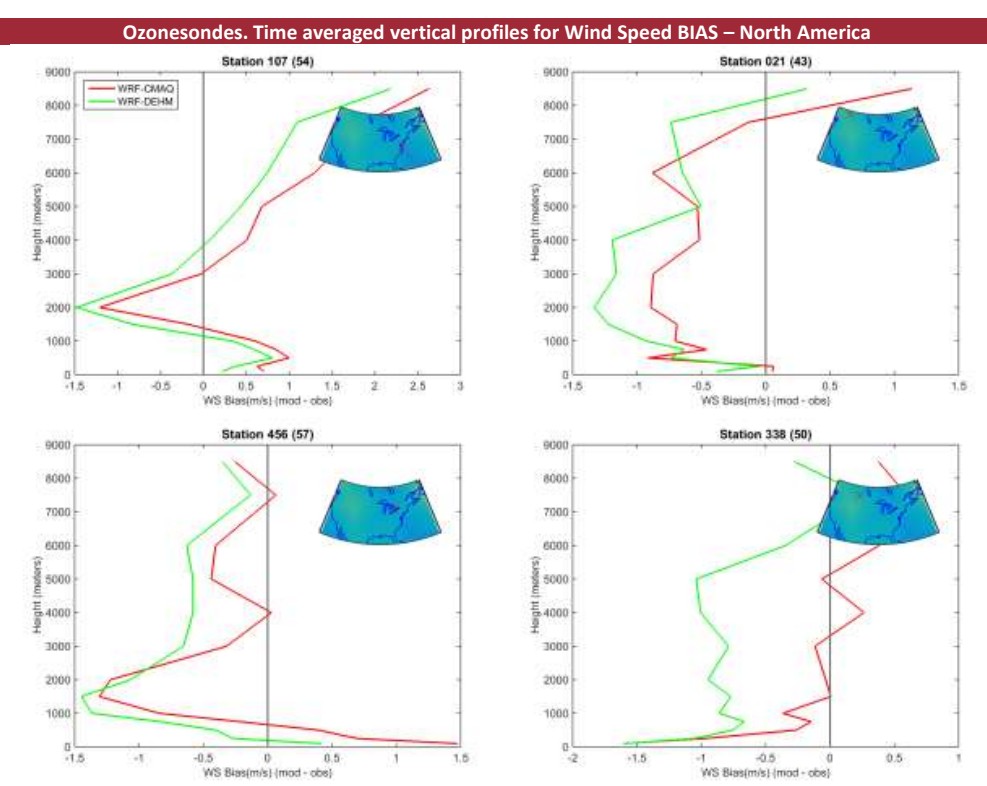




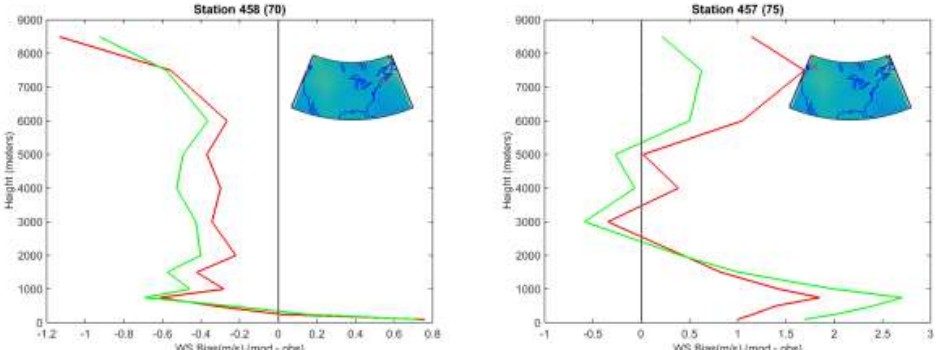

**FIGURE 6.** Mean Bias (mod − obs) for the vertical profiles of wind speed measured by ozonesondes launched from the North American locations indicated on the inset map of each panel. The number of hourly profiles available for each site is reported in the parenthesis at the top of each panel

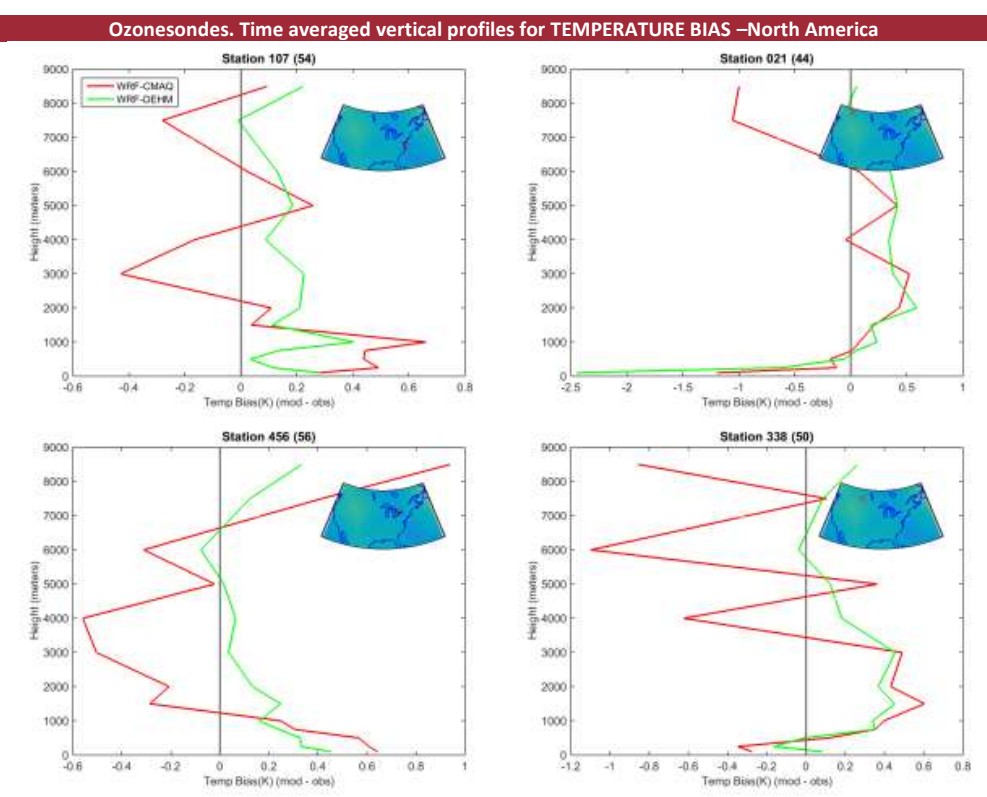



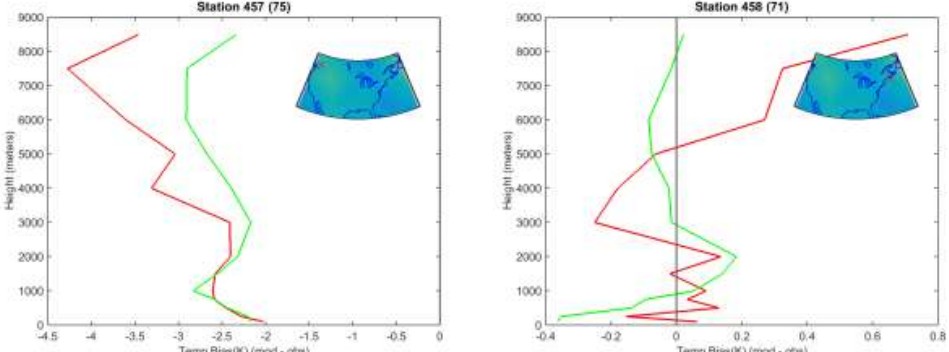

**FIGURE 7.** Mean Bias (mod − obs) for the vertical profiles of Temperature measured by ozonesondes launched from the North American location indicated on the inset map of each panel. The number of hourly profiles available for each site is reported in the parenthesis at the top of each panel

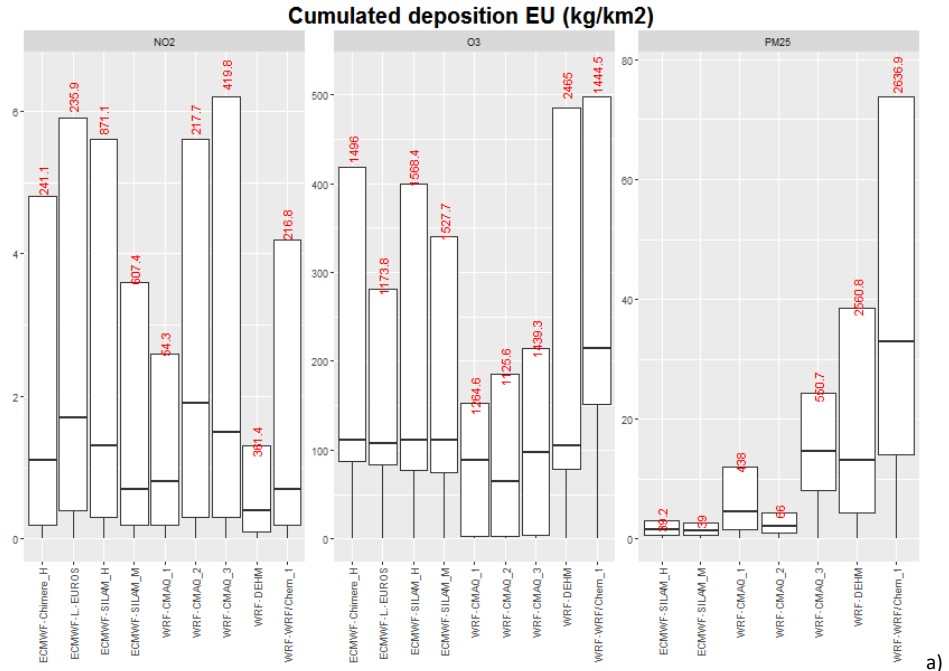

a)





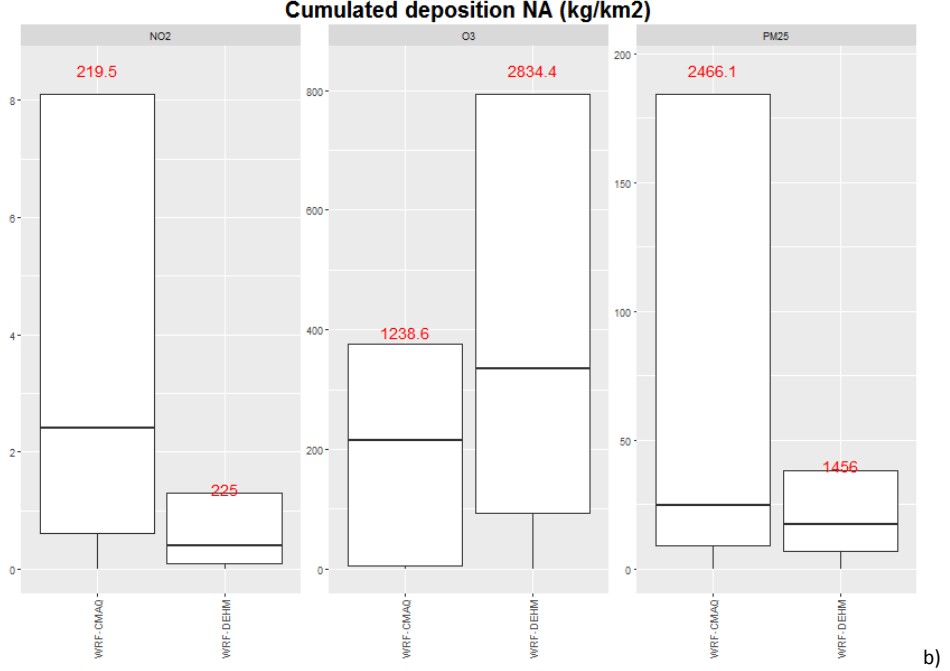

**Figure 8**. Cumulated modelled deposition per unit area over the continental regions of a) EU and b) NA for the full year of 2010. The boxes extend between the minimum and the 5[th] percentile, while the maximum is reported by the number at the top of each box. results are displayed for the models and species for which data have been made available.





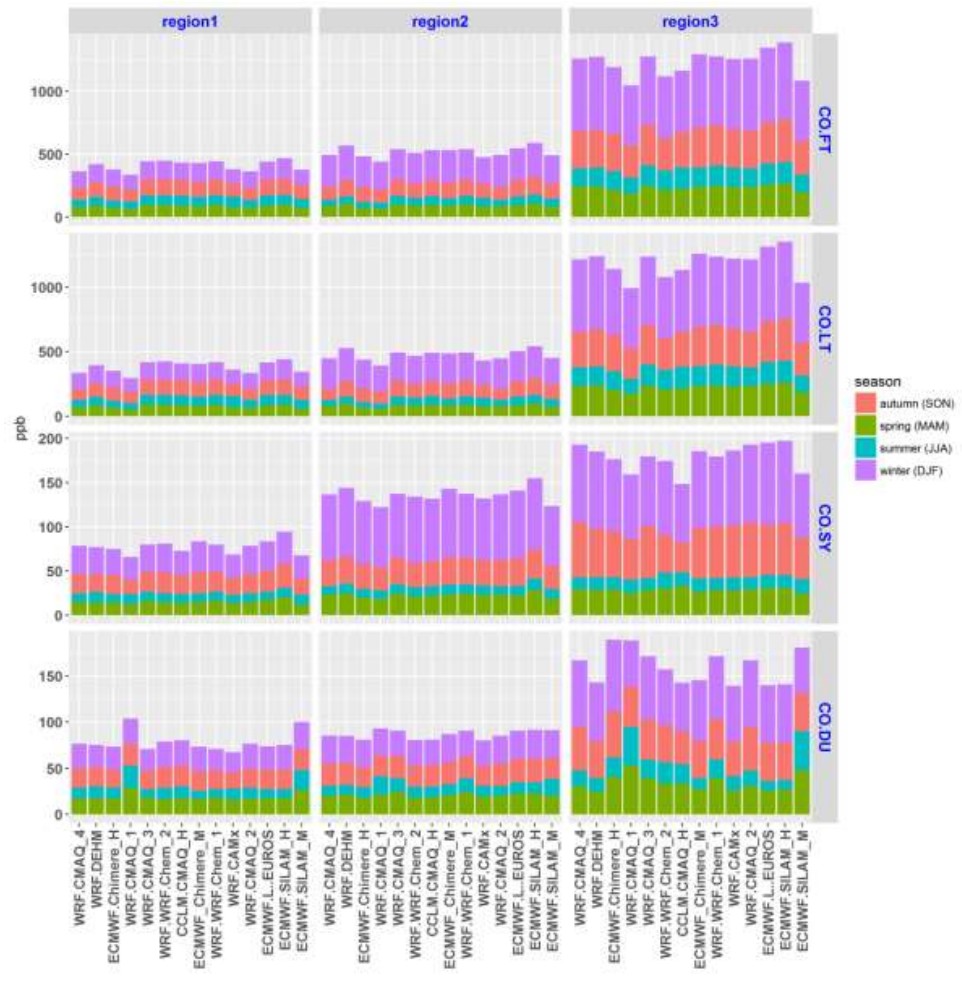

a)




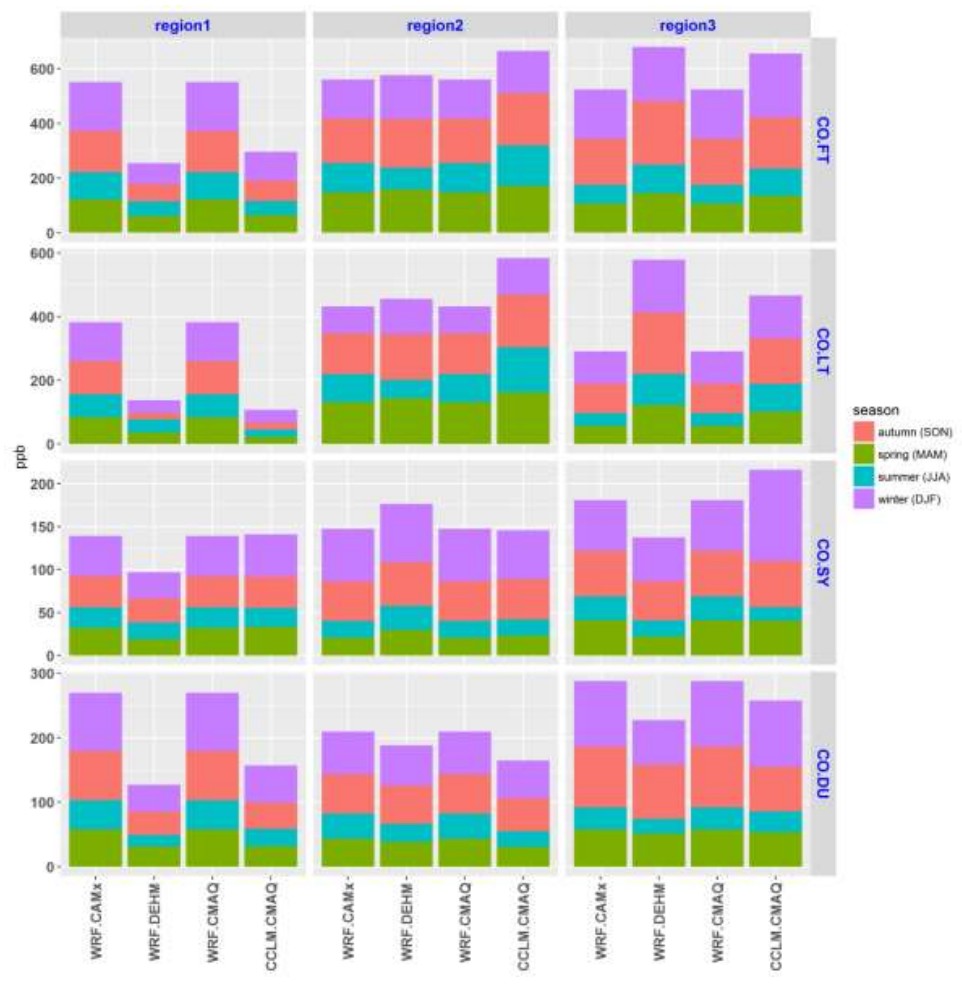

b)

**FIGURE 9**. RMSE (ppb) for CO by spectral component and season (panel *a* for Europe and *b* for North America). FT is the full (un-filtered) time series, LT, SY, DU, are the Long Term, Synoptic and diurnal components, respectively.



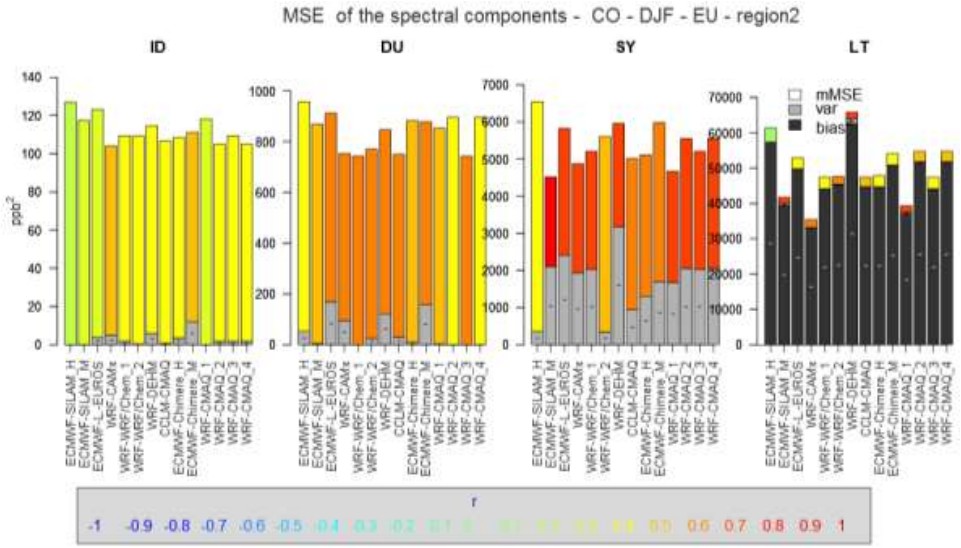

a)

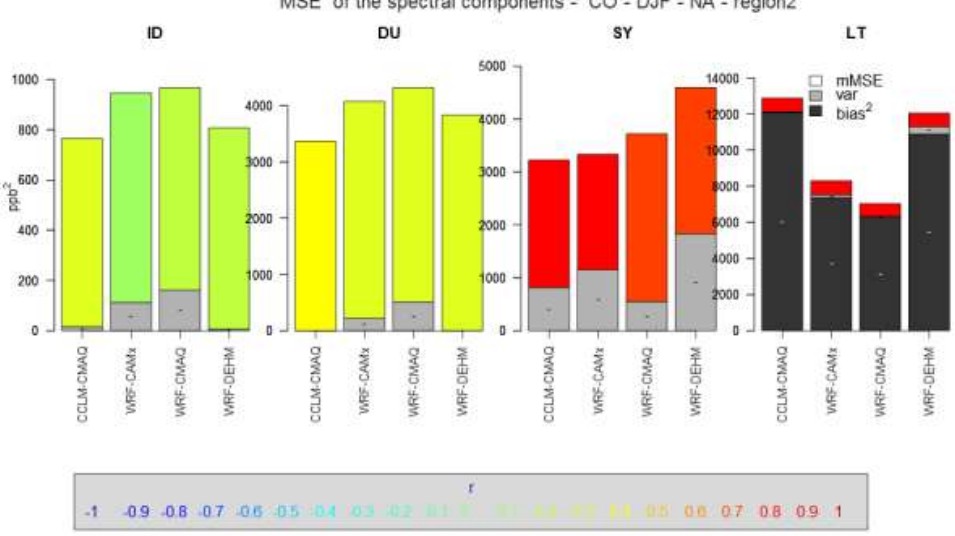

b)

**FIGURE 10.** MSE (ppb$^2$) breakdown into bias squared, variance and *mMSE* for the spectral components of the spatial average time series of CO during the months of December, January, and February (DJF), based on EQ.6. The bias is entirely accounted for by the LT component. The signs within the bias and variance portion of the bars indicate model overestimation (+) or underestimation (-) of the bias and variance. The colour of the *mMSE* share of the error is coded based on the values of *r*, the correlation coefficient, according to the colour scale at the bottom of each plot. Top panel: EU; lower panel: NA. Similar plots for the other two sub-regions are reported in the supplementary material.





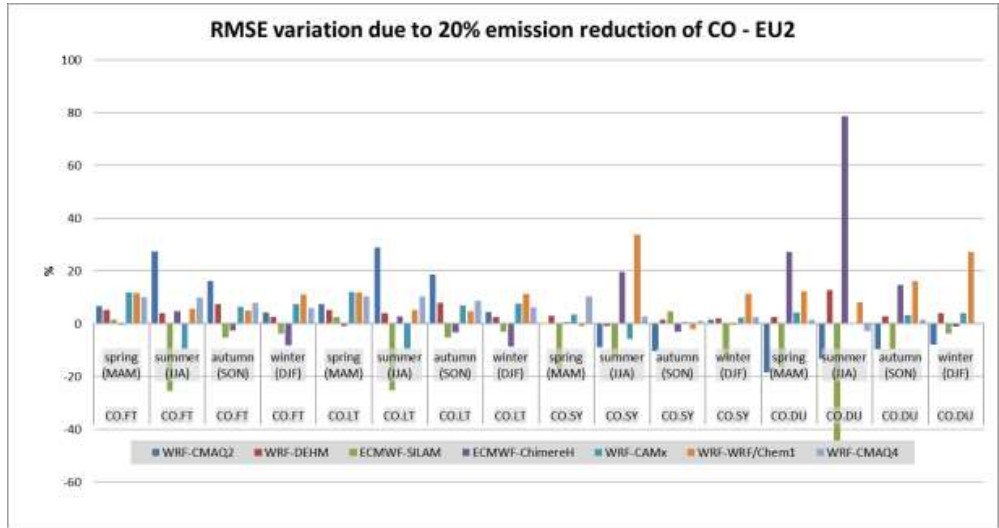

**FIGURE 11.** RMSE variation between the 's20%' scenario (anthropogenic emission and boundary condition reduced by 20%) and the base case for CO in EU2





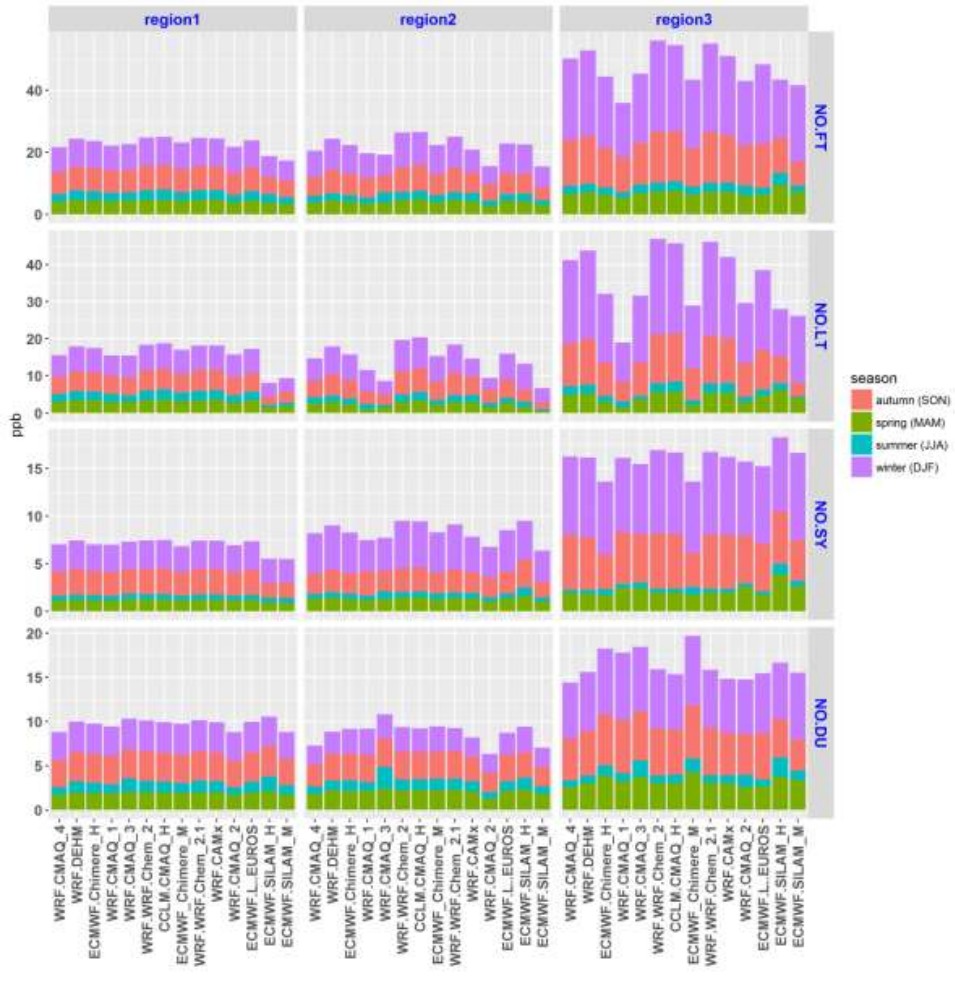

a)




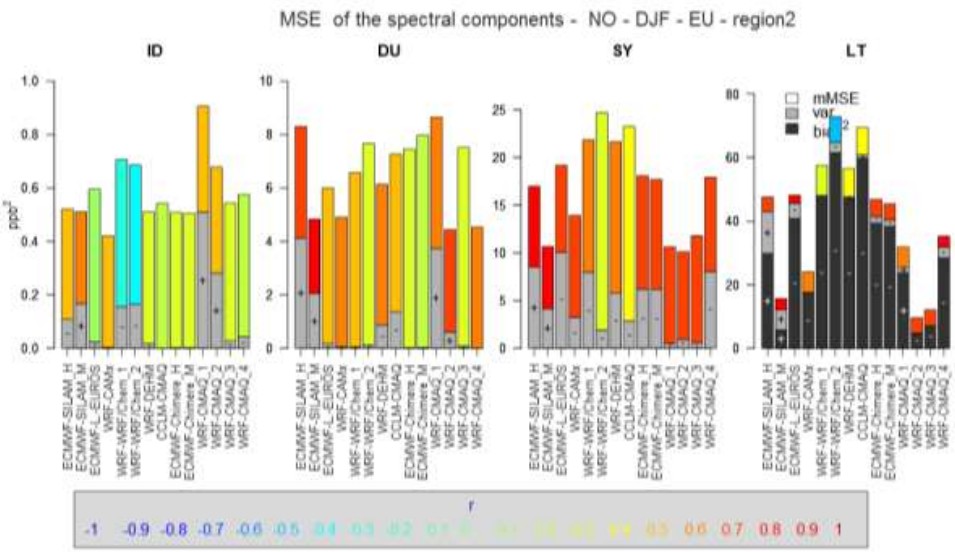

b)

**FIGURE 12.** Top panel: as in **FIGURE 9** for NO (EU only). Lower panel: as in **FIGURE 10** for NO (EU only)

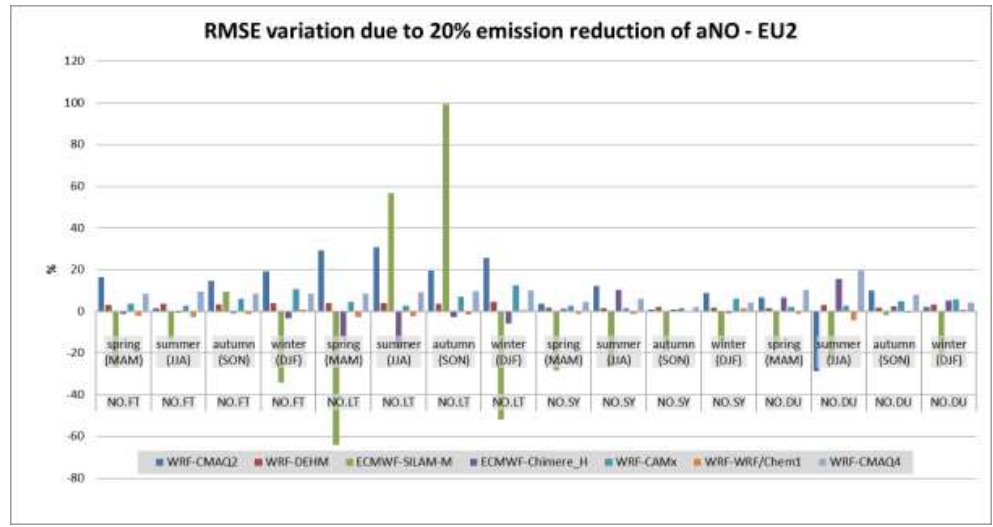

**FIGURE 13.** RMSE variation between the 's20%' scenario (anthropogenic emission and boundary condition reduced by 20%) and the base case for anthropogenic NO (aNO) in EU2





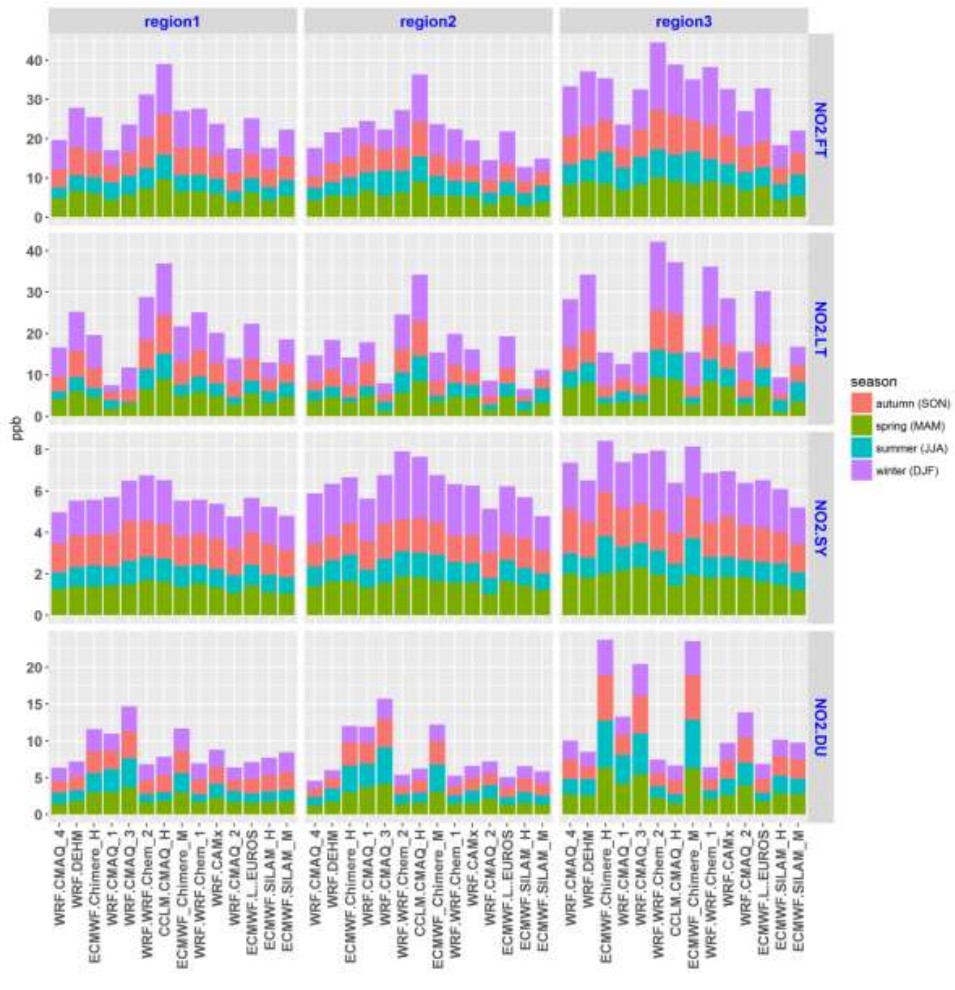

a)



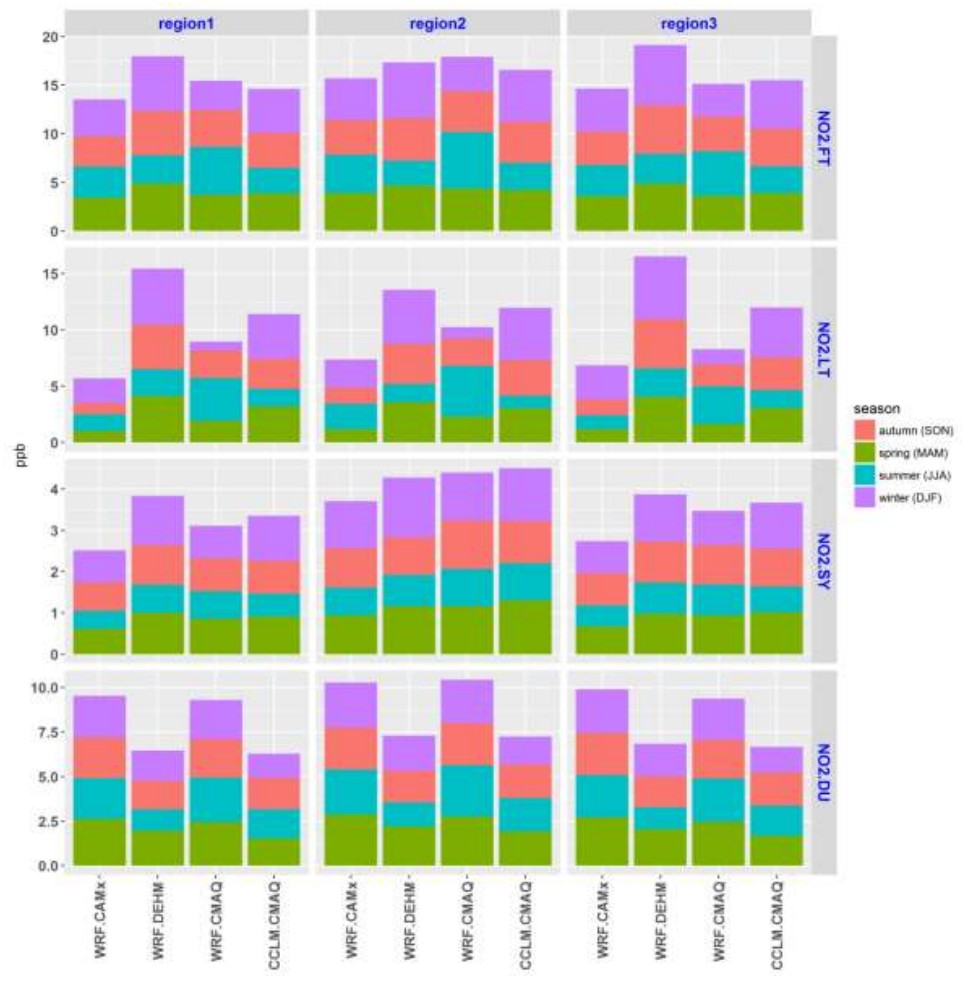

b)

**FIGURE 14.** As in **FIGURE 9** for $NO_2$





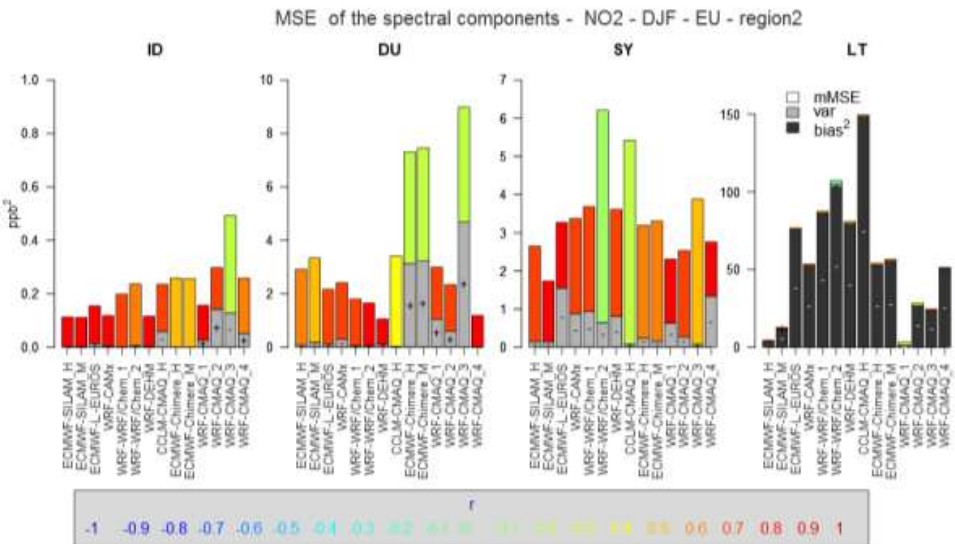

a) Urban NO$_2$ in EU2 sub-region (223 stations)

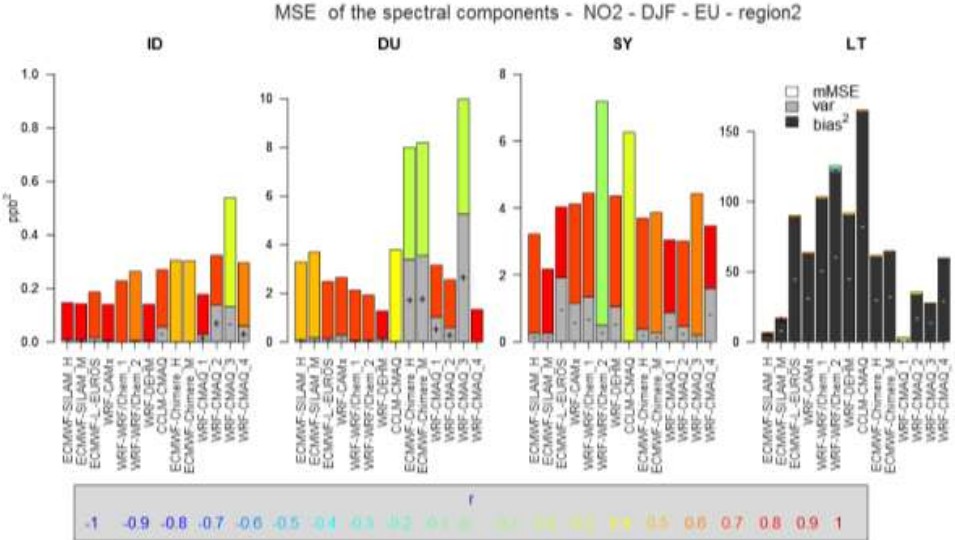

b) Rural NO$_2$ in EU2 sub-region (159 stations)

FIGURE 15. AS IN FIGURE 10 FOR NO$_2$ IN EU2. UPPER PANEL: URBAN SITES ONLY (223 STATIONS); LOWER PANEL: RURAL SITES ONLY (159 STATIONS)



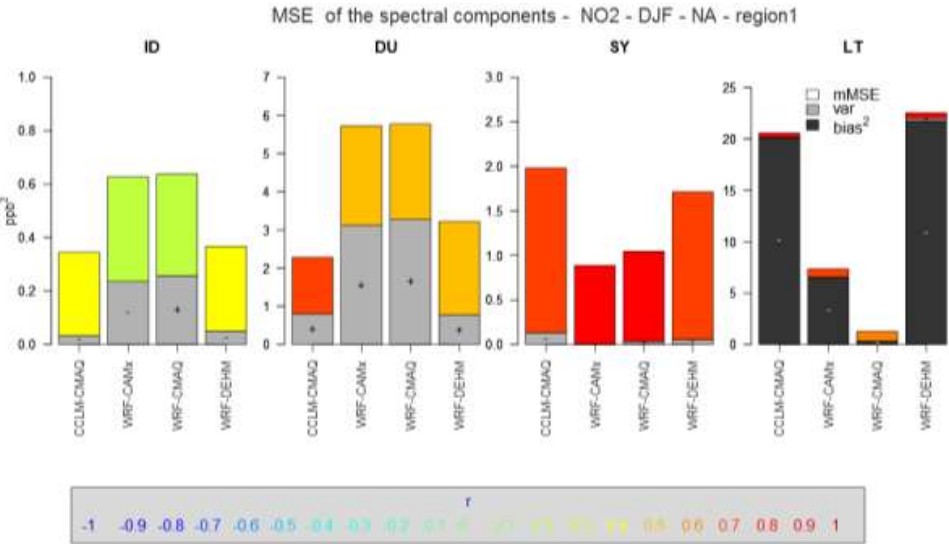

a) NA1 urban (39 stations)

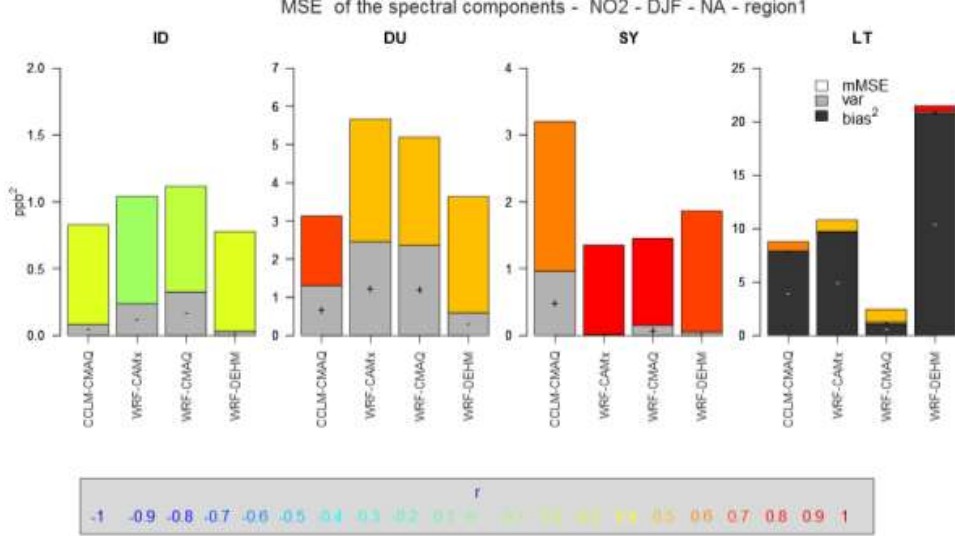

b) NA1 rural (10 stations)

**FIGURE 16. AS IN FIGURE 10 FOR NO₂ IN NA1. UPPER PANEL: URBAN SITES ONLY (39 STATIONS); LOWER PANEL: RURAL SITES ONLY (10 STATIONS).**





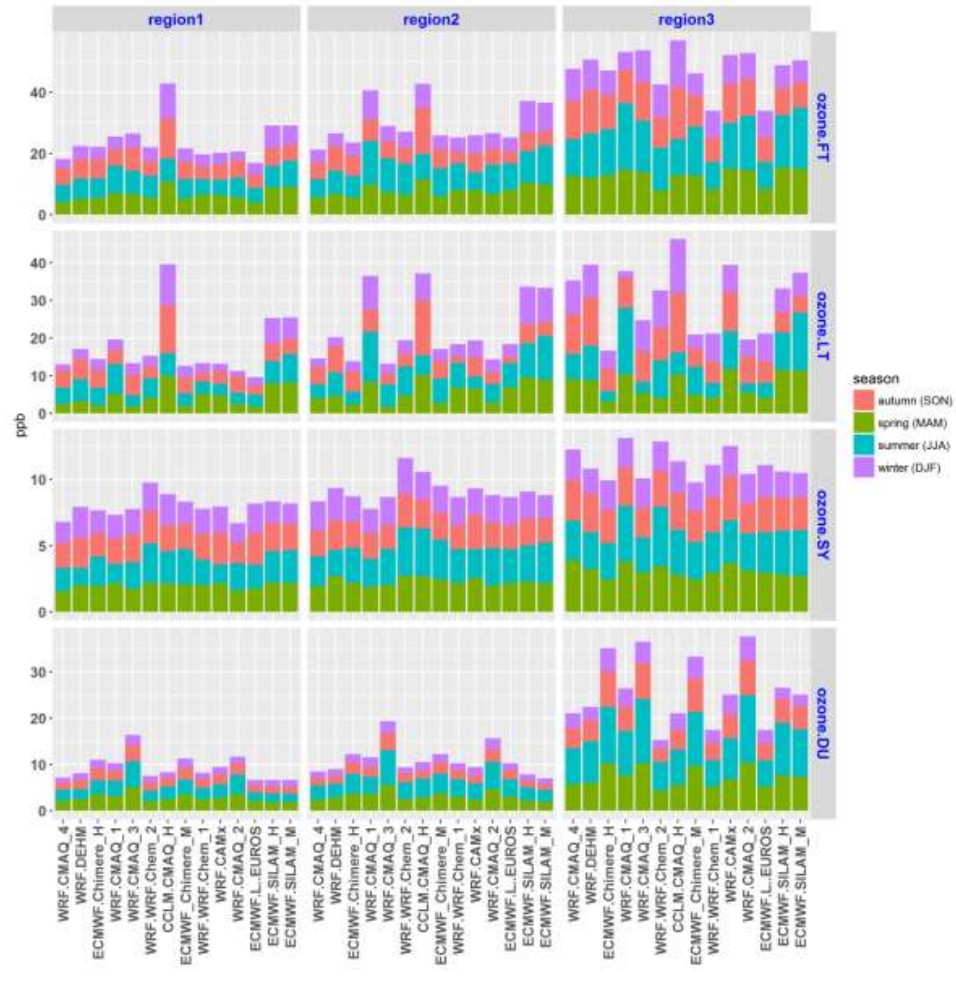

A)





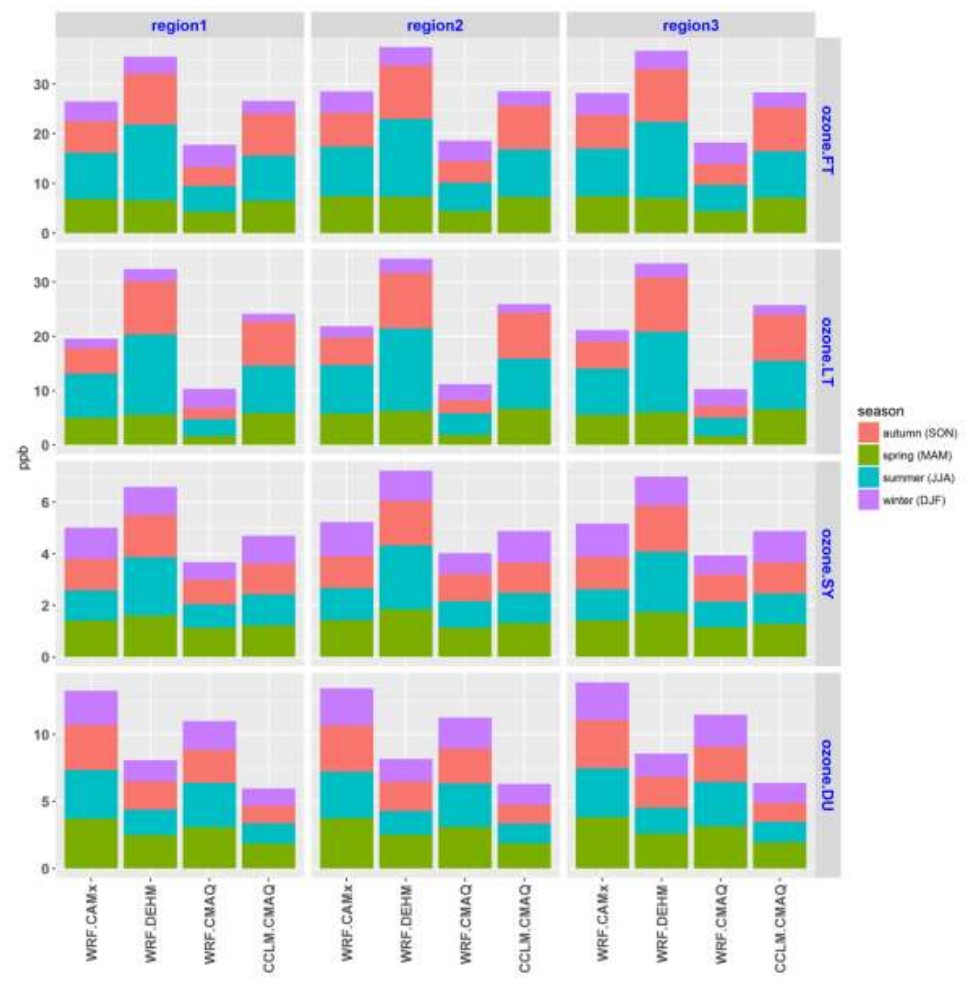

B)

**FIGURE 17. AS IN FIGURE 9 FOR OZONE**



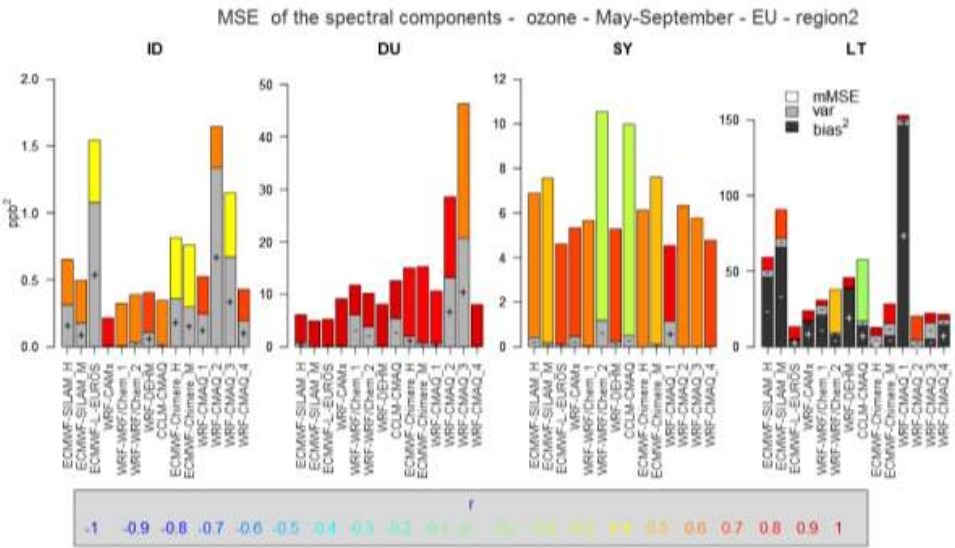

a)

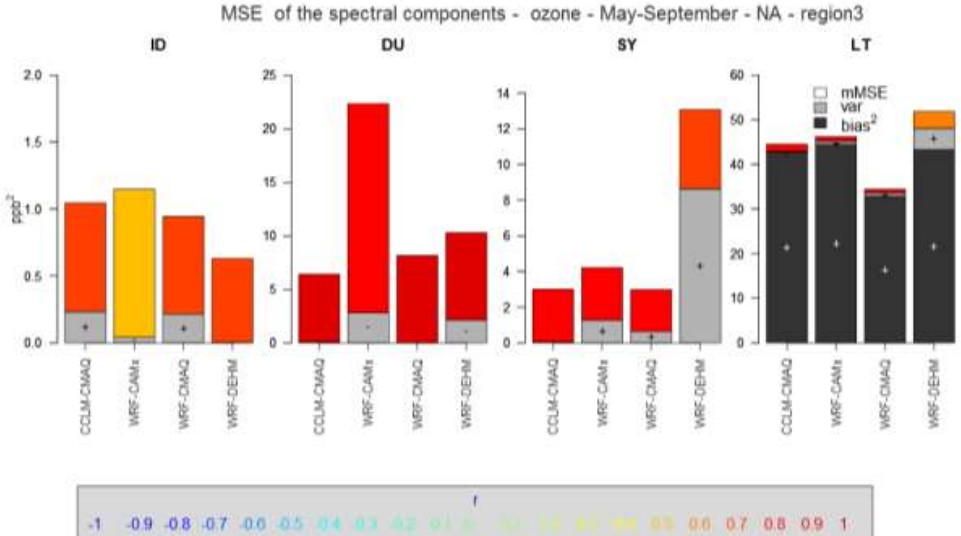

b)

FIGURE 18. AS IN FIGURE 10 FOR OZONE DURING THE MONTHS FROM MAY TO SEPTEMBER





**Figure 19**. Ozone mixing-ratio profiles measured by ozonesondes launched from the European location indicated on the inset map (lower-right corner) of each panel. The profiles are time-averaged over the number of hourly records reported in the parenthesis at the top of each panel. Legend as in the first panel.

**FIGURE 20 .** As in **FIGURE** 19 for North America



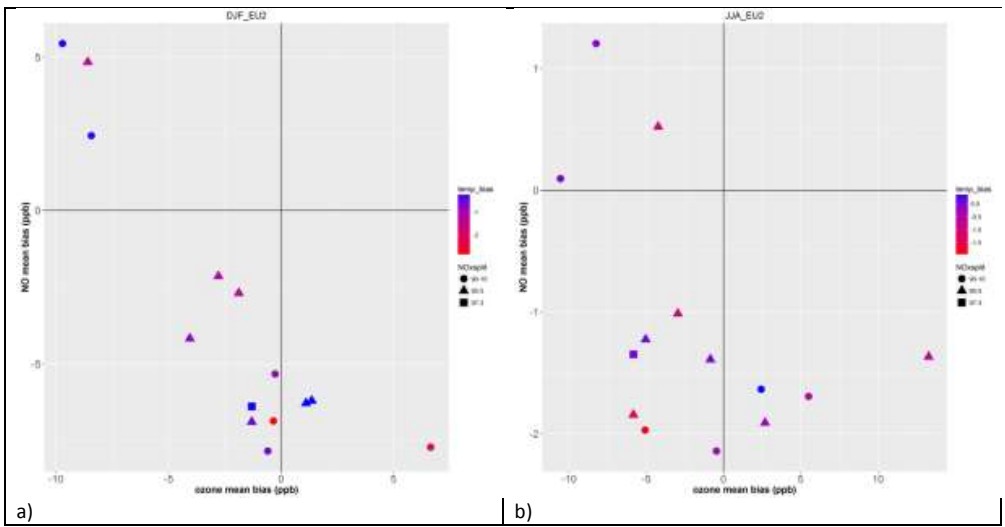

**FIGURE 21.** Ozone vs NO modelled mean bias for the EU2 sub-region, color-coded by temperature bias and symbols according to the NO$_x$ emission fraction of NO and NO$_2$. Each point represents a model. *a)* winter months and *b)* summer months.





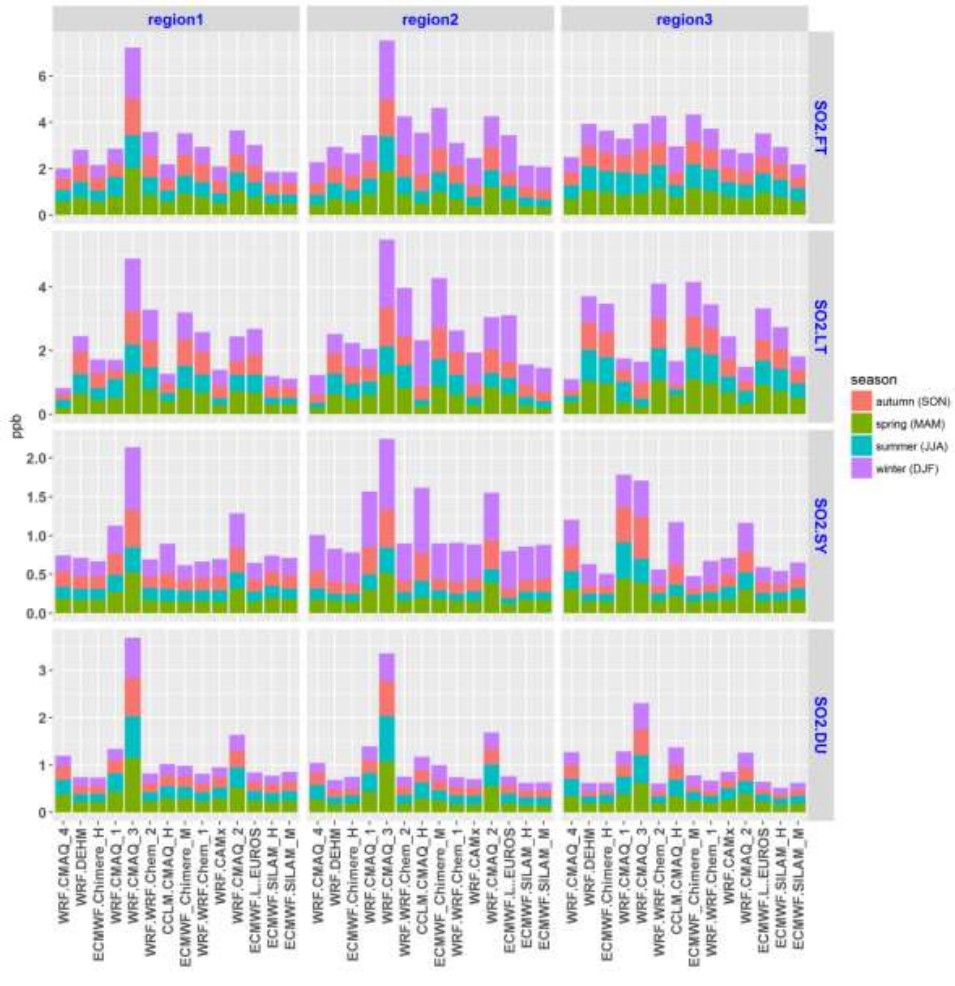

a)



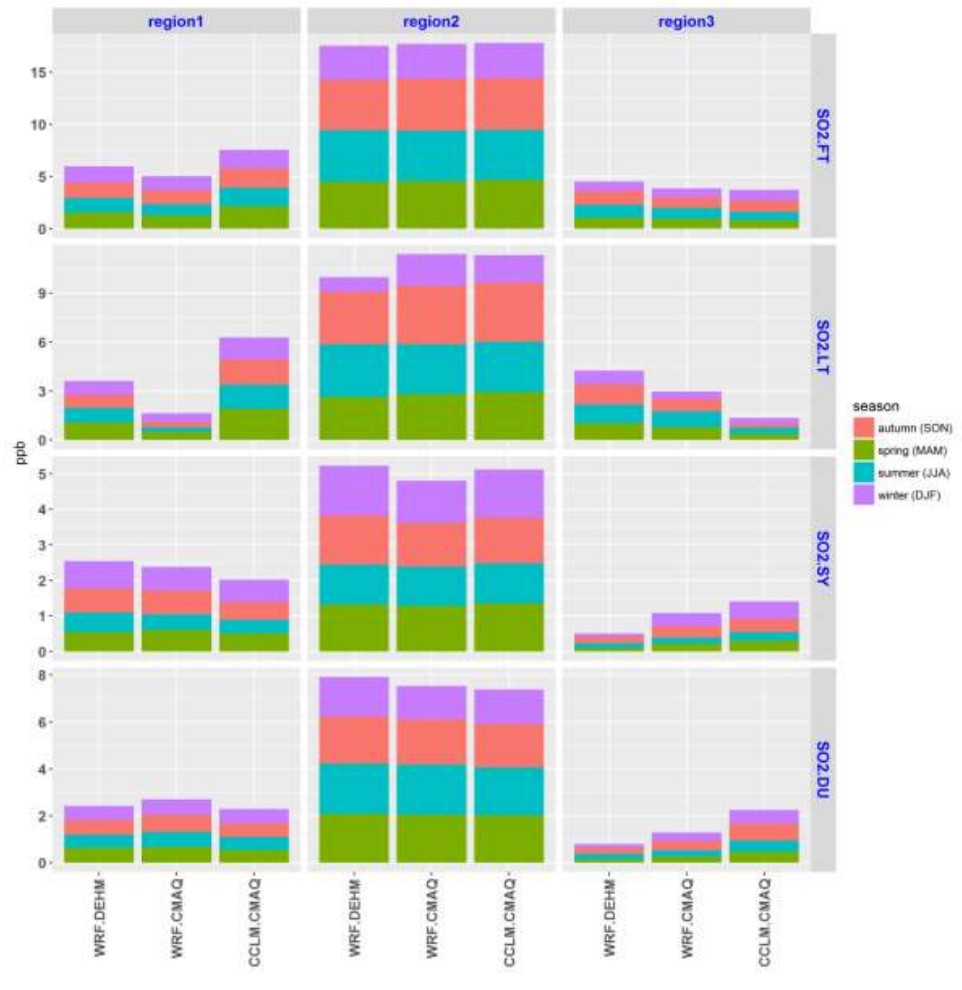

b)

**FIGURE 22. AS IN FIGURE 9 FOR SO$_2$**





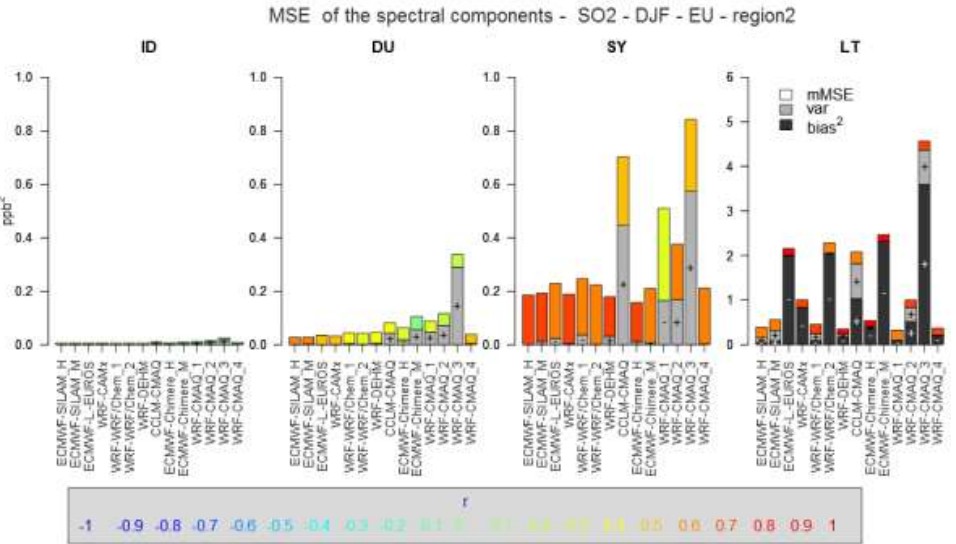

a)

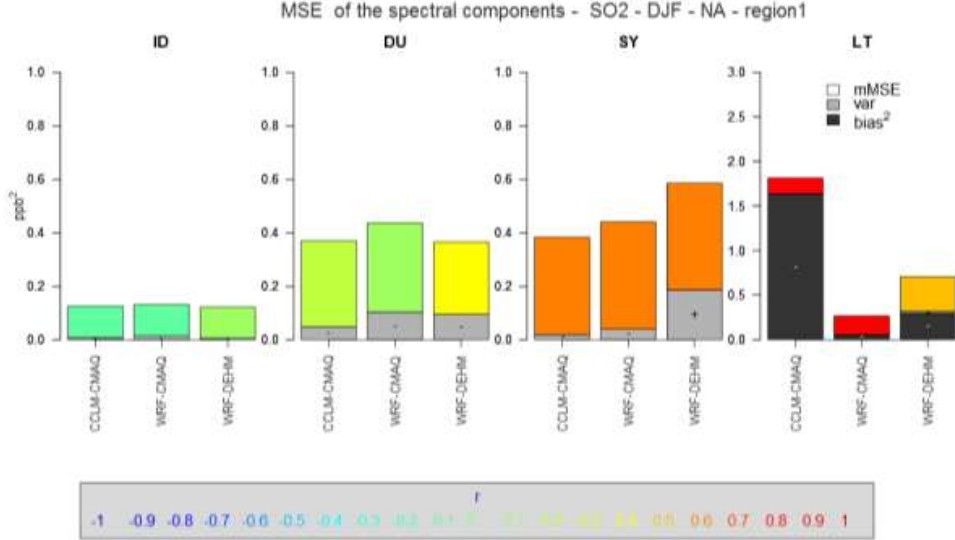

b)

FIGURE 23. AS IN FIGURE 10 FOR SO$_2$





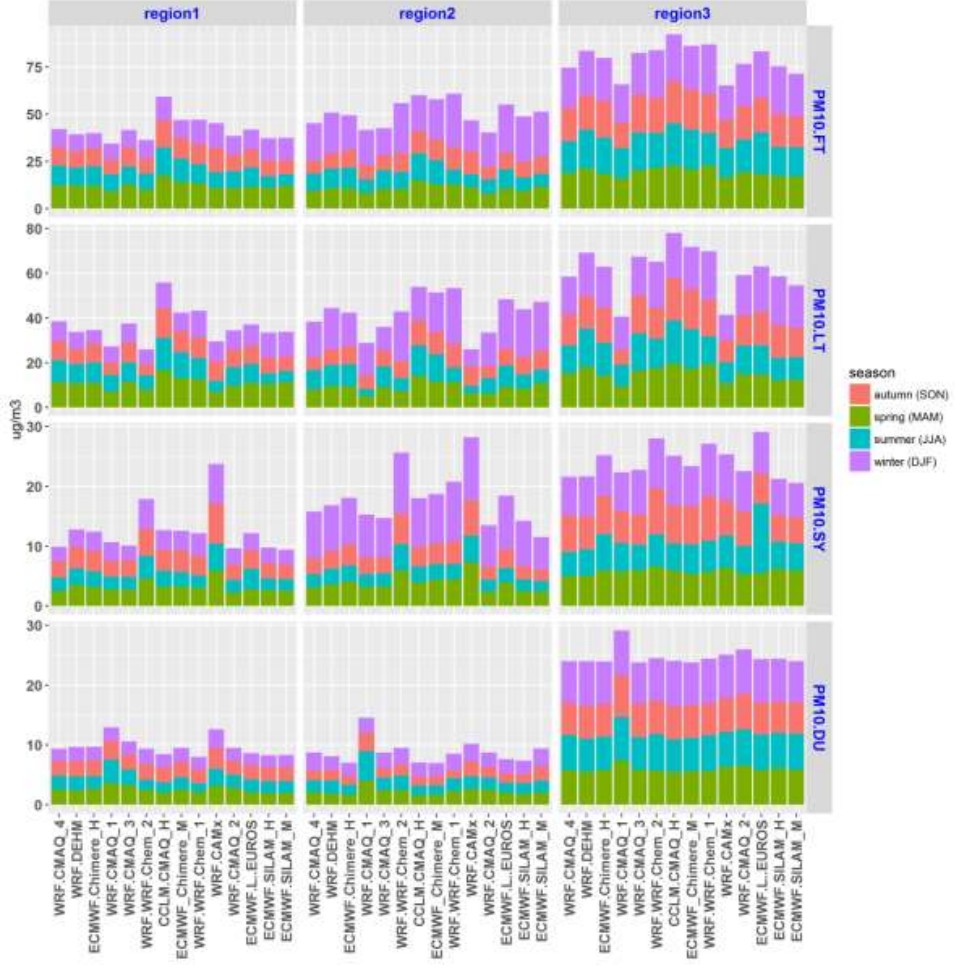

**FIGURE 24.** As in Figure 9 for $PM_{10}$ in Europe (error units in μg/m$^3$)





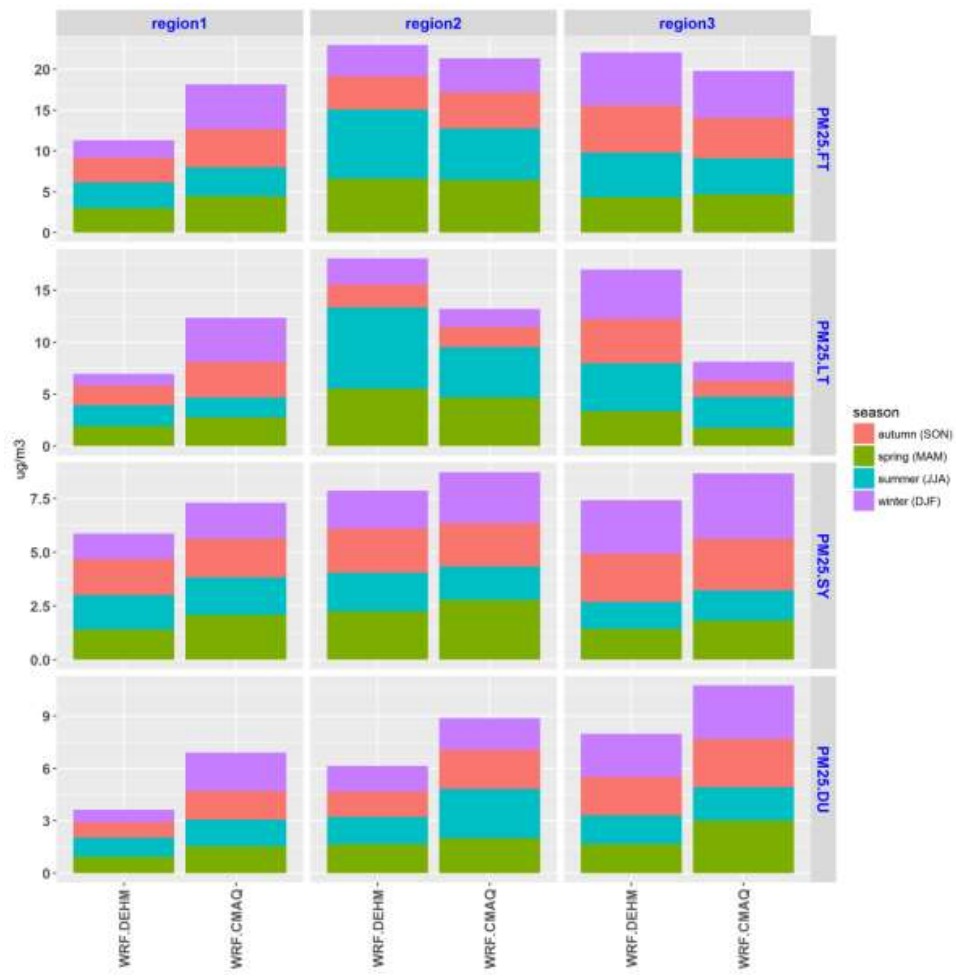

**FIGURE 25.** As in Figure 9 for $PM_{2.5}$ in North America (error units in µg/m$^3$)





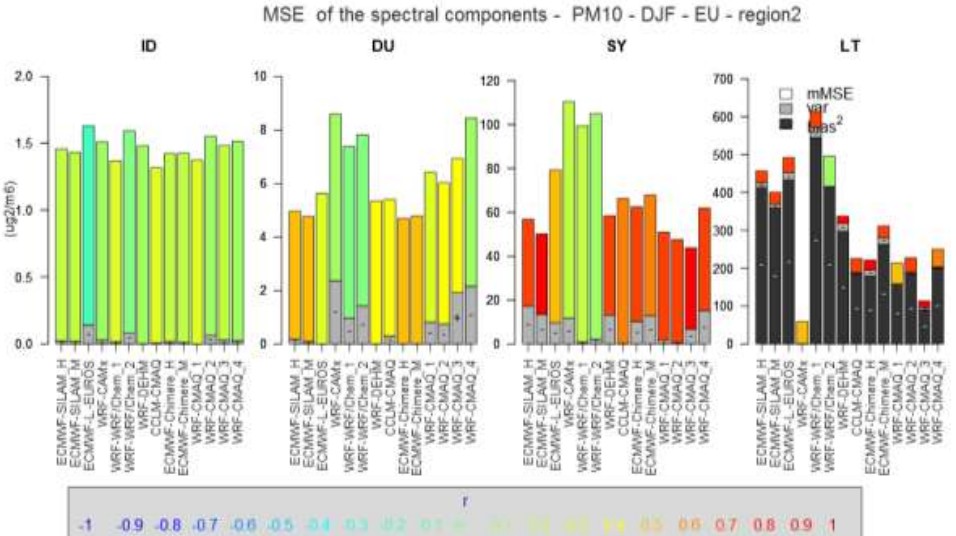

**FIGURE 26.** As in Figure 10 for PM$_{10}$ in Europe (error units in μg/m$^3$)

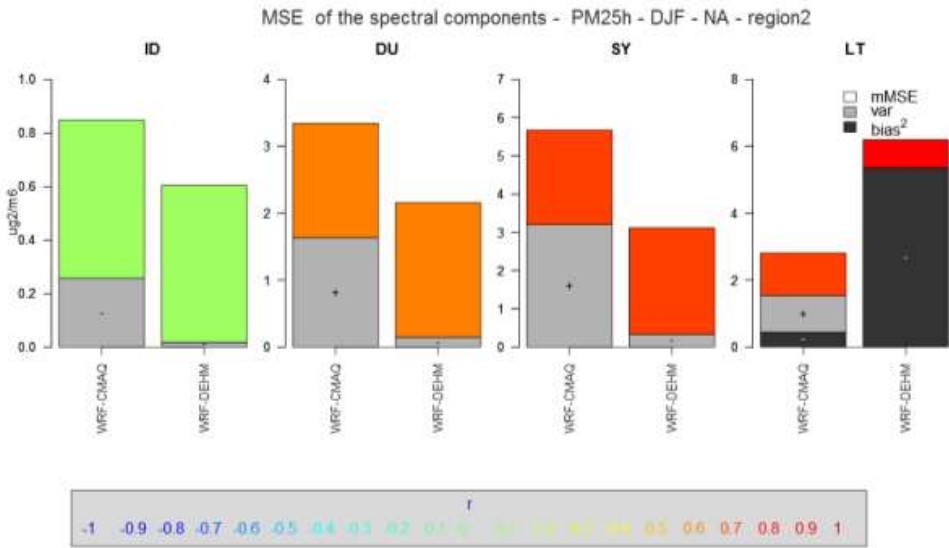

**FIGURE 27.** As in Figure 10 for hourly PM$_{2.5}$ in North America (error units in μg/m$^3$)











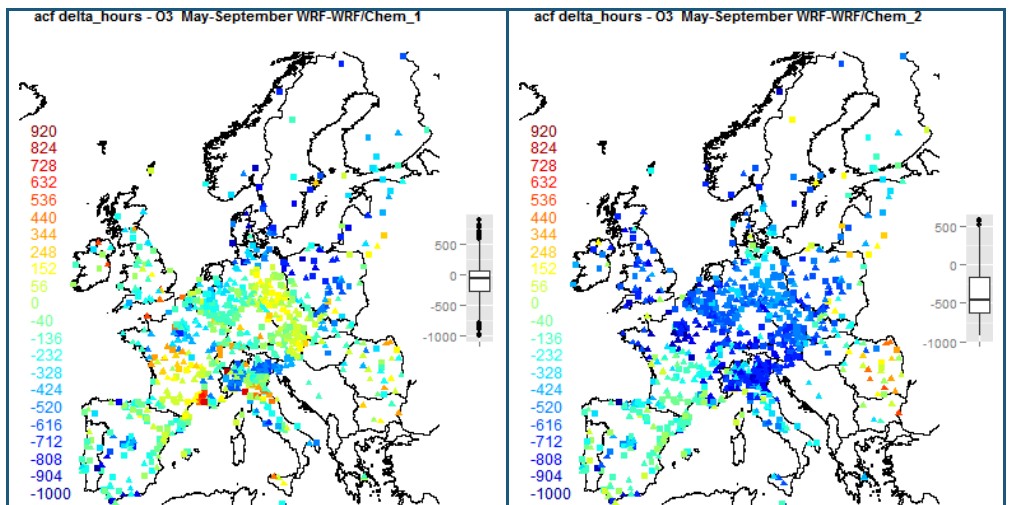

**FIGURE 28.** Spatial map of the ozone monitoring stations colored based on the 'delta hour' values, i.e. the difference in hours between the zero of the autocorrelation function (acf) for the modelled ozone minus the zero of the acf of the observed one. The acf is calculated on the long term component for the months of May to September. Negative values indicate an excess of removal (viceversa for positive values). The box on the right summarises the delta hour percentile distribution.

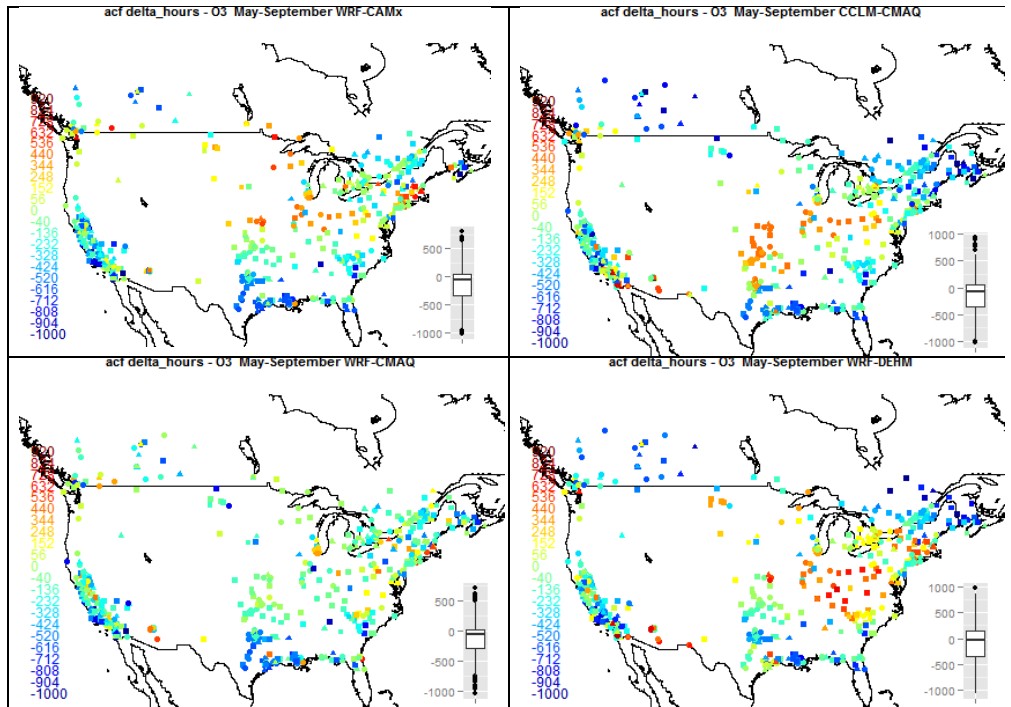

**FIGURE 29.** As in Figure 28 for North America