# Peer review of "Evaluation and Error Apportionment of an Ensemble of Atmospheric Chemistry Transport Modelling Systems: Multi-variable Temporal and Spatial Breakdown"

_Atmospheric Chemistry and Physics, 2016_

## Referee Comment (RC1) · P. A. Makar (Referee) · 14 Oct 2016

My review is in the uploaded pdf file attached - since I included an equation in my review as well as a few font changes which would not translate well as plain text. The text part of the review is repeated below as well:

My rating for this paper is minor revisions; no additional analysis is necessary. At the same time, I have a number of comments, questions and suggestions for the authors, which would improve the usefulness and "cite-ability" of the paper by the general research community. Main Points: (1) Regarding the limitations of KZ filtering (page 5,

lines 190 to 199): The authors state that "a clear-cut separation of the components of Equation (8) is not achievable, since the separation is a non-linear function of the parameters m and k ... and the leakage among the components mixes together in each component different physical processes". I agree with the authors that the choice of m and k values which have been used to date in their and other analyses quoted, along with the construction of equation (8) from the differences between KZ low-pass filters of relatively close m,k pairs, results in unwanted energy overlap across the spectral components. However, there are other options which could be used to minimize the potential for energy overlap. For example, the frequency analysis of the KZ(103,5), KZ(13,5), KZ(3,3) pairs carried out by Hogrefe et al (2000) (their Figure 1 on page 2086 of that article) shows the nature of the overlap issue – the KZ filter does not have a sharp cut-off in energy as a function of frequency, so that, for example, the low-pass KZ(3,3) passes 100% of the 1/week variation, while the KZ(13,5) passes about 13% of the 1/week variation (with the result that about 13% of the 1/week energy overlaps between the "SY" and "DU" time series, and differences between the two may have interference due to this overlap). The unmodified KZ filter is thus imprecise, though there are strategies which could reduce this imprecision. For example, rather than making use of the KZ filter as a band-pass through differencing, one could choose m,k values which represent the complete elimination of energy for frequencies higher than the given limit. Specifically, the frequency of the KZ filter's 50% energy pass limit is given by the equation below: (see attached pdf file) (1) From inspection of Hogrefe et al (2000)'s energy diagram, it can be seen that the low frequency cutoff limit (i.e. the frequency above which 99% or more of the energy will be removed by the low-pass filter) is about 2.82 times the 50% frequency from the formula above. One can thus choose values of m,k for which most of the energy is removed (e.g. a KZ(523,3) will remove 99% of the energy corresponding to periods shorter than 30 days, KZ(95,5) will remove 99% of the energy corresponding to periods shorter than 1 week, KZ(17,3) will remove 99% of the energy corresponding to periods shorter than 1 day). Using these KZ(m,k) values (and comparing the analyses for them) will also show the impact of the different time

scales just as well as the band-pass approach currently in use by the authors - without the issue of energy overlap due to attempting to use KZ as a band-pass. This as an alternative to attempting a band-pass by differencing two close low-pass filters. Another option is to use the modified KZ filter known as the KZ Fourier Transform (KZFT), wherein the original moving average is multiplied by a complex exponential function centered on the desired center wavelength. This is a better option for band-pass than the differencing in the references quoted by the authors, though it has the disadvantage of being a very narrow band-pass (see Yang and Zurbenko, WIREs Comp Stat, 2, pp 340-351, 2010). My point here is not that the authors approach is invalid (it has limitations, and they've stated its limitations accurately) – but there are other ways to make use of the KZ filter which will be less prone to energy overlap (and thus blurring of the impacts of time scale) aside from the strategy used to date. i.e. while a "clear cut separation of the components of equation 8 is not achievable", one doesn't necessarily need to use equation (8) to recover the effects of different time scales with a KZ filter, and there are other strategies which can get around this problem. A few lines of discussion acknowledging these possibilities should be added to the existing discussion. (2) The discussion on the emissions inventories (lines 211 to 237) was a bit hard to follow. Lines 211 to 220 read like a single inventory was used, while lines 224 to 225 mention two inventories, and which inventories were used for which models is not always clear. Some of this seemed to contradict some of the information about the individual modelling systems appearing later in the manuscript (where modified emissions are mentioned in some model system descriptions), with the result that the reader is not able to determine exactly which emissions inventories were used with which model, and the extent to which emissions were invariant between modelling systems. The authors should clarify this by including the emissions inventory(/ies) employed in each model in their summary table comparing the models, and modify the text accordingly. (3) The text descriptions of the models were uneven in the level of detail – some described all of the individual model parameterizations with references, some were much shorter, some overlapped the information in the table, some did not, some described

processes not described in others. This makes it difficult for the reader to understand the differences between the different modelling systems, hence draw inferences for the differences in model results. Rather than repeat the table, could the authors use the text in this section to describe only those components of the models which are unique from the others, particularly for the case of multiple implementations of the same model (e.g. have one WRF-CHEM main description followed by a paragraph describing the variations used in the study, ditto for WRF-CMAQ, etc.)? Part of what readers of the article will want to do is determine which key differences between the models are responsible for some of the differences in model results – this is difficult to do with the current formatting. (4) Data analysis methodology, lines 441 – 443 and 449 – 451: the means of hole-filling for data gaps in the temporal records for the accepted stations should be described (e.g. local interpolation for smaller gaps? Average over all values for all gaps?). Lines 449 to 451 are a bit unclear: why was spatial averaging carried out and what were the domains? I think this may need a line or two at this point in the text to the effect of "hierarchical clustering was used to determine sub-regions with similar characteristics – spatial averaging within these sub-regions was carried out due to the similarity of the observation data within these regions implying they will experience common chemistry". . . or words to that effect. (5) For the analysis itself (sections 3 and 4): the analysis tended to focus on how the models performed, as opposed to why differences in performance took place. The former is a valuable service in describing the state of the science, which has now appeared in all three phases of AQMEII – but the latter is of interest for those wishing to use the comparisons to further improve model performance. I'm hoping that the authors could take the time (I'm thinking a few days of discussion followed by an additional page of text in the manuscript) to delve a little bit deeper in their evaluation to suggest/speculate why certain models had poor performance for some predicted variables while others had better performance, in order to provide guidance to the community on how to move the science behind these simulations forward. Some examples: a. Lines 518-522: This subset of models had the worst performance for wind speed – what makes them different from the other models

in this regard? A particular variation of the met driver? Different surface characteristics? b. Lines 548-550: This is an important result – a common problem across many models. For those models which seemed to be the least affected by this problem – what makes them different from the other models? c. Dry deposition discussion (section 3.2): WRF-DEHM was different from the other models – why? What is different about that model's deposition setup which might give rise to this result? d. Lines 573 – 576: There is a factor of 7 difference between the different model's PM2.5 deposition for the EU – what are the main differences in model PM2.5 processes between the models which could contribute to these differences? e. Section 3.3.1 – most of the error seems to reside in the LT component as bias – but not all models are the same; can the authors suggest to what components of the models the differences might be attributed? f. Lines 720-724: The common model EU negative bias of the mean NO2 is an important result – noting that the winter bias is usually positive, this implies that the summer bias may be quite negative. What possible causes might contribute to this bias, based on the different models' performance? Common positive bias of the PBL height (except in winter) perhaps? Photolysis rates too high? Shading effects missing, forest canopy or urban canopy? Emissions estimates for residential combustion low? – Line 751 suggests emissions as the key feature – but there is variation across the models which might give some insights into other factors. g. Lines 869-878: Most SO2 emissions are due to large stack sources. How are SO2 emissions distributed in the vertical in the different models? Are they all using the same plume rise algorithm? Is there any correlation between model vertical resolution and SO2 performance (LT bias)? The ECMWF-L-EUROS, WRF-WRF/Chem2, and ECMWF-chimere models had a large negative bias – are there any commonalities between these models that might account for this common negative bias? For that matter, what are the main differences between WRF-WRF/Chem1 and WRF-WRF/Chem2 which might account for the substantial difference in SO2 bias between these two relatively similar models? Meanwhile WRF-CMAQ3 has a large positive bias – what makes it different from the other implementations? h. Section 3.3.6: the SY correlation for PM2.5 is poor for three specific

models (WRF-CAMx, WRF-Chem1, and WRF-Chem2) – why? What do these models have in common and/or are different from the other models? i. Section 4 – the models' performance for this covariance analysis seemed to show the most variation across northern Germany and the Benelux countries; compare WRF-CAMx and ECMWF-L-EUROS to WRF-CMAQ3, CCLM-CMAQ-N. The ECMWF based models seemed to get positive numbers there, WRF based models negative. The implication is a meteorological driver bias leading to a difference in O3 memory. What met factors might be having this effect? Is there a corresponding regional temperature bias, for example? WRF-Chem1 and WRF-Chem2 had different performance – what's different between these implementations which might lead to these differences. The above are a few examples I noticed from the work – which shows in detail the extent to which the models differed, and at different time scales, but doesn't discuss why they might be different to any great extent. I recommend the authors include a paragraph or three in the conclusions suggesting possible causes for these differences, and recommendations for their investigation. (6) Several times in the discussion, the authors attribute common poor diurnal (DU timescale) performance on poor meteorological performance, since the latter has a significant diurnal variation. I agree that this may be one possible cause of the problem – another might be poor quality of the diurnal portion of the temporal variation in the driving emissions (c.f. Makar, P.A., Nissen, R., Teakles, A., Zhang, J., Zheng, Q., Moran, M.D., Yau, H., diCenzo, C., Turbulent transport, emissions and the role of compensating errors in chemical transport models, Geosci. Model Dev., 7, 1001-1024, 2014), where we showed some examples of the impact of poor temporal splitting of specific source types on model performance). How well does the temporal variation in the input CO emissions in the EU (see lines 607-616) correspond to observed near-source variations? Also, DU and smaller time-scale performance may correspond to errors in the wind direction taking the modelled plumes from sources in a different direction from reality. In that respect, a wind direction comparison in addition to wind speed would be very useful (is this do-able with the submitted data)?

Minor issues: Line 397: HZG has not been defined.

Line 441: the means of hole filling for data gaps should be outlined – were averages of the entire period used for all gaps, or were smaller gaps filled by local interpolation, for example?

The inset map figures are I think supposed to show the station locations for the vertical profiles – these locations are very difficult to make out. I don't see why the inset maps need to show any sort of concentration field (impossible to read that for their size anyway) – please replace with a white background with a large symbol showing the station location.

Lines 560 to 565: Not really clear to the reader how the deposition figures were generated; please clarify. A total accumulation in deposition would be a single number for each model, while these are distributions. The different models had different horizontal resolutions – were the deposition outputs from the models accumulated to a common grid prior to calculating the distributions shown? Otherwise this may be an apples to oranges comparison; a model with a higher resolution would tend to have a greater variability than a lower resolution model due to less spatial averaging of surface characteristics.

Line 711-712: this lack of dependence on the NO2/NOx emissions ratio should not be a great surprise given the fast chemistry between NO2 and NO.

Lines 781-784, lines 830-834: the SY component low precision is interesting – is there a seasonality that might be linked to downslope winds in mountainous areas? EU3 being surrounded by mountains – this made me wonder about tropopause fold events. These can sometimes have a big impact on ozone downwind, if a mechanism (such as convection or foehn wind circulation) exists to transfer the ozone further towards the surface from the middle troposphere – cf Makar, P.A., Gong, W., Mooney, C., Zhang, J., Davignon, D., Samaali, M., Moran, M.D., He, H., Tarasick, D.W., Sills, D., and Chen, J., Dynamic adjustment of climatological ozone boundary conditions for Air-Quality Forecasts, Atmos. Chem. Phys. 10 (6), 8997-9015. Do the different met models have

a mechanism to parameterize troposphere/stratosphere exchange events? What was the upper boundary condition employed by the models for ozone (and other species)? Those with a higher top and a more detailed meteorology might capture fold events better than those with a lower top and/or less detailed meteorology.

Lines 805 – 808: my own work suggests that the bias error may be due to the absence of forest shading in most air-quality models (EGU presentation and ITM conference proceedings so far, paper under review) – this would also be consistent with the NO2 underprediction showing up in the EU results.

Text on Figure 21 is too small to read.

Section 3.3.4: This makes sense in terms of the chemistry, but the driving causes for those chemical changes are less clear. Temperature gradient or PBL height might be worth checking – is the bias due to too stable / low PBL in winter (too high in summer)?

Line 1081: probably should be "conclusions" rather than "considerations" in this sentence.

Please also note the supplement to this comment:
http://www.atmos-chem-phys-discuss.net/acp-2016-682/acp-2016-682-RC1-supplement.pdf

---

## Referee Comment (RC2) · Anonymous Referee #2 · 11 Nov 2016

Solazzo et al. (S2016) present an interesting model-observation analysis, making use of a range of model data from the third phase of the Air Quality Model Evaluation International Initiative (AQMEII), of primary and secondary air pollutants important for our understanding of the chemistry of the atmosphere and how well models simulate it. This sort of study is vital as more and more model evidence is used to link exposure to air pollution and impacts.

The model and observed time series were broken down using spectral decomposition - breaking the time series into it's spectral components - and further analysis

focused on separating the mean square error between the model and observations to better characterise the sources of error with the aim of attributing them to specific sources/processes. This technique was recently published by the same authors Solazzo and Galmirini, 2016) and the work here greatly extends previous analysis.

In general I think this paper should be published following several minor revisions. In my opinion, more emphasis needs to be spent on evaluating models at the process level and this paper raises a promising avenue for others. As the authors say, this method has not completely determined the causes of model error - but I think that modifications to and expansions of the method will lead to improved insight in the future.

Major comments: My main criticism of this paper comes about in the presentation of the results. There are multiple figures at such poor resolution that I almost had to give up looking at them. Many of the graphics look like they are plotted in R and I would encourage the authors to save the graphics in a high resolution output (pdf, eps etc). I also think that the reader would benefit from a consistent set of axes limits for plots in EU and NA and the ordering of the models should follow those used in Figures 23 and 26 (i.e. clustering the WRF-Chem variants together and the WRF-CMAQ variants together). When I did this "by hand" I found that the intra model variability was large - no doubt reflecting different model options (chemistry schemes etc).

My other minor criticism is the choice of the domains. I just wonder if country specific (or State specific in the USA) boundaries were used could we learn more about emission estimate biases? I would imagine that the averaging over states and countries in the current classification will smear out these effects to some extent depending on their heterogeneity.

Minor comments: line 138: Delete the first "known".

line 162/Section 2.2 opening paragraph: There are other spectral filtering methods (e.g. Bowdalo et al., 2016 ACP). A comment on these would be useful and why the kz approach has been selected.

line 191: re-phrase the sentence starting "A clear-cut...".

line 197: add "do" after "they".

line 336: add "wave" for radiation.

line 458: what is the source of the meteorological data that you compare against?

line 647: It's not clear what you mean by timescale here? Do you mean the e-folding lifetime? Should that not depend on the concentration of NO?

line 699: remove duplicate round bracket.

line 726: insert "is" after "bias".

line 728, 826, 823, ...: add space between mixing ration and units (e.g. 11.5ppb -> 11.5 ppb). This is a common error so please search the document for this.

line 766: insert space "Figure17and".

line 824: Do the authors really believe that vertical mixing can be analysed through something as complex as ozone? I would suggest that vertical mixing needs sources with no chemistry to be understood (e.g. Rn or Pb).

lines 841: Is this really good?

line 858: "sulphates" should be "sulfates".

line 881: What about oxidants? Could the corr(biasO3, biasSO2) be useful?

line 928: What about temperature effects? Could corr(biasTemp, biasPM) be useful?

––––––––––––––––––––––––––––

---

## Author Comment (AC1) · 19 Dec 2016

"*Evaluation and Error Apportionment of an Ensemble of Atmospheric Chemistry Transport Modelling Systems: Multi-variable Temporal and Spatial Breakdown*", by Efisio Solazzo et al, ACP, 2016.

We are thankful to both reviewers for the positive comments and useful suggestions. We have revised the manuscript in many parts to take on board the suggestions and improve the exposure and discussion of the results. All the figures have been replaced to accommodate change of scale and grouping of models. A new table has been added, summarising the configuration of the WRF model adopted by the modelling groups. The evaluation of model results has been extended to include also the wind direction. More than twenty new references have been added to integrate the discussion of the results.

**Response to reviewer #1**.

My rating for this paper is minor revisions; no additional analysis is necessary. At the same time, I have a number of comments, questions and suggestions for the authors, which would improve the usefulness and "cite-ability" of the paper by the general research community.

Main Points:
(1) Regarding the limitations of KZ filtering (page 5, lines 190 to 199): The authors state that "a clear-cut separation of the components of Equation (8) is not achievable, since the separation is a non-linear function of the parameters m and k … and the leakage among the components mixes together in each component different physical processes". I agree with the authors that the choice of m and k values which have been used to date in their and other analyses quoted, along with the construction of equation (8) from the differences between KZ low-pass filters of relatively close m,k pairs, results in unwanted energy overlap across the spectral components. However, there are other options which could be used to minimize the potential for energy overlap. For example, the frequency analysis of the KZ(103,5), KZ(13,5), KZ(3,3) pairs carried out by Hogrefe et al (2000) (their Figure 1 on page 2086 of that article) shows the nature of the overlap issue – the KZ filter does not have a sharp cut-off in energy as a function of frequency, so that, for example, the low-pass KZ(3,3) passes 100% of the 1/week variation, while the KZ(13,5) passes about 13% of the 1/week variation (with the result that about 13% of the 1/week energy overlaps between the "SY" and "DU" time series, and differences between the two may have interference due to this overlap). The unmodified KZ filter is thus imprecise, though there are strategies which could reduce this imprecision. For example, rather than making use of the KZ filter as a band-pass through differencing, one could choose m,k values which represent the complete elimination of energy for frequencies higher than the given limit. Specifically, the frequency of the KZ filter's 50% energy pass limit is given by the equation below:

$$\omega_0(m,k) = \frac{\sqrt{6}}{\pi} \sqrt{\frac{1-\left(\frac{1}{2}\right)^{\frac{1}{2k}}}{m^2-\left(\frac{1}{2}\right)^{\frac{1}{2k}}}} \tag{1}$$

From inspection of Hogrefe et al (2000)'s energy diagram, it can be seen that the low frequency cutoff limit (i.e. the frequency above which 99% or more of the energy will be removed by the low-pass filter) is about 2.82 times the 50% frequency from the formula above.    One can thus choose values of m,k for which most of the energy is removed (e.g. a KZ(523,3) will remove 99% of the energy corresponding to periods shorter than 30 days, KZ(95,5) will remove 99% of the energy corresponding to periods shorter than 1 week, KZ(17,3) will remove 99% of the energy corresponding to periods shorter than 1 day).    Using these KZ(m,k) values (and comparing the analyses for them) will also show the impact of the different time scales just as well as the band-pass approach currently in use by the authors - without the issue of energy overlap due to attempting to use KZ as a band-pass. This as an alternative to attempting a band-pass by differencing two close low-pass filters. Another option is to use the modified KZ filter known as the KZ Fourier Transform (KZFT), wherein the original moving average is multiplied by a complex exponential function centered on the desired center wavelength. This is a better option for band-pass than the differencing in the references quoted by the authors, though it has the disadvantage of being a very narrow band-pass (see Yang and Zurbenko, WIREs Comp Stat, 2, pp 340-351, 2010).

My point here is not that the authors approach is invalid (it has limitations, and they've stated its limitations accurately) – but there are other ways to make use of the KZ filter which will be less prone to energy overlap (and thus blurring of the impacts of time scale) aside from the strategy used to date.    i.e. while a "clear cut separation of the components of equation 8 is not achievable", one doesn't necessarily need to use equation (8) to recover the effects of different time scales with a KZ filter, and there are other strategies which can get around this problem. A few lines of discussion

acknowledging these possibilities should be added to the existing discussion.

Response. Quoting Rao et al (1997): '*Poor separation leaves together in each component completely different physical phenomena*', we recognise that overlapping of frequencies is a serious shortcoming of the applied filter and we specifically warn the reader about it. Keeping in mind that the main scope of this work is to attribute error to processes, however, we don't believe that the overlapping among frequencies is the main obstacle for not having succeeded to achieve our main goal.

The analysis we propose in this study is mostly applicative - in the sense that we wish to apply existing methods to investigate the error of a suite of models. The kz filtering we adopt is based on solid theory and abundant applications that makes it, in the formulation used in manuscript, the most robust available (proposed and adapted to ozone by Zurbenko and colleagues over the years). There are several combinations of (m,k) parameters and many other methods that could have been used, but we opt for the formulation mostly applied and documented (along with its shortcoming) rather than exploiting a new one in this applicative context. In previous publications (Hogrefe et al 2003; Kioutsioukis and Galmarini 2014; Galmarini et al 2013; Solazzo et al 2015 and 2016) the authors have described to some length other available options and the reason as to why the kz band pass was chosen. About the method proposed by the reviewer, it is certainly an option that would require dedicated testing that are beyond the aim of the current study. The following text has been added to the revised section 2.2:

'*Other spectral techniques could be used but either they do not guarantee the absence of signal leakage (e.g. anomaly perturbation method) or require special treatment of missing data (e.g. wavelet transform method) (Rao et al., 1997; Eskridge et al., 1997), or are more convoluted (e.g. kz-Fourier Transform), or simply have not been applied as frequently as the kz filter to air quality data (e.g. Bowdalo et al., 2016). Hogrefe et al. (2003) provided an exhaustive comparison among four techniques for separating different time scales in atmospheric variables (kz, kz-Fourier Transform, wavelet transform and elliptic filter) and concluded that they all gave qualitatively similar results in terms of the variance distribution among components and that no single filter outperformed the others for all applications.*'.

(2) The discussion on the emissions inventories (lines 211 to 237) was a bit hard to follow. Lines 211 to 220 read like a single inventory was used, while lines 224 to 225 mention two inventories, and which inventories were used for which models is not always clear. Some of this seemed to contradict some of the information about the individual modelling systems appearing later in the manuscript (where modified emissions are mentioned in some model system descriptions), with the result that the reader is not able to determine exactly which emissions inventories were used with which model, and the extent to which emissions were invariant between modelling systems. The authors should clarify this by including the emissions inventory(/ies) employed in each model in their summary table comparing the models, and modify the text accordingly.

Response. We have clarified in the text and added the adopted emissions in the summary table 1.The following text have been added/modified:

'*The 'standard' emission inventories are those developed for the second phase of AQMEII for EU and NA and extensively described in Pouliot et al. (2015). For EU, the TNO-MACC-II (Netherlands Organization for Applied Scientific Research, Monitoring Atmospheric Composition and Climate) inventory of anthropogenic emissions for the year 2009 was used, while biogenic emissions (meteorology-dependent) were specifically calculated for the year of 2010 by several groups. Five modelling systems have used the EDGAR-HTAPv2.2 emission inventory (Janssens-Maenhout et al., 2015), which complements the standard MACC inventory in regions outside EU (Table 1). The two inventories (MACC and HTAP) are approximately the same over the common part of EU (the standard MACC inventory does not cover North Africa, while it does cover eastern Europe, including Russia and Turkey.), and only differ for regions outside the EU borders but within the domain boundaries, such as North Africa. Some discrepancies might exist among the two inventories (e.g in the emissions from ships). For Chimere, the MACC inventory over France and the UK was spatially redistributed considering national inventories (having higher spatial resolution), while for the other countries it was redistributed by considering point source locations, land-use and population. For processing the HTAP inventory, population was not used as a parameter for spatially distributing the emissions.*

*For the NA domain, the 2008 National Emission Inventory was used as the basis for the 2010 emissions, providing the inputs and datasets for processing with the Sparse Operating Operator Kernel Emissions (SMOKE) processing system (Mason et al., 2012). Specific updates for the year of 2010 were made for several sectors, including mobile sources, power plants, wildfires, and biogenic emissions. Details are given in Im at al. (2015a,b) and Pouliot et al. (2015).*

*Typically, emission processors use annual emission total, while AQ models require hourly input values. Therefore, proxies*

*variables and surrogate fields are used to spatially disaggregate the annual total and to allocate them temporally. The overall model accuracy heavily depends on the degree of similarity between the disaggregation of total emission and the true spatial and temporal distribution (Makar et al., 2014). Furthermore, the emissions for EU, being compiled on a country-wise basis, are affected by gaps and inconsistency across borders which require further processing and manipulation (Pouliot et al., 2015).'*

(3) The text descriptions of the models were uneven in the level of detail – some described all of the individual model parameterizations with references, some were much shorter, some overlapped the information in the table, some did not, some described processes not described in others. This makes it difficult for the reader to understand the differences between the different modelling systems, hence draw inferences for the differences in model results. Rather than repeat the table, could the authors use the text in this section to describe only those components of the models which are unique from the others, particularly for the case of multiple implementations of the same model (e.g. have one WRF-CHEM main description followed by a paragraph describing the variations used in the study, ditto for WRF-CMAQ, etc.)? Part of what readers of the article will want to do is determine which key differences between the models are responsible for some of the differences in model results – this is difficult to do with the current formatting.

Response. The section 2.3.1 has been revised significantly. A new table (Table 2) summarises the options used by the WRF runs and a new paragraph summarises the features of the CMAQ runs. Features of each modelling systems (including Chimere, SILAM, L.-Euros, CCLM) are described individually. There is some overlap between the tables and the information provided in the text, but that does no harm. Any further information can be retrieved from the references provided.

**TABLE 2. CONFIGURATION OF THE WRF MODEL BY MODELLING GROUP**

| Operated by | Input data | Number of Vertical levels | 1th Layer Height | PBL model | Surface Layer | Land Surface | Cloud Microphysics | Cumulus Convection | SW/LW Radiation | Data Assimilation |
|---|---|---|---|---|---|---|---|---|---|---|
| University of L'Aquila | ECMWF | 33 | 10m | MYNN | MM5 Similarity | NOAH | Morrison | Grell-Freitas | RRTMG | Grid analysis nudging nudging above PBL |
| University of Murcia | ECMWF | 33 | 21m | YSU | Eta Similarity | NOAH | Lin | Kain- Fritsch 2 | RRTMG | Grid analysis nudging nudging above PBL |
| Ricerca Sistema Energetico | ECMWF | 33 | 25m | YSU | Eta Similarity | NOAH | Morrison | Grell-Freitas | RRTMG | Grid Analysis nudging also within the PBL |
| University of Aarhus | ECMWF | 29 | 20m | MYJ | Eta Similarity | NOAH | WSM5 | Kain-Fritsch2 | CAM | Grid analysis nudging nudging above PBL |
| Istanbul Technical University | NCEP FNL | 30 | 10m | YSU | Eta Similarity | NOAH | WSM3 | Kain-Fritsch2 | Dudhia/RRTM | Grid Analysis nudging also within the PBL |
| Kings College | NCEP GFS | 23 | 14m | ACM2 | Pleim-Xiu | RUC | WSM6 | Kain-Fritsch 2 | Dudhia/RRTM | Grid Analysis nudging also within the PBL |
| Ricardo E&E | NCEP GFS | 23 | 15m | ACM2 | Pleim-Xiu | RUC | WSM6 | Kain-Fritsch 2 | Dudhia/RRTM | Grid analysis nudging nudging above PBL |
| University of Hertfordshire | ECMWF | 36 | 25m | ACM2 | Pleim-Xiu | 5-layer thermal diffusion | Morrison | Kain-Fritsch2 | RRTMG | Grid analysis nudging nudging above PBL |
| U.S. Environmental | NCEP NAM | 35 | 20m | ACM2 | Pleim-Xiu | Pleim-Xiu | Morrison | Kain-Fritsch2 | RRTMG | Grid analysis nudging |

| Protection Agency | analysis | | | | | | | | | above PBL; |
|---|---|---|---|---|---|---|---|---|---|---|
| RAMBOLL Environ | NCEP NAM analysis | 35 | 20m | ACM2 | Pleim-Xiu | Pleim-Xiu | Morrison | Kain-Fritsch2 | RRTMG | Grid analysis nudging above PBL |

(4) Data analysis methodology, lines 441 – 443 and 449 – 451: the means of hole-filling for data gaps in the temporal records for the accepted stations should be described (e.g. local interpolation for smaller gaps? Average over all values for all gaps?).

Response. No imputation on missing values has been performed, in the sense that the missing data have been treated as missing during the analysis. The kz filter can handle missing data and the score statistics has been calculated only on complete pair of model-observations, as it was done for the previous AMQEII-related analyses.

Lines 449 to 451 are a bit unclear: why was spatial averaging carried out and what were the domains?

Response. Spatial averaging has been carried out on the selected sub-regions shown in figure 1. The reason for spatially averaging is to ease the display of the results. The purpose of the clustering the observed signal is indeed to identify a pool of receptors where the signal is homogeneous up to a given similarity threshold (set to 0.65). The spatial average carried out on that pool of stations returns a smoothed signal that is representative of the cluster.

I think this may need a line or two at this point in the text to the effect of "hierarchical clustering was used to determine sub-regions with similar characteristics – spatial averaging within these sub-regions was carried out due to the similarity of the observation data within these regions implying they will experience common chemistry"… or words to that effect.

Response. Done as suggested

(5) For the analysis itself (sections 3 and 4): the analysis tended to focus on how the models performed, as opposed to why differences in performance took place. The former is a valuable service in describing the state of the science, which has now appeared in all three phases of AQMEII – but the latter is of interest for those wishing to use the comparisons to further improve model performance. I'm hoping that the authors could take the time (I'm thinking a few days of discussion followed by an additional page of text in the manuscript) to delve a little bit deeper in their evaluation to suggest/speculate why certain models had poor performance for some predicted variables while others had better performance, in order to provide guidance to the community on how to move the science behind these simulations forward.

Response. Our study offers an outlook of where state-of-the-science regional AQ models currently stand, for a variety of species and meteorological fields at the surface and in the troposphere, with a time scale analysis enlarged to several of the most well-known and applied models worldwide. In this respect the work presented here is unprecedented.

After three phases of AQMEII and a number of related publications, we have noticed that advancements in model performance are rather limited or absent, and that the discussion based on 'speculation' and 'conjectures' about possible causes of model error was not helping towards enhancing the modelling experience, and risking to become sterile. Indeed, avoiding 'speculation' and 'conjectures' about possible causes of model error was the driving motivation of the error apportionment method we have devised. So far, the method has proven helpful in framing the time-scale of the error. The *time scale information* is passed to modellers who, based on the feedback received, try i) to detect the process responsible for the error at that time scale and/or ii) together with the AQMEII community to explore methods for a deeper investigation before applying changes to the model. With this work, we currently are at the beginning of the implementation, i.e. at the *information* step. There is an intrinsic limit in any diagnostic methodology that can be overcome only by exercising the model, in all its features and possibilities, which cannot be done by a large community. Although there is an added value in evaluating more models than one at a time, the results presented in this study are meant to guide the individual groups to target the direction they want to move.

The literature is rich with possible motivations behind the inaccuracy of the models for many years now, with (excluding a limited number of modelling teams) little tangible advancement. We try not to add any more hypotheses as to why the errors occur, and the reason is that in most of the cases the *modellers simply don't know*. AQ models have grown in complexity and nonlinearity beyond our capacity to control each process in isolation, to the extent that we are actually questioning the fruitfulness of future evaluation studies that do not envision specific sensitivity analysis (sensitivity to processes and conditions). Furthermore, there is another type of non-commensurable error (almost) never considered, that is the error produced by the modellers/users in manipulating, extracting, transforming, and submitting the results for analysis. Our experience with AQMEII indeed suggests that, for any given variable, about one in 13-14 models

provided 'bugged' data. Sometimes these errors are easily spotted and corrected/removed, some other times they are subtle and elusive to an averaged model inter-comparison such as the one presented here.

That said, it is well recognised that even small, subtle changes in model configuration strongly influence the air quality calculations and that air quality model scores vary by time of the day, season, region, emission regime, etc. The exercise to synthetize the enormous amount of information provided by AQMEII3 has brought up some interesting commonalities among model deficiencies, but any attempt to link those to processes has not been fruitful because of the high non-linearity of the interactions among different components of the models. *We have clearly stated in the conclusions that we have failed our main objective*. A second manuscript (a sort of follow up or Part II), is in preparation for this same ACP special issue where we focus on two models only (CMAQ and Chimere), making use of additional sensitivity runs where we explore at some depth the causes of model error for ozone.

In the revised manuscript we have expanded the discussion at least for the poorest model performance and revised the description of modelling configurations, but by looking at different or similar configuration of the models little can be gained. An example above all is the new wind direction comparison for EU3, where WRF models using the same settings behave very differently.

Some examples:

a. Lines 518-522:    This subset of models had the worst performance for wind speed – what makes them different from the other models in this regard? A particular variation of the met driver? Different surface characteristics?

Response. The modellers cannot explain the causes of the models error, and no robust explanations could be found depending on the models configuration, which are also common to other models (see new Table 2 where the settings of WRF are summarised)

b. Lines 548-550:    This is an important result – a common problem across many models. For those models which seemed to be the least affected by this problem – what makes them different from the other models?

Response. If the reviewer is referring to the overestimation of vertical WS profile by WRF-WRF/Chem1 model, while all the other models seems to under-predict the WS, we have clarified that the only consistent difference between WRF/Chem1 and all the other models is the difference in nudging. WRF/Chem1 adopts nudging of WS only during spin-up preceding the 72-hour run, while the other models keep the nudging active during the entire run.

c. Dry deposition discussion (section 3.2): WRF-DEHM was different from the other models – why?    What is different about that model's deposition setup which might give rise to this result?

Response. We have removed the discussion about dry deposition to avoid confusion

d. Lines 573 – 576: There is a factor of 7 difference between the different model's PM2.5 deposition for the EU – what are the main differences in model PM2.5 processes between the models which could contribute to these differences?

Response. We have removed the discussion about dry deposition to avoid confusion

e. Section 3.3.1 – most of the error seems to reside in the LT component as bias – but not all models are the same; can the authors suggest to what components of the models the differences might be attributed?

Response. Given the homogeneity of the CO error, we believe that the error stems from a common cause, quite likely emissions. Based on known results from relevant literature we have also listed a range of other possible causes: PBL stability (same as PM10), lack of temperature dependent emissions, wind speed overestimation, poor representation of the diurnal variation of the emissions.

f. Lines 720-724: The common model EU negative bias of the mean NO2 is an important result – noting that the winter bias is usually positive, this implies that the summer bias may be quite negative.    What possible causes might contribute to this bias, based on the different models' performance?    Common positive bias of the PBL height (except in winter) perhaps?    Photolysis rates too high?    Shading effects missing, forest canopy or urban canopy?    Emissions estimates for residential combustion low? – Line 751 suggests emissions as the key feature – but there is variation across the models which might give some insights into other factors.

Response. The bias for $NO_2$ is negative in winter and summer (table S7 and figure 15). We have added some discussion in the revised manuscript, pointing to possible causes of $NO_2$ bias, including PBL error and unknown processes (systematic error), such as shading. The following text has been added:

*'The bias is probably caused by a combination of factors, including emissions estimate (e.g. underestimation of residential combustion), PBL height and vertical mixing at night (when wood combustion emissions tend to be maximum, e.g. Denier Van Der Gon et al., 2015), and missing processes acting as systematic errors, such as shading effects of forested canopies (e.g. Makar et al., 2016).'*

g. Lines 869-878: Most SO2 emissions are due to large stack sources. How are SO2 emissions distributed in the vertical in the different models? Are they all using the same plume rise algorithm? Is there any correlation between model vertical resolution and SO2 performance (LT bias)? The ECMWF-L-EUROS, WRF-WRF/Chem2, and ECMWF-chimere models had a large negative bias – are there any commonalities between these models that might account for this common negative bias? For that matter, what are the main differences between WRF-WRF/Chem1 and WRF-WRF/Chem2 which might account for the substantial difference in SO2 bias between these two relatively similar models? Meanwhile WRF-CMAQ3 has a large positive bias – what makes it different from the other implementations?

Response. We have complemented the discussion of $SO_2$ with more information. Again, attempting to group the performance by features does not lead to consistent conclusions. Models like CMAQ2, CMAQ3, CMAQ4 use the same PBL module and similar vertical structure (CMAQ3 has more layers) but the bias error they produce is different (that of CMAQ3 is much higher). The following text has been added:

*'The majority of models employed the prescribed vertical distribution by EMEP (Vestreng and Støren, 2000), while CMAQ4 in EU and WRF-CMAQ in NA adopted the Briggs plume rise algorithm (Briggs, 1971; 1972) accounting for the effects of modelled meteorology, and SILAM and CCLM-CMAQ adopted the sector dependent vertical emission profiles as in Bieser et al. (2011b).'*

and

*'By contrast, WRF/Chem2, Chimere and L.-Euros show significant low bias (the latter two models have the smallest number of vertical layers). Overall, though, the bias error does not group consistently by PBL scheme and/or vertical resolution. For example, CMAQ2, CMAQ3, CMAQ4 employ the same PBL scheme based on ACM2 and have comparable number of vertical levels (CMAQ3 has even more), but the bias of CMAQ3 is much larger than that of CMAQ4 and CMAQ2 which, in turn, have comparable bias but opposite in sign. The two instances of WRF/Chem show significantly different biases, which might be due to the different PBL and cloud schemes, influencing the $SO_2$ oxidation (Table 2).'*

h. Section 3.3.6: the SY correlation for PM2.5 is poor for three specific models (WRF-CAMx, WRF-Chem1, and WRF-Chem2) – why? What do these models have in common and/or are different from the other models?

Response. We do not have an explanation for this as we cannot find any robust link between the commonality of the error to that of the models configuration. We have added the following text in support of the performance of the WRF/Chem1 model:

*'possibly due to the low nitrate concentration and high sulphate concentration during winter months, resulting from the GOCART parameterization of the aqueous cloud chemistry.'*

i. Section 4 – the models' performance for this covariance analysis seemed to show the most variation across northern Germany and the Benelux countries; compare WRF-CAMx and ECMWF-L-EUROS to WRF-CMAQ3, CCLM-CMAQ-N. The ECMWF based models seemed to get positive numbers there, WRF based models negative. The implication is a meteorological driver bias leading to a difference in O3 memory. What met factors might be having this effect? Is there a corresponding regional temperature bias, for example? WRF-Chem1 and WRF-Chem2 had different performance – what's different between these implementations which might lead to these differences.

Response. We actually found similarity with the bias of ozone, rather than with met drivers, but not consistently, in the sense that for some models (CMAQ1 for example) the similarity fails. The rate by which the memory decays with time is modulated by the large scale circulation and the met driver surely plays a fundamental role (more in terms of geopotential height), as well as the fluxes at the boundaries (emission, deposition and boundary conditions). Again, we could speculate about possible causes of model differences but we could not detect any robust pattern explainable in terms of differences in model configuration.

These above are a few examples I noticed from the work – which shows in detail the extent to which the models differed, and at different time scales, but doesn't discuss why they might be different to any great extent. I recommend the authors include a paragraph or three in the conclusions suggesting possible causes for these differences, and recommendations for their investigation.

Response. We have added some further considerations in the conclusions, not mainly related to 'possible' causes of model errors but rather to way to proceed in the evaluation to allow diagnosing more precisely the causes of errors. The

following textx has been added:

*'The bias is, by far, the primary source of error and the most important from a model evaluation/development point of view. Because it is essentially a shift of the mean concentration, the causes of it need to be sought in processes and conditions at the boundaries that have a systematic effect of displacing the concentration values while approximately preserving the shape of the distribution. Thus, processes like emission timing, chemistry transformation, autocorrelation structures, stratospheric intrusion, atmospheric stability are unlikely responsible for systematic bias-type error (while they can be source of casual inaccuracy for limited periods). On the other hand deposition fluxes, magnitude of emission, input from the lateral boundaries are more probable sources of bias error. The effect of meteorology is more complex, as errors in synoptic circulation can induce surface wind velocity and direction to be inaccurate, and thus negatively impacting on the long term modelled concentrations causing bias error.'*

And

*'Future AQ model evaluation activities should envision sensitivity simulations and process specific analyses. The 'theory of evaluation' based on information theory currently being developed by the hydrology modelling community (Nearing et al., 2016 and references therein) is a promising way forward and the AQ community should be prepared to catch those developments. Ongoing work (Solazzo et al., 2017) is being devoted to deepen the investigation of causes of model errors by focusing on two models (CMAQ for NA and CHIMERE for EU), for which additional model runs have been carried out to frame the effect of fluxes (emissions, boundary conditions and deposition) on modelled ozone.'*

(6) Several times in the discussion, the authors attribute common poor diurnal (DU timescale) performance on poor meteorological performance, since the latter has a significant diurnal variation.  I agree that this may be one possible cause of the problem – another might be poor quality of the diurnal portion of the temporal variation in the driving emissions (c.f. Makar, P.A., Nissen, R., Teakles, A., Zhang, J., Zheng, Q., Moran, M.D., Yau, H., diCenzo, C., Turbulent transport, emissions and the role of compensating errors in chemical transport models, Geosci. Model Dev., 7, 1001-1024, 2014), where we showed some examples of the impact of poor temporal splitting of specific source types on model performance).  How well does the temporal variation in the input CO emissions in the EU (see lines 607-616) correspond to observed near-source variations?  Also, DU and smaller time-scale performance may correspond to errors in the wind direction taking the modelled plumes from sources in a different direction from reality.  In that respect, a wind direction comparison in addition to wind speed would be very useful (is this do-able with the submitted data)?

Response. We have included the reference to the paper as suggested as a possibility of the error in CO, but have no means to investigate its validity. We have added the discussion about the wind direction (section 3.2 and Figure 8).

Minor issues:
Line 397:   HZG has not been defined.
Response. Done

Line 441:   the means of hole filling for data gaps should be outlined – were averages of the entire period used for all gaps, or were smaller gaps filled by local interpolation, for example?
Response. No gap filling was used. See reply to major comment 4

The inset map figures are I think supposed to show the station locations for the vertical profiles – these locations are very difficult to make out. I don't see why the inset maps need to show any sort of concentration field (impossible to read that for their size anyway) – please replace with a white background with a large symbol showing the station location.
Response. The inset maps do not show the concentration field, but just the default background Matlab uses for that type of plots. We have replaced the figures.

Lines 560 to 565:   Not really clear to the reader how the deposition figures were generated; please clarify.   A total accumulation in deposition would be a single number for each model, while these are distributions.   The different models had different horizontal resolutions – were the deposition outputs from the models accumulated to a common grid prior to calculating the distributions shown?   Otherwise this may be an pples to oranges comparison; a model with a higher resolution would tend to have a greater variability than a lower resolution model due to less spatial averaging of surface characteristics.
Response. We have removed the plots and discussion about deposition from the main text to avoid confusion and misinterpretation.

Line 711-712: This lack of dependence on the $NO_2/NO_x$ emissions ratio should not be a great surprise given the fast chemistry between $NO_2$ and NO.
Response. We have clarified in the text.

Lines 781-784, lines 830-834: the SY component low precision is interesting – is there a seasonality that might be linked to downslope winds in mountainous areas? EU3 being surrounded by mountains – this made me wonder about tropopause fold events. These can sometimes have a big impact on ozone downwind, if a mechanism (such as convection or foehn wind circulation) exists to transfer the ozone further towards the surface from the middle troposphere – cf Makar, P.A., Gong, W., Mooney, C., Zhang, J., Davignon, D., Samaali, M., Moran, M.D., He, H., Tarasick, D.W., Sills, D., and Chen, J., Dynamic adjustment of climatological ozone boundary conditions for Air-Quality Forecasts, *Atmos. Chem. Phys.* 10 (6), 8997-9015. Do the different met models have a mechanism to parameterize troposphere/stratosphere exchange events? What was the upper boundary condition employed by the models for ozone (and other species)? Those with a higher top and a more detailed meteorology might capture fold events better than those with a lower top and/or less detailed meteorology.
Response. That is a possibility and we have acknowledged in the revised text with reference to the work suggested by the reviewer. The SY error being larger in EU3 is in line with that hypothesis.

Lines 805 – 808: my own work suggests that the bias error may be due to the absence of forest shading in most air-quality models (EGU presentation and ITM conference proceedings so far, paper under review) – this would also be consistent with the NO2 underprediction showing up in the EU results.
Response. We added the hypothesis suggested by the reviewer in the revised manuscript.

Text on Figure 21 is too small to read.
Response. Corrected

Section 3.3.4: This makes sense in terms of the chemistry, but the driving causes for those chemical changes are less clear. Temperature gradient or PBL height might be worth checking – is the bias due to too stable / low PBL in winter (too high in summer)?
Response. We have added some considerations about the mixing in the revised section 3.3.4.

Line 1081: probably should be "conclusions" rather than "considerations" in this sentence.
Response. Rephrased

---

## Author Comment (AC2) · 19 Dec 2016

"*Evaluation and Error Apportionment of an Ensemble of Atmospheric Chemistry Transport Modelling Systems: Multi-variable Temporal and Spatial Breakdown*", by Efisio Solazzo et al, ACP, 2016.

We are thankful to both reviewers for the positive comments and useful suggestions. We have revised the manuscript in many parts to take on board the suggestions and improve the exposure and discussion of the results. All the figures have been replaced to accommodate change of scale and grouping of models. A new table has been added, summarising the configuration of the WRF model adopted by the modelling groups. The evaluation of model results has been extended to include also the wind direction. More than twenty new references have been added to integrate the discussion of the results.

**Response to reviewer #2**.

Solazzo et al. (S2016) present an interesting model-observation analysis, making use of a range of model data from the third phase of the Air Quality Model Evaluation International Initiative (AQMEII), of primary and secondary air pollutants important for our understanding of the chemistry of the atmosphere and how well models simulate it. This sort of study is vital as more and more model evidence is used to link exposure to air pollution and impacts. The model and observed time series were broken down using spectral decomposition - breaking the time series into it's spectral components - and further analysis focused on separating the mean square error between the model and observations to better characterise the sources of error with the aim of attributing them to specific sources/processes. This technique was recently published by the same authors Solazzo and Galmirini, 2016) and the work here greatly extends previous analysis.

In general I think this paper should be published following several minor revisions. In my opinion, more emphasis needs to be spent on evaluating models at the process level and this paper raises a promising avenue for others. As the authors say, this method has not completely determined the causes of model error - but I think that modifications to and expansions of the method will lead to improved insight in the future.
Response. We thank the reviewer for the comment. We agree that the direction the evaluation of models should move is indeed towards processes, but that is not feasible at the moment as would require a new design of the AQMEII activity. A follow up paper focusing on two models only (CMAQ and Chimere) is in preparation for this same special issue dealing with more in depth diagnostic analysis, making use of extra model runs to determine the degree of impact on error of fluxes at the boundaries (deposition, emission and boundary conditions). We have highlighted in the conclusion section our thoughts about future directions of AQ model evaluation.

Major comments
My main criticism of this paper comes about in the presentation of the results. There are multiple figures at such poor resolution that I almost had to give up looking at them. Many of the graphics look like they are plotted in R and I would encourage the authors to save the graphics in a high resolution output (pdf, eps etc).
Response. The figures have been produced as 'tiff' with high resolution and imported onto a word processor. The conversion of the document to 'pdf' for submission produces figures hardly readable. At print out stage we will provide 'pdf' figures as separate files, but for the initial submission we could not find an alternative to converting the document to pdf, which reduced the quality of the figures drastically.

I also think that the reader would benefit from a consistent set of axes limits for plots in EU and NA and the ordering of the models should follow those used in Figures 23 and 26 (i.e. clustering the WRF-Chem variants together and the WRF-CMAQ variants together). When I did this "by hand" I found that the intra model variability was large -no doubt reflecting different model options (chemistry schemes etc).
Response. We have redone all figures and used same axis limits for both continents where possible (not for CO, for example as the EU bias is one order of magnitude higher than in NA). The models have been grouped following the reviewer's suggestion.

My other minor criticism is the choice of the domains. I just wonder if country specific (or State specific in the USA) boundaries were used could we learn more about emission estimate biases? I would imagine that the averaging over states and countries in the current classification will smear out these effects to some extent depending on their heterogeneity.
Response. The reviewer is right, that is indeed the case. That of countries borders is an issue we have dealt with in the previous analyses of AQMEII, especially in the Solazzo and Galmarini (2015) paper, where we showed that not only the results of the models for Europe tend to group by countries due to country-specific

emission profiles, but also the measurements showed very robust clustering properties by network (in North America) and, again, by country in Europe (Airbase network) due to lack of harmonised reporting and instrumental settings.

Nonetheless, in the context of a multi-pollutant and multi-scale screening analysis of regional scale models such as the one presented here, country-based investigation would have required much more space in an already lengthy manuscript. One of the aims of the cluster analysis described in section 2.3.2 is to identify sub-regions where the LT and SY components do present homogeneous features, allowing to overcome country-specific discussion. For these two components, thus, we do not expect heterogeneity induced by emissions. The effect of heterogeneity of emissions (for Europe) is therefore limited to the DU component. The following text has been added to the revised section 2.3.2:

*'As noted in the introduction, unsupervised hierarchical clustering was used to determine sub-regions where the LT and SY components showed similar characteristics – spatial averaging within these sub-regions was carried out due to the similarity of the observation data within these regions implying they will experience common physical and chemical characteristics. Errors due to the heterogeneity induced by country-specific emission profiles (in EU) are therefore included in the DU component.'.*

Minor comments

line 138: Delete the first "known". Done as suggested

line 162/Section 2.2 opening paragraph: There are other spectral filtering methods (e.g. Bowdalo et al., 2016 ACP). A comment on these would be useful and why the kz approach has been selected. Done as suggested

line 191: re-phrase the sentence starting "A clear-cut...". Done

line 197: add "do" after "they". Done as suggested

line 458: what is the source of the meteorological data that you compare against? As mentioned in the Acknowledgements: 'Data from meteorological station monitoring networks were provided by NOAA and Environment Canada (for the US and Canadian meteorological network data) and the National Center for Atmospheric Research (NCAR) data support section'.

line 647: It's not clear what you mean by timescale here? Do you mean the e-folding. Removed

line 1043 lifetime? Should that not depend on the concentration of NO? 20-30 days is an averaged value to provide an overall sense of the time-scale involved.

line 699: remove duplicate round bracket. Done as suggested

line 726: insert "is" after "bias". Done as suggested

line 728, 826, 823, ...: add space between mixing ration and units (e.g. 11.5ppb -> 11.5 ppb). This is a common error so please search the document for this. Done as suggested

line 766: insert space "Figure17and". Done as suggested

line 824: Do the authors really believe that vertical mixing can be analysed through something as complex as ozone? I would suggest that vertical mixing needs sources with no chemistry to be understood (e.g. Rn or Pb). Rephrased

lines 841: Is this really good? It was overlooked. It has been clarified now that CMAQ overestimates surface observations.

line 858: "sulphates" should be "sulfates". Done as suggested

line 881: What about oxidants? Could the corr($bias_{O_3}$, $bias_{SO_2}$) be useful? We could not find any consistent pattern relating $bias_{O_3}$ with $bias_{SO_2}$

line 928: What about temperature effects? Could corr($bias_{Temp}$, $bias_{PM}$) be useful? We have summarised the

information of $corr(bias_{Temp}, bias_{PM10})$ in the revised text (section 3.3.6). We could not detect any consistent pattern relating $bias_{Temp}$ to $bias_{PM2.5}$. We have added the following text to the revised manuscript:

*'The analysis of $corr(bias_{Temp}, bias_{PM10})_{LT}$ shows that the error of these two variables are related, especially during the spring months and more consistently in EU3 (up to 0.74 for the WRF/Chem1 model) and during autumn in EU1 (the bias of Temp and the bias of $PM_{10}$ are anti-correlated up to -0.67 for CMAQ1). Other models (e.g. the CAMx model), on the other hand, do not show any significant correlation.'*